# DeltaProduct: Improving State-Tracking in Linear RNNs via Householder Products

**Julien Siems**[*◇], **Timur Carstensen**[*◇♣], **Arber Zela**[◇],
**Frank Hutter**[△♣◇], **Massimiliano Pontil**[♡♠], **Riccardo Grazzi**[*†★]
Equal contribution[*], University of Freiburg[◇], ELLIS Institute Tübingen[♣], Microsoft Research[★]
Prior Labs[△], CSML, Istituto Italiano di Tecnologia[♡], AI Centre, University College London[♠]
juliensiems@gmail.com   timurcarstensen@gmail.com   riccardograzzi4@gmail.com

## Abstract

Linear Recurrent Neural Networks (linear RNNs) have emerged as competitive alternatives to Transformers for sequence modeling, offering efficient training and linear-time inference. However, existing architectures face a fundamental trade-off between expressivity and efficiency, dictated by the structure of their state-transition matrices. Diagonal matrices, used in models such as Mamba, GLA, or mLSTM, yield fast runtime but have limited expressivity. To address this, recent architectures such as DeltaNet and RWKV-7 adopted a diagonal plus rank–1 structure, which allows simultaneous token and channel mixing, improving associative recall and, as recently shown, state-tracking when allowing state-transition matrices to have negative eigenvalues. Building on the interpretation of DeltaNet's recurrence as performing one step of online gradient descent per token on an associative recall loss, we introduce DeltaProduct, which instead takes multiple ($n_h$) steps per token. This naturally leads to diagonal plus rank–$n_h$ state-transition matrices, formed as products of $n_h$ generalized Householder transformations, providing a tunable mechanism to balance expressivity and efficiency. We provide a detailed theoretical characterization of the state-tracking capability of DeltaProduct in finite precision, showing how it improves by increasing $n_h$. Our extensive experiments demonstrate that DeltaProduct outperforms DeltaNet in both state-tracking and language modeling, while also showing significantly improved length extrapolation capabilities.

## 1 Introduction

The Transformer architecture [1] has revolutionized natural language processing through its self-attention mechanism, enabling both parallel computation across the sequence length and effective context retrieval. Consequently, Transformers have largely replaced recurrent models like LSTMs [2], which exhibit slower training and poorer retrieval performance. However, the quadratic computational complexity of Transformers with sequence length presents challenges when dealing with longer sequences. Linear RNNs have emerged as a promising solution that combines parallel training across the sequence length with linear inference-time complexity. At the core of these models are the state-transition matrices governing the recurrence, which fundamentally determine the expressivity of a linear RNN [3]. Early linear RNNs like S4 [4] and LRU [5] used token-independent state-transition matrices. For superior expressivity, current linear RNNs now exclusively use token-dependent matrices. Among these Mamba [6, 7], GLA [8], and mLSTM [9] use diagonal state-transition matrices for efficient sequence processing. Newer architectures have incorporated non-diagonal structures, often diagonal plus rank-1, enabling simultaneous mixing of

---

[†]Work started while at Istituto Italiano di Tecnologia.

39th Conference on Neural Information Processing Systems (NeurIPS 2025).

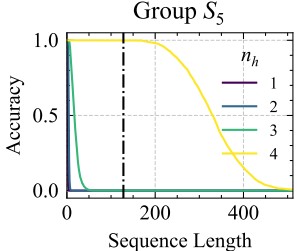
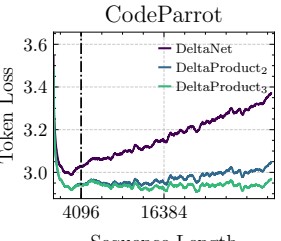

Figure 1: (*Left*) DeltaProduct$_{n_h}$ learns higher-order permutation groups like $S_5$ in one layer, while DeltaNet ($n_h$=1) is limited to $S_2$ (parity). (*Right*) Length extrapolation of DeltaProduct improves significantly with higher $n_h$.

information across both tokens and channels of the hidden state. This innovation has led to more expressive models such as (Gated) DeltaNet [10, 11], TTT-Linear [12], RWKV-7 [13], and Titans [14], which demonstrate superior language modeling and in-context retrieval performance, often with only a reasonable decrease in training efficiency.

Recent work has revealed a fundamental trade-off between training efficiency and expressivity of linear RNNs, dictated by the structure of their state-transition matrices [3, 15, 16]. Models with diagonal state-transition matrices, such as Mamba and GLA, are highly efficient to train but face severe expressivity limitations - for instance, they cannot perform addition modulo 3 on sequences of arbitrary length in finite precision [16, Theorem 2]. Transformers also face similar limitations [17, 18], since they can be seen as special linear RNNs with a state-transition matrix equal to the identity, albeit with an infinite dimensional state [19]. DeltaNet partially overcomes these limitations through generalized Householder matrices, achieving greater expressivity, though it still requires multiple layers for some tasks. At the other extreme, linear RNNs with full state-transition matrices offer maximal expressivity [20], capable of recognizing any regular language with a single layer [3], but are prohibitively expensive to train.

To bridge this gap, we propose *DeltaProduct*, a method that balances expressivity and efficiency of the recurrence computation. While DeltaNet's recurrence performs a single gradient descent step per token on the squared loss of a linear key-to-value mapping [10, 21], DeltaProduct takes $n_h$ gradient steps using additional keys and values, yielding state-transition matrices that are products of $n_h$ generalized Householder matrices. This connection between the number of optimization steps and the matrix structure provides an elegant way to interpolate between diagonal and dense matrices: increasing the number of gradient steps automatically increases the number of Householder matrices in the product, providing a tunable mechanism to control the recurrence's expressivity. DeltaProduct enables precise control over the norm of state transition matrices, ensuring it remains $\leq 1$ to maintain stability during training on long sequences. We contribute DeltaProduct to the flash-linear-attention library [22], our experiment code is provided here.

Concretely, we make the following contributions:

- We propose *(Gated) DeltaProduct*, which generalizes (Gated) DeltaNet by using products of generalized Householder transformations as state-transition matrices (Section 4).

- We provide a detailed theoretical characterization of the expressivity of DeltaProduct in finite precision and how it improves by increasing $n_h$ (Section 4.1). Notably, we prove that for any $n_h \geq 1$, DeltaProduct with at most 4 layers (3 if $n_h \geq 2$) can solve any group word problem, and Gated DeltaProduct with a finite number of layers can recognize any regular language.

- We empirically validate DeltaProduct's superior performance across multiple domains: solving complex state-tracking tasks beyond DeltaNet's capabilities (see Figure 1) and improving language modeling performance with significantly enhanced length extrapolation (Section 5), which we study through analysis of the hidden state's effective rank.

## 2   Background

**Linear RNNs.** Linear RNNs consist of stacked layers, each processing an input sequence of vectors $\boldsymbol{x}_1, \ldots, \boldsymbol{x}_t \in \mathbb{R}^l$ (output of the previous layer) to produce an output sequence $\hat{\boldsymbol{y}}_1, \ldots, \hat{\boldsymbol{y}}_t \in \mathbb{R}^p$. We write the forward pass of each layer placing emphasis on the linear recurrence (as in [16])

$$\boldsymbol{H}_i = \boldsymbol{A}(\boldsymbol{x}_i)\boldsymbol{H}_{i-1} + \boldsymbol{B}(\boldsymbol{x}_i), \qquad \hat{\boldsymbol{y}}_i = \text{dec}(\boldsymbol{H}_i, \boldsymbol{x}_i) \qquad \text{where } i \in 1, \ldots, t \qquad (1)$$

$\boldsymbol{H}_0 \in \mathbb{R}^{n \times d}$ is the initial hidden state, $\boldsymbol{A} : \mathbb{R}^l \to \mathbb{R}^{n \times n}$ maps the input to a state-transition matrix, $\boldsymbol{B} : \mathbb{R}^l \to \mathbb{R}^{n \times d}$ controls the information added to the new hidden state, and dec $: \mathbb{R}^{n \times d} \times$

$\mathbb{R}^l \to \mathbb{R}^p$ determines the output of the recurrence. The functions $\boldsymbol{A}$, $\boldsymbol{B}$, and dec are learnable, with dec typically containing a feedforward neural network. Different linear RNN variants are distinguished by their specific implementations of these functions. For example, Mamba [6, 7], GLA [8], and mLSTM [9] use variations of a diagonal state-transition matrix $\boldsymbol{A}(\boldsymbol{x}_i)$. The linearity of the recurrence allows it to be parallelized along the sequence length, either via a chunkwise parallel form [8, 23, 24] or using a parallel scan [6, 25–28]. For a comparison of different linear RNN architectures see Yang et al. [10, Table 4].

**DeltaNet.** We base our work on the DeltaNet architecture [29, 30], which has recently seen renewed interest through the work of Yang et al. [10, 11] who demonstrate how to parallelize DeltaNet across the sequence length on GPUs. The DeltaNet recurrence is parameterized as

$$\boldsymbol{A}(\boldsymbol{x}_i) = \boldsymbol{I} - \beta_i \boldsymbol{k}_i \boldsymbol{k}_i^\top, \; \boldsymbol{B}(\boldsymbol{x}_i) = \beta_i \boldsymbol{k}_i \boldsymbol{v}_i^\top, \; \mathrm{dec}(\boldsymbol{H}_i, \boldsymbol{x}_i) = \psi(\boldsymbol{H}_i^\top \boldsymbol{q}_i) \tag{2}$$

where $\beta_i \in [0, 1]$, $\boldsymbol{q}_i, \boldsymbol{k}_i \in \mathbb{R}^n$ (with $\|\boldsymbol{q}_i\| = \|\boldsymbol{k}_i\| = 1$), $\boldsymbol{v}_i \in \mathbb{R}^d$ are outputs of learnable functions of $\boldsymbol{x}_i$. $\boldsymbol{A}(\boldsymbol{x}_i)$ is a generalized Householder transformation [31], which is symmetric and has eigenvalues 1 (multiplicity $n - 1$) and $1 - \beta_i$ (multiplicity 1). From a geometric perspective, $\beta_i$ determines the transformation type, interpolating between identity ($\beta_i = 0$) and projection ($\beta_i = 1$). DeltaNet also has a natural interpretation from an online learning perspective [10], as each step of its recurrence is one step of online gradient descent on a quadratic loss with step size $\beta_i$:

$$\mathcal{L}_i(\boldsymbol{H}) = 1/2\|\boldsymbol{H}^\top \boldsymbol{k}_i - \boldsymbol{v}_i\|_2^2; \; \boldsymbol{H}_i = \boldsymbol{H}_{i-1} - \beta_i \nabla \mathcal{L}_i(\boldsymbol{H}_{i-1}) = \boldsymbol{H}_{i-1} - \beta_i \, \boldsymbol{k}_i(\boldsymbol{k}_i^\top \boldsymbol{H}_{i-1} - \boldsymbol{v}_i^\top)$$

$$\text{DeltaNet: } \boldsymbol{H}_i = (I - \beta_i \boldsymbol{k}_i \boldsymbol{k}_i^\top)\boldsymbol{H}_{i-1} + \beta_i \boldsymbol{k}_i \boldsymbol{v}_i^\top$$

**State-Tracking and Word Problems.** State-tracking is the ability of a model to keep track of the state of a system while only observing the updates that are applied to it. It can be modeled as a monoid word problem, which consists in mapping sequences $x_1, \ldots, x_t$, with $x_i$ being an element of a monoid $G$, into sequences $y_1, \ldots, y_t$, where $y_i = x_i \cdot x_{i-1} \cdots x_1$ and $\cdot$ is the associative operation of the monoid. Recognizing a regular language can be accomplished by solving the word problem of a finite monoid associated to the language and will be the focus of this work. Problems where $G$ is also a group (group word problems) with finite elements, are notoriously hard to solve for both Transformers and linear RNNs. Group word problems for the symmetric or permutation groups are particularly important, since any group is isomorphic to a subgroup of a symmetric group. For instance, if we denote with $S_n$ the group of permutations of $n$ elements, parity corresponds to the $S_2$ word problem, which cannot be solved in finite precision by Transformers [17] and diagonal Linear RNNs [15] with positive values, while the $S_5$ word problem cannot be solved by these models even when the precision can grow logarithmically with the sequence length, since both Transformers and Linear RNNs belong to the $\mathrm{TC}^0$ circuit complexity class while $S_5$ is in $\mathrm{NC}^1$ [3, 18, 32]. In contrast, with unconstrained, full state-transition matrices any regular language can be recognized in one layer (see e.g. [3, Theorem 5.2]). However, training using full unstructured matrices is very inefficient and also unstable without any control on the norm.

## 3   Related Work

Linear RNNs have recently been studied from two main perspectives: state-space models and causal linear attention. State-space models, originating from continuous dynamical systems, inspired variants such as S4 [4], H4 [33], and LRU [5] (see Tiezzi et al. [34] for a comprehensive survey). Models like Mamba [6, 7] further enhance these by incorporating input-dependent gating mechanisms, significantly improving language modeling performance. In parallel, Katharopoulos et al. [19] showed that causal linear attention Transformers can be reformulated as RNNs with linear sequence-length scaling. Following this, Gated Linear Attention (GLA) [8] introduced gating mechanisms similar to Mamba. Recent studies explored more expressive recurrences via non-diagonal transition matrices, such as DeltaNet [10, 29, 35], TTT-Linear [12], RWKV-7 [13], B'MOJO [36], and Titans [14]. Additionally, Beck et al. [9] introduced xLSTM, combining linear and nonlinear RNN architectures inspired by LSTM [2]. Another line of work explores recurrences in depth, which have been shown to increase the expressivity and reasoning capabilities [37, 38]. For instance, concurrent work explores how fixed-point iterations of a diagonal linear RNN can increase its expressivity turning it non-linear at the fixed-point [39, 40]. Unlike our approach, which enhances the expressivity by increasing the complexity of the linear recurrence, their approach works by applying the same recurrence multiple times, effectively increasing the depth of the model without increasing the parameter count. This approach is orthogonal to ours and the two can be potentially combined.

Products of structured matrices [41] have previously been used in the state-update of *non-linear* RNNs – including (Givens) rotation matrices [42–44], Kronecker products [45], Householder reflections [46]—chosen for their orthogonal, norm-preserving properties that encourage long-term dependency learning [47, 48]. Recently, Biegun et al. [49] applied rotation matrices as state-transition matrices in non-selective state-space models. In contrast, DeltaProduct uses a linear recurrence with state-transition matrices adaptive to the current token, expressed as products of generalized Householder matrices.

**State-Tracking.** Recent work by Grazzi et al. [16] demonstrates that expanding the eigenvalue range of linear RNNs' state transition matrices from $[0, 1]$ to $[-1, 1]$ significantly enhances their expressivity. They show how DeltaNet's eigenvalue range can be extended from $[0, 1]$ to $[-1, 1]$, simply by multiplying $\beta_i$ by 2, allowing DeltaNet to perform reflections when $\beta_i = 2$, which enables it to handle state-tracking tasks such as parity checking and, more generally, any *group word problem* where each element of the input sequence corresponds to a permutation of at most two elements, while for other tasks, DeltaNet requires multiple layers [16, Theorem 2 and 6]. Their theoretical results also cover the products of Householders used as state-transition matrices of DeltaProduct, showing that they allow to solve any group word problem in one layer (Theorem 3) and recognize any regular language (Theorem 4), when $n_h$ is large enough and with a finite number of layers. Here, we extend that work by providing experimental evidence of the benefit of larger $n_h$ and, leveraging the analysis by Peng et al. [13], more refined theoretical results on the expressivity, e.g. with a greatly improved dependency on $n_h$. The recent RWKV-7 [13] uses a state transition matrix of the form $\mathrm{diag}(\boldsymbol{w}_t) - c\boldsymbol{k}_t(\boldsymbol{k}_t \odot \boldsymbol{a}_t)^\top$, with $\|\boldsymbol{k}_t\| = 1$, $\boldsymbol{a}_t, \boldsymbol{w}_t \in [0, 1]^n$ and $c \in \{1, 2\}$, which provides a potentially asymmetric rank-1 update in contrast to DeltaNet's symmetric update, allowing it to recognize any regular language with only 4 layers. However, the increased expressivity comes at the cost of losing the guarantee on the stability of the recurrence. In contrast, (Gated) DeltaProduct has a stable recurrence since the spectral norm of every state-transition matrix is always $\leq 1$.

## 4  DeltaProduct

While DeltaNet's recurrence can be seen as performing one step of online gradient descent per token, DeltaProduct builds upon DeltaNet by further refining the hidden state by taking *multiple steps per token*. This naturally leads to a more expressive state-transition matrix formed as a product of generalized Householder matrices, where each additional step expands the range of achievable linear transformations. Formally, for each input token $\boldsymbol{x}_i$ to the layer we generate $n_h$ keys as $\boldsymbol{k}_{i,j} = \psi(\boldsymbol{W}_j\boldsymbol{x}_i)/\|\psi(\boldsymbol{W}_j\boldsymbol{x}_i)\|_2$, $n_h$ values as $\boldsymbol{v}_{i,j} = \boldsymbol{V}_j\boldsymbol{x}_i$, and $n_h$ betas as $\beta_{i,j} = \phi(\boldsymbol{U}_j\boldsymbol{x}_i)$ where $\boldsymbol{W}_j, \boldsymbol{V}_j, \boldsymbol{U}_j$, are learnable weight matrices specific to the $j$-th gradient step, $\psi$ is a nonlinearity (we pick SiLU as in DeltaNet), while $\phi$ is either the sigmoid or $2\times$ the sigmoid to increase the expressivity. Then, we compute $n_h$ gradient descent steps using the losses $\mathcal{L}_{i,j}(\boldsymbol{H}) = \|\boldsymbol{H}^\top \boldsymbol{k}_{i,j} - \boldsymbol{v}_{i,j}\|_2^2/2$, i.e., for $j = 1 \ldots n_h$

$$\boldsymbol{H}_{i,j} = \boldsymbol{H}_{i,j-1} - \beta_{i,j}\nabla\mathcal{L}_{i,j}(\boldsymbol{H}_{i,j-1}) = \left(\boldsymbol{I} - \beta_{i,j}\boldsymbol{k}_{i,j}\boldsymbol{k}_{i,j}^\top\right)\boldsymbol{H}_{i,j-1} + \beta_{i,j}\boldsymbol{k}_{i,j}\boldsymbol{v}_{i,j}^\top,$$

where $\boldsymbol{H}_{i,0} = \boldsymbol{H}_{i-1}$ and $\boldsymbol{H}_{i,n_h} = \boldsymbol{H}_i$. Unrolling, we get $\boldsymbol{H}_i = \boldsymbol{A}(\boldsymbol{x}_i)\boldsymbol{H}_{i-1} + \boldsymbol{B}(\boldsymbol{x}_i)$ with

$$\boldsymbol{A}(\boldsymbol{x}_i) = \prod_{j=1}^{n_h}\left(\boldsymbol{I} - \beta_{i,j}\,\boldsymbol{k}_{i,j}\boldsymbol{k}_{i,j}^\top\right), \quad \boldsymbol{B}(\boldsymbol{x}_i) = \sum_{j=1}^{n_h}\left(\prod_{k=j+1}^{n_h}\left(\boldsymbol{I} - \beta_{i,k}\,\boldsymbol{k}_{i,k}\boldsymbol{k}_{i,k}^\top\right)\right)\beta_{i,j}\,\boldsymbol{k}_{i,j}\boldsymbol{v}_{i,j}^\top. \quad (3)$$

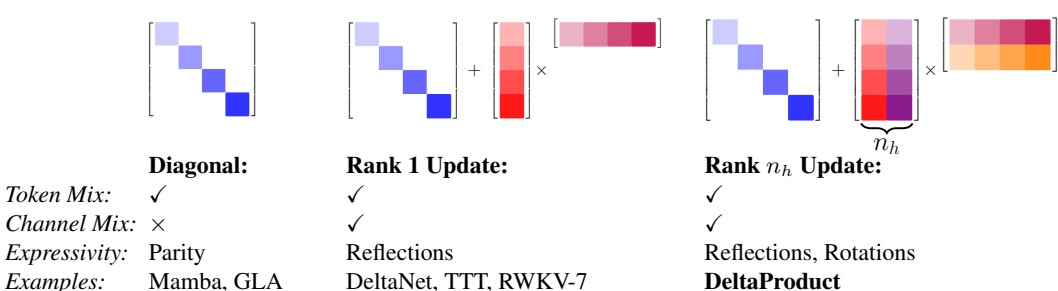

|  | **Diagonal:** | **Rank 1 Update:** | **Rank $n_h$ Update:** |
|---|---|---|---|
| *Token Mix:* | ✓ | ✓ | ✓ |
| *Channel Mix:* | × | ✓ | ✓ |
| *Expressivity:* | Parity | Reflections | Reflections, Rotations |
| *Examples:* | Mamba, GLA | DeltaNet, TTT, RWKV-7 | **DeltaProduct** |

Figure 2: Overview of state-transition matrices $\boldsymbol{A}(\boldsymbol{x}_i)$ in linear RNNs.

Hence, by taking multiple gradient descent steps per token, DeltaProduct's state-transition matrices are *products of generalized Householder transformations*, and by expanding such a product, $\boldsymbol{A}(\boldsymbol{x}_i)$ takes the form of identity plus a matrix of rank at most $n_h$ as shown in Figure 2. As DeltaNet extends to Gated DeltaNet by incorporating a forget gate [11], DeltaProduct can similarly be extended to *Gated DeltaProduct* by letting $\boldsymbol{A}(\boldsymbol{x}_i) = g_i \prod_{j=1}^{n_h}(\boldsymbol{I} - \beta_{i,j}\,\boldsymbol{k}_{i,j}\boldsymbol{k}_{i,j}^\top)$ where the scalar gate $g_i \in [0,1]$ is adopted from Mamba 2 [7] and $\boldsymbol{B}(\boldsymbol{x}_i)$ remains unchanged. This formulation enables DeltaProduct to interpolate between generalized Householder ($n_h = 1$ as in DeltaNet) and dense matrices (of norm $\leq 1$), since increasing $n_h$ can increase the rank of $\boldsymbol{A}(\boldsymbol{x}_i)$.

**Expressivity of Householder products.** While any state transition matrix of DeltaNet can model a single Householder reflection (with $\beta_i = 2$), DeltaProduct's can model any orthogonal matrix. This is a consequence of the Cartan-Dieudonné theorem, which establishes that any $n \times n$ orthogonal matrix can be expressed as a product of at most $n$ reflections (as illustrated in Figure 3 for $n = 2$). The Householder product exhibits interesting properties in special cases. When all Householder keys are identical, the product simplifies to a single Householder with a scaled beta parameter, offering no additional expressivity (Prop. 1.1). Conversely, when the keys are mutually orthogonal, the Householder product simplifies to an identity plus a symmetric rank $n_h$ matrix (Prop. 1.2). Only when the keys are non-trivially linearly dependent can we obtain non-symmetric matrices, potentially yielding complex eigenvalues (Prop. 1.3). An important consequence of using Householder products is that it allows us to effectively bound the norm of $\boldsymbol{A}(\boldsymbol{x}_i)$. This is because the norm of the product is upper bounded by the product of the norms (each $\leq 1$), which ensures the stability of the recurrence [16, Prop. 1.1]. This bound would not be possible with the more direct parametrization $\boldsymbol{A}(\boldsymbol{x}_i) = I - \sum_{j=1}^{n_h}\beta_{i,j}\boldsymbol{k}_{i,j}\boldsymbol{k}_{i,j}^\top$, which also restricts the matrix to be symmetric.

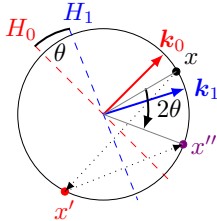

Figure 3: Two reflections produce a 2D rotation: Reflecting $x$ across planes $H_0$ and $H_1$ (with normals $\boldsymbol{k}_0$ and $\boldsymbol{k}_1$) yields a rotation by $2\theta$, where $\theta$ is the angle between the planes.

## 4.1 State-Tracking Capabilities of (Gated) DeltaProduct

We present two theorems that characterize the state-tracking capabilities of DeltaProduct. Compared to Grazzi et al. [16, Theorem 3 and 4], we focus on results that hold for any $n_h \geq 1$. We defer proofs, and more details to Appendix B, where we also include results on dihedral groups (Theorem 7) and on finite subgroups of the orthogonal and special orthogonal groups (Theorem 4).

**Theorem 1.** *For any $n \in \mathbb{N}$ there exists a DeltaProduct model with one of the following configurations that can solve the **word problem** of the symmetric group $S_n$: (i) one layer with $n_h = n-1$ [16, Theorem 3] (ii) 3 layers with $n_h > 1$ (iii) 4 layers with $n_h = 1$. The construction for (ii) and (iii) requires that the MLP at the second last layer computes a lookup-table of size $2m \times (n!)^{2m}$, function of the last $2m$ input tokens and the position modulo $2m$ with $m = \lceil (n-1)/n_h \rceil$.*

**Theorem 2.** *For any $n_h \geq 1$ and any regular language, there exists a Gated DeltaProduct model with a finite number of layers (dependent on the language) that recognizes it.*

The proof for Theorem 1 uses the same idea as the construction for the theoretical results of Peng et al. [13] for RWKV-7. Each element of $S_n$ can be mapped to a permutation matrix, but DeltaProduct's state transition matrices can only model permutations of up to $n_h + 1$ elements. Therefore, if $n_h + 1 < n$, early layers decompose each product of $m = \lceil (n-1)/n_h \rceil$ consecutive permutations into $m$ simpler permutations, which are applied in the recurrence of the last layer but in a delayed fashion. To get such a decomposition and account for the delay, the MLP at the second-last layer computes a potentially large lookup table, function of the past $2m$ tokens and the position modulo $2m$. To prove Theorem 2, we use the Krohn-Rhodes decomposition [50], similarly to Grazzi et al. [16, Theorem 4], where each automaton is decomposed into multiple *permutation-reset* automata, and model each using the same technique of Theorem 1, exploiting gates for the resets.

**Comparison to other non-diagonal Linear RNNs.** Table 1 provides a comparison of the expressivity of different non-diagonal linear RNNs. (Gated) DeltaProduct with $n_h > 1$ has improved expressivity compared to DeltaNet, and, up to 3 layers, even compared to RWKV-7. Moreover, increasing $n_h$ has clear benefits: reducing the number of layers or the size of the lookup table. Since DeltaProduct can solve the $S_5$ word problem, it is outside of the TC$^0$ complexity class, just as RWKV-7. One might expect DeltaProduct to be able to model any useful state-transition matrix

Table 1: Expressivity of non-diagonal Linear RNNs shown through the formal language problems they can solve in finite precision. $S_n, \mathbb{Z}_n, D_n$ are the symmetric, cyclic and dihedral groups of order $n$, while $\mathrm{O}(n), \mathrm{SO}(n)$ are the orthogonal and special orthogonal group of order $n$. For $S_n$, "only $k$-permutations" means that input sequences can contain only permutations of up to $k$ elements. LS is the size of the lookup table computed in the second-last layer's MLP. $|\Sigma|$ and $|Q|$ are the sizes of the alphabet and set of states of a finite state automaton recognizing the regular language. Gated variants' state updates can also model constant (*reset*) transitions.

| Layers | **(Gated) DeltaNet** | **RWKV-7** | **(Gated) DeltaProduct**$_{n_h > 1}$ |
|---|---|---|---|
| 1 | $S_n$ only 2-permutations.[a] | $S_n$ only 2-permutations | $S_n$ only $(n_h + 1)$-permut.[a] Finite subgroups of $\mathrm{O}(n_h)$, $\mathrm{SO}(n_h + 1)$ if $n_h$ is even.[g] |
| 2 | $\mathbb{Z}_n$[b],$D_n$[c] | $\mathbb{Z}_n, D_n$ | |
| 3 | | | $S_n$ with LS $2m \times (n!)^{2m}$ where $m = \lceil (n-1)/n_h \rceil$.[d] |
| 4 | $S_n$ with LS $2(n-1) \times (n!)^{2(n-1)}$.[d] | $S_n$ with LS as DeltaNet. Reg. lang. with LS $2|Q| \times (|\Sigma|)^{2|Q|}$.[e] | |
| $f(|Q|)$ | Gated: Regular languages[f] | | Gated: Regular languages[f] |

[a] [16, Thm. 3]  [b] [16, Thm. 6]  [c] Thm. 7.  [d] Thm. 1.  [e] [13, Thm. 3]  [f] Thm. 2  [g] Thm. 4

since it can model updates of arbitrarily high rank when $n_h$ is equal to the number of rows of the hidden state. This is because DeltaProduct's state-transition matrices $\boldsymbol{A}(\boldsymbol{x}_i)$ satisfy the spectral norm condition $\|\boldsymbol{A}(\boldsymbol{x}_i)\| = \max_{\|\boldsymbol{y}\|=1} \|\boldsymbol{A}(\boldsymbol{x}_i)\boldsymbol{y}\|_2 \leq 1$, ensuring a stable recurrence. RWKV-7 relaxes this constraint and can represent matrices with higher spectral norms. In particular, copy matrices – identity matrices where one column is replaced by another – with $\|\boldsymbol{A}(\boldsymbol{x}_i)\| = \sqrt{2}$ (when $c = 2$), allowing RWKV-7 to recognize any regular language in just four layers. However, this may lead to instability in the recurrence (see Appendix B.6). We could enhance expressivity at the cost of stability by replacing the Householder matrices in DeltaProduct with RWKV-7's state transition matrices. This modification enables us to prove a result analogous to Theorem 1, but for regular languages rather than group word problems—see Appendix B.4 for details. Specifically, this approach allows the resulting linear RNN to recognize any regular language within a *single layer*, provided $n_h$ is sufficiently large.

**Remark 1.** *For any **regular language** recognized by a finite-state automaton (FSA) having $n$ states there exists a **one layer linear RNN** using $n_h = n$ products of RWKV-7 matrices as state-transition matrices that can recognize it. This is because a linear RNN with unconstrained state-transition matrices can recognize any regular language in a single layer [3, Theorem 5.2] by modeling FSA evaluation through matrix-vector products [51]. Peng et al. [13, Lemma 3] further showed that any transition matrix of an FSA with $n$ states can be expressed as products of $n$ matrices, each of which is either a swap, copy, or the identity matrix, all of which are representable by an RWKV-7 matrix.*

The above discussion oulines a trade-off between the expressivity of RWKV-7 matrices and the guaranteed stability of generalized Householders used in DeltaProduct. It is an open question whether there exists a continuous parameterization of state-transition matrices which yields stable recurrences and still allows to recognize any regular language in a finite and fixed number of layers.

## 5 Experiments

We evaluate DeltaProduct on state-tracking and standard language modeling to assess its expressivity and efficiency. Throughout the experiments we use either the suffix $[-1, 1]$ or $[0, 1]$ after each method, to denote the eigenvalue ranges of its state transition matrices. We present additional experiments on languages of different levels of the Chomsky hierarchy [52] in Appendix C.2.

### 5.1 Implementation

We use the same macro architecture used by Gated DeltaNet. Since each step of (Gated) DeltaProduct follows the same recurrence structure as (Gated) DeltaNet, we can reuse its implementation written in Triton [53], available through the FLASH-LINEAR-ATTENTION library [22], which uses the chunk-wise parallel form for the recurrence.

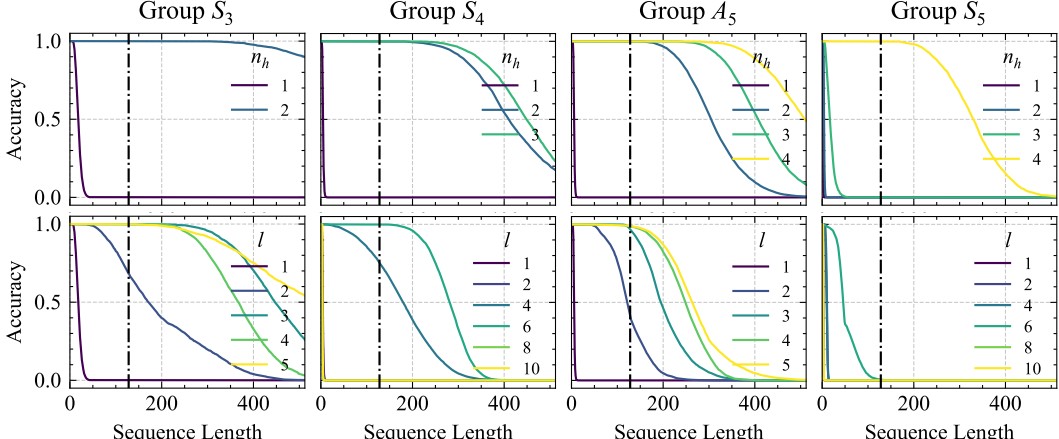

Figure 5: Accuracy on state-tracking tasks for permutation groups $S_3$, $S_4$, $A_5$, and $S_5$, plotted against sequence length (x-axis). *(Top row)* Varying the number of Householder products $n_h$ for a single layer DeltaProduct$_{n_h}[-1,1]$. *(Bottom row)* Varying the number of layers $l$ of DeltaProduct$_1[-1,1]$/DeltaNet$[-1,1]$ (single Householder). Dashed vertical line at training context length 128. Higher $n_h$ improves extrapolation to longer sequences of permutations, e.g., $S_3$ can be learned with $n_h = 2$ with a single layer while three layers are required when keeping $n_h = 1$.

However, DeltaProduct differs by using $n_h$ keys, values and betas per token, resulting in a recurrence $n_h$ times longer than DeltaNet's. Therefore, we arrange keys (and similarly values and betas) as $[\mathbf{k}_{1,1}, \ldots, \mathbf{k}_{1,n_h}, \mathbf{k}_{2,1}, \ldots, \mathbf{k}_{2,n_h}, \ldots]$, while for gating we construct the expanded sequence of gates as: $[g_1, 1, \ldots, 1, g_2, 1, \ldots, 1, \ldots]$ where each gate $g_i$ is followed by $(n_h - 1)$ ones to match the number of keys and values, so that we use only one gate for each token. Once the recurrence is evaluated, we keep only every $n_h$-th element of the output, so that the output sequence retains the same length as the input sequence.

**Throughput.** The training (and prefill time) required for the recurrence increases linearly with $n_h$, since we use the same chunk size for the chunkwise parallel form. In contrast, since we keep the embedding dimension fixed, the cost for the MLP following the recurrence does not vary with $n_h$. To remedy the parameter-overhead introduced by the additional key and value projections due to increased $n_h$, we demonstrate the throughput when matching parameters in Figure 4. Matching parameters simply by scaling the head dimension is unfavorable (bottom subplot, $n_h = 2$) since head dimensions that are not a power

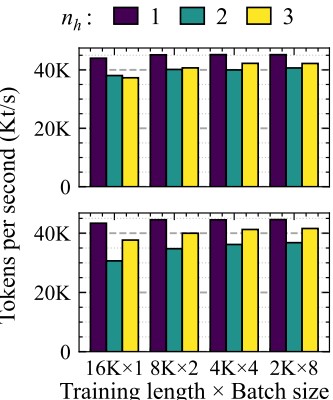

Figure 4: Training throughput of parameter matched 1.3B DeltaProduct$_{n_h}$ on a H100. Matched via: *(Top)* scaling the number of heads, *(Bottom)* scaling the head dimension.

of 2 will get padded to the next power thereof, effectively giving up the remaining dimensions at no reduction in runtime. See Appendix C.3.2 for additional results on smaller models. Note that if $n_h \geq 1$, we could parallelize the recurrence to have a faster runtime also during autoregressive generation. Note that the throughput results are obtained using an optimized Triton kernel implementation (developed by Songlin Yang and Yu Zhang, available in the flash-linear-attention library) that achieves a 20% faster forward pass than DeltaNet's kernel.

### 5.2 State-Tracking

**Setup.** We evaluate DeltaProduct's ability to capture complex state dynamics using group word problems of increasing difficulty, specifically on the permutation groups $S_3$, $S_4$, $A_5$, and $S_5$, as implemented by Merrill et al. [3]. These tasks consist in tracking how a sequence of permutations rearranges elements. An intuitive parallel is the shell game, where one needs to track the position of a hidden object after each shuffle. We train on sequences of 128 permutations and measure extrapolation up to 512. Throughout, we use the extended eigenvalue range, allowing eigenvalues in $[-1, 1]$. We find that DeltaProduct models fail to learn even the training context-length when restricted to

the standard eigenvalue range $[0, 1]$, regardless of the number of Householder transformations $n_h$ as shown in Figure 12. See Appendix C.1 for details on the experiments.

**Single layer, varying $n_h$.** Figure 5 (top row) demonstrates the benefits of increasing the number of Householders $n_h$ per token for a single layer DeltaProduct. Grazzi et al. [16, Theorem 3] presents a construction for permutations of $n$ elements requiring $n - 1$ Householders and keys of size $n$. In agreement, we find that for $S_3$ achieving reliable performance beyond sequence lengths of 128 requires $n_h = 2$, while $S_5$ needs $n_h = 4$. Unexpectedly, $S_4$ and $A_5$ can extrapolate robustly using only $n_h = 2$ despite the theorem suggesting 3 and 4, respectively. This efficiency arises from their isomorphism to subgroups of $\mathrm{SO}(3, \mathbb{R})$, i.e. the group of 3D rotations, [54, Ch. 1, Sec. 2.4] which only require $n_h = 2$ and keys of size 3 (see Theorem 4). Specifically, $S_4$ is isomorphic to the rotation group of the cube (illustrated in Figure 6) and $A_5$ to the rotation group of the dodecahedron. See Appendix C.1 for details on the isomorphisms.

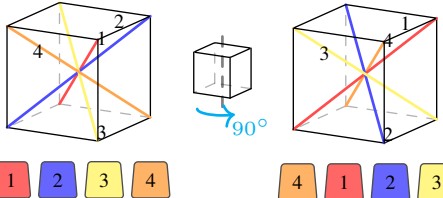

To empirically validate whether DeltaProduct$_2[-1, 1]$ exploits the isomorphism of $S_4$ to the rotation group of the cube, we verified two hypotheses: whether both Householders act as reflections ($\beta_{i,0} = \beta_{i,1} = 2$) composing to form rotations (see Figure 3), and whether the keys are in a three-dimensional subspace. By recording $\beta_{i,0}$ and $\beta_{i,1}$ values (representing the first and second Householder in the product) across all 24 permutations of $S_4$, we find that a single head has indeed learned to use both Householder transformations as reflections where $\beta_{i,0} = \beta_{i,1} = 2$, effectively creating rotation matrices as shown in Ap-

Figure 6: Rotating a cube permutes its diagonals according to the $S_4$ group. This example shows how a $90°$ rotation of the cube leads to the 4-cycle $(1 \to 2 \to 3 \to 4 \to 1)$.

pendix C.1. This pattern is evident in Figure 7 (left), where this head predicts both $\beta_{i,0}$ and $\beta_{i,1}$ approximately at 2, confirming that the model successfully learns to approximate rotations by combining two reflections. Note that the eigenvalues of the Householder product become complex in this case allowing it to perform rotations (Prop 1.3). To further verify whether the keys are in a three-dimensional subspace, we apply Principal Component Analysis [55] to the key vectors of this head. The results in Figure 7 (right) demonstrate that three principal components account for over 95% of the variance in the key space. This finding strongly supports our theoretical understanding, as it indicates that the model primarily operates in a three-dimensional subspace, which aligns with the structure of $\mathrm{SO}(3, \mathbb{R})$.

**Multiple layers, $n_h = 1$.** The bottom row of Figure 5 explores the expressivity of multi-layered DeltaNet$[-1, 1]$ (i.e., $n_h = 1$). While increasing layers with $n_h = 1$ improves performance, it is less effective than increasing $n_h$ and degrades length-extrapolation performance. Specifically, to fit the training context length, $S_3$ required 3 layers, $S_4$ needed 6 layers, and $A_5$ required 3. For $S_5$, even 10 layers proved insufficient. This suggests that simply adding depth is less effective in practice than increasing $n_h$, despite theoretical constructions showing that $S_3$ can be solved with just 2 layers (Theorem 7) and any group word problem can be solved with 4 layers (with a very wide MLP).

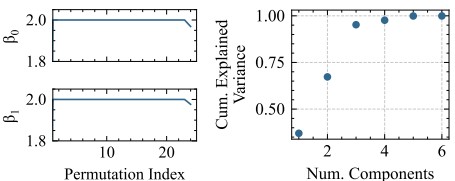

Figure 7: *(Left)* Estimated $\beta$ values for DeltaProduct$_2[-1, 1]$ on all permutations of $S_4$, clustering near 2 (reflection). *(Right)* PCA of key vectors shows that the first three components explain most of the variance.

## 5.3 Language Modeling

**Setup.** We trained two model variants: DeltaProduct$_{n_h}[-1, 1]$ and Gated DeltaProduct$_{n_h}[-1, 1]$ using the FineWeb dataset [57]. We provide details about the training pipeline and hyperparameters in Appendix C.3.1. To assess length extrapolation, we measured the cross-entropy loss beyond the training context length of 4096 tokens on CodeParrot [58] for coding, OpenThoughts-114k-Math [59] for math, and TriviaQA [60] for knowledge retrieval. We evaluated the models using language understanding, reasoning, and retrieval benchmarks from lm-eval-harness [61], with task specifics in Appendix C.3.3. Throughout our experiments we find that the training process remained stable even as we increased $n_h$ (see Appendix C.3.4).

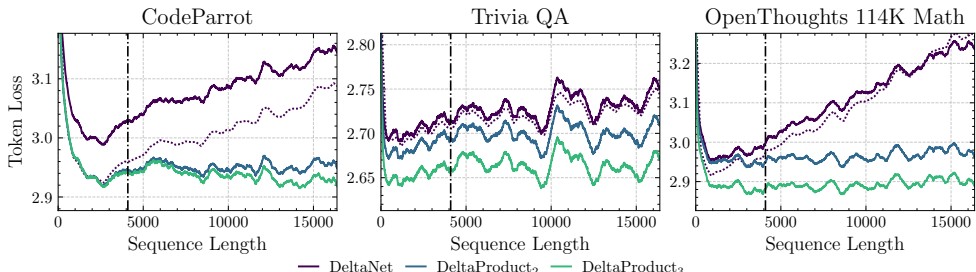

Figure 8: Length extrapolation results. Solid and dashed lines represent models with 8 and 12 heads respectively. Note that DeltaProduct$_2$[$-1, 1$] with 8 heads (392M parameters) matches the parameter count of DeltaNet ($n_h = 1$) with 12 heads (dotted line), while achieving significantly better length extrapolation. For each index of the sequence, we report the moving average over 501 tokens as suggested by Lin et al. [56].

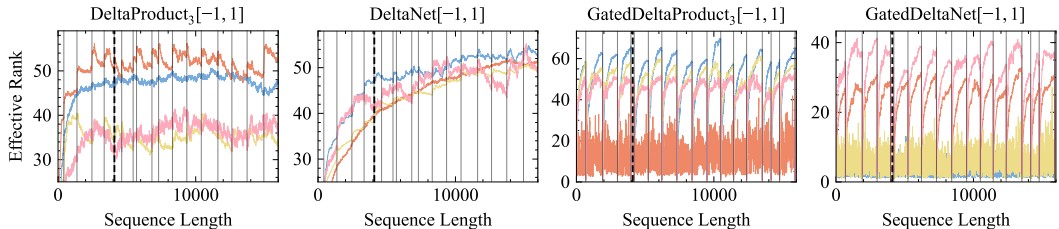

Figure 9: Effective rank of $\boldsymbol{H}_i$ for 4 of 8 heads in layer 20/24 on trivia-qa sequences. Solid vertical lines mark new question-answer pairs; dashed vertical line indicates 4096-token training context length; colored lines show effective rank per head over the sequence.

**Length extrapolation results.** Remarkably, as shown in Figure 8, DeltaProduct's length extrapolation performance increases sharply when going from one to two Householders, and at $n_h = 3$, the performance degradation is minimal across the sequence length. We hypothesize that DeltaProduct achieves better length extrapolation by enhancing DeltaNet's forgetting mechanism. While DeltaNet requires $n$ rank-1 updates to reset its state to zero, DeltaProduct can accelerate this forgetting process by a factor of $n_h$. However, our experiments show that DeltaProduct$_2$[$-1, 1$] still performs better with a forget gate, as demonstrated by its improved results when compared to the non-gated version (see Appendix C.3.5).

**Analyzing state dynamics through effective rank.** To test our hypothesis towards a better forgetting mechanism, we compare how the information density of the hidden state $\boldsymbol{H}_i$ changes over time for (Gated) DeltaProduct$_3$[$-1, 1$] and (Gated) DeltaNet[$-1, 1$]. We propose to measure the information density of $\boldsymbol{H}_i$ using its effective rank [62] defined as $\mathrm{erank}(\boldsymbol{H}_i) = \exp(-\sum_k p_k \log p_k)$, where $p_k = \sigma_k / \sum_i |\sigma_i|$ and $\sigma_i$ is the $i$-th singular value of $\boldsymbol{H}_i$, which satisfies $1 \leq \mathrm{erank}(\boldsymbol{H}_i) \leq \mathrm{rank}(\boldsymbol{H}_i)$. Note that Parnichkun et al. [63] also used the effective rank in the context of linear RNNs, but measured on a different quantity: a linear operator associated to the recurrence. Similarly, Peng et al. [13] conducted an interpretability study on the hidden state of RWKV-7 using the average stable rank [64] across layers. In Figure 9, we present the effective rank of DeltaProduct$_3$, Gated DeltaProduct$_3$, Gated DeltaNet, and DeltaNet across a sequence of tokens from the TriviaQA dataset, which consists of question-answer pairs that test common knowledge. We observe that some heads of DeltaProduct learn to update their state with new information at the beginning of sequence (BOS) tokens and then decay over the rest of the sequence. In comparison, a few heads of Gated DeltaNet and Gated DeltaProduct learn to reduce the

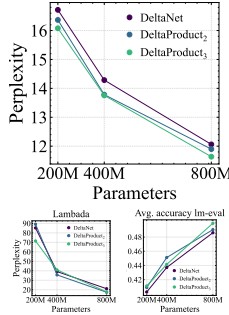

Figure 10: Scaling analysis w.r.t. (*top*) final perplexity on FineWeb, (*bottom*) Lambda and lm-eval tasks.

effective rank of the hidden state close to zero after each BOS token, while the other heads maintain a very low effective rank throughout the sequence. In contrast, DeltaNet's effective rank increases substantially beyond the training context. We attribute DeltaNet's inability to extrapolate to longer sequences to this issue, as the training and extrapolation regimes differ, resulting in a distribution shift. We present additional results for other layers and the CodeParrot dataset in Appendix C.3.5.

**Scaling Analysis.** We test whether it is favorable to increase model size through Householder products as opposed to increasing model capacity through e.g., the head dimension. We adjust either the head dimension in DeltaNet, or the number of Householder products ($n_h$) in DeltaProduct to reach a given size. Figure 10 (top) shows that DeltaProduct scales better in terms of training perplexity. The results on lm-eval-harness tasks, in Figure 10 (bottom), reinforce our findings as DeltaProduct maintains its performance advantage at the largest scale. In total, we test two methods to reach parameter equivalence at each model scale: scaling the number of heads or the head dimension. We find that the latter shows more consistent scaling. Results for scaling the number of heads can be found in Appendix C.3.6. In Appendix C.3.3 we report additional results, including gated variants, where we find that both DeltaProduct and Gated DeltaProduct on average outperform their baseline counterparts (DeltaNet$[-1, 1]$ and Gated DeltaNet$[-1, 1]$) across the considered language modeling benchmarks from lm-eval harness when we increase $n_h$. Interestingly, DeltaProduct$_3[-1, 1]$ achieves comparable performance to Gated DeltaNet$[-1, 1]$, despite lacking a forget gate.

## 6   Conclusion and Future Work

We present DeltaProduct, an extension of DeltaNet that uses products of Householder transformations as state-transition matrices. Our approach bridges the gap between structured and dense matrices, with each recurrence step interpretable as multiple steps of gradient descent on an associative recall loss (compared to DeltaNet's single step). The number of Householder matrices ($n_h$) serves as a tunable parameter balancing expressivity and computational efficiency. Our experiments demonstrate DeltaProduct's superior performance over DeltaNet in state tracking, formal language recognition, and language modeling, with particularly strong length extrapolation results. DeltaProduct represents a promising step towards developing sequence models that are more capable while still remaining scalable. **Limitation.** The main limitation of DeltaProduct is its increased computational cost, which scales linearly with $n_h$ during training. **Future Work.** Future research could explore more expressive and possibly stable matrix parameterizations or an adaptive version of DeltaProduct determining the number of Householders per token similar to Graves [65] in order to reduce computation. The additional parameters introduced with higher $n_h$ could be reduced through LoRA MLPs as done in RWKV-7 [13]. In addition, one could combine DeltaProduct with fixed point RNNs [39, 40]. Our DeltaProduct implementation could be further optimized through custom kernels as suggested in the recent works by Cirone and Salvi [66] or Beck et al. [67]. We also identify promising applications for DeltaProduct in reasoning tasks, where the higher token counts align well with the strength of linear RNNs. Given that state-tracking benefits reasoning tasks [39], future work should examine how increasing $n_h$ affects reasoning.

## Acknowledgements

We would like to thank Songlin Yang, Eddie Bergman, Arjun Krishnakumar, Alma Lindborg, and Julie Naegelen for constructive discussions and feedback. We acknowledge the support and assistance of the Data Science and Computation Facility and its Support Team, in particular Mattia Pini, in using the IIT High-Performance Computing Infrastructure, on which we run part of our experiments. This research was partially supported by the following sources: PNRR MUR Project PE000013 CUP J53C22003010006 "Future Artificial Intelligence Research (FAIR)", funded by the European Union – NextGenerationEU, and EU Project ELSA under grant agreement No. 101070617. TAILOR, a project funded by EU Horizon 2020 research and innovation programme under GA No 952215; the Deutsche Forschungsgemeinschaft (DFG, German Research Foundation) under grant number 417962828; the European Research Council (ERC) Consolidator Grant 'Deep Learning 2.0' (grant no. 10). This research was partially funded by the Deutsche Forschungsgemeinschaft (DFG, German Research Foundation) under grant number 539134284, through EFRE (FEIH 2698644) and the state of Baden-Württemberg. Frank Hutter acknowledges financial support by the Hector Foundation. The authors acknowledge support from ELLIS and ELIZA. Funded by the European Union. The authors gratefully acknowledge the computing time made available to them on the high-performance computers and at the NHR Centers at TU Dresden and KIT. These centers are jointly supported by the Federal Ministry of Research, Technology and Space of Germany and the state governments participating in the NHR. Views and opinions expressed are however those of the author(s) only and do not necessarily reflect those of the European Union or the ERC. Neither the European Union nor the ERC can be held responsible for them.

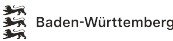 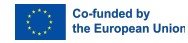

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

## Supplementary Material

The supplementary material is structured as follows:

Appendix A presents a proposition that provides special cases of the generalized Householder product for specific choices of keys and betas and characterizes the spectrum of the product of two generalized Householders.

Appendix B characterizes the expressivity of DeltaProduct.

- B.2 demonstrates how DeltaProduct can solve any group word problem in a single layer when $n_h$ is sufficiently high, or alternatively using up to 4 layers when $n_h$ is limited.
- B.3 shows how DeltaProduct can recognize any regular language in a finite number of layers using the Krohn-Rhodes decomposition.
- B.4 details the expressivity of a linear RNN with products of RWKV-7 state-transition matrices.
- B.5 demonstrates how DeltaNet can solve any dihedral group in 2 layers using a specific construction.
- B.6 discusses the tradeoff between expressivity and stability in linear RNNs.

Appendix C provides comprehensive details on our experiments and additional results.

We provide the code for our experiments at `https://github.com/automl/DeltaProduct`

**Notation.** Mathematical objects are typically styled as follows. Matrices: uppercase letters (e.g., $\boldsymbol{A}, \boldsymbol{H}, \boldsymbol{M}$). Vectors: lowercase letters (e.g., $\boldsymbol{k}, \boldsymbol{v}, \boldsymbol{x}$). Standard sets: $\mathbb{R}$ for reals, $\mathbb{C}$ for complexes, $\mathbb{N}$ for naturals. Common symbols and operations are denoted as follows. $\boldsymbol{I}$: Identity matrix. $\top$: Transpose operator (e.g., $\boldsymbol{k}^\top$). $\|\boldsymbol{v}\|$ or $\|\boldsymbol{v}\|_2$: Euclidean norm of a vector $\boldsymbol{v}$. $\|\boldsymbol{A}\|$: the operator norm for a matrix $\boldsymbol{A}$. $|\cdot|$: Absolute value for real scalars, or modulus for complex numbers. $\odot$: Element-wise (Hadamard) product. $\sigma(\boldsymbol{A}), \rho(\boldsymbol{A})$: Spectrum (set of eigenvalues) and spectral radius of matrix $\boldsymbol{A}$. $\mathrm{tr}(\boldsymbol{A}), \det(\boldsymbol{A})$: Trace and determinant of matrix $\boldsymbol{A}$. $\delta_{ij}$: Kronecker delta (1 if $i = j$, 0 otherwise). $\boldsymbol{e}_i \in \{0, 1\}^n$ is the $i$-th element of the canonical basis of $\mathbb{R}^n$.

## A  Spectral Properties and Simplifications of Householder Product Matrices

The following proposition characterizes conditions under which the product structure simplifies and the spectrum is real, contrasting with the general case which allows for complex spectra. It provides illustrative special cases for the more general Proposition 1 in Grazzi et al. [16].

**Proposition 1.** *Let $\boldsymbol{A} \in \mathbb{R}^{n \times n}$ be a matrix defined as the product of $n_h \geq 1$ generalized Householder transformations: $\boldsymbol{A} = \prod_{j=1}^{n_h} \boldsymbol{H}_j$ where each $\boldsymbol{H}_j = \boldsymbol{I} - \beta_j \boldsymbol{k}_j \boldsymbol{k}_j^\top$, with $\boldsymbol{k}_j \in \mathbb{R}^n$ being a unit vector ($\|\boldsymbol{k}_j\|_2 = 1$) and $\beta_j \in [0, 2]$. Let $\sigma(\boldsymbol{A}) \subset \mathbb{C}$ denote the spectrum (set of eigenvalues) of $\boldsymbol{A}$. Then, all eigenvalues $\lambda \in \sigma(\boldsymbol{A})$ satisfy $|\lambda| \leq 1$ and the following hold.*

1. *(**Identical Direction Vector**) Let $\boldsymbol{k} \in \mathbb{R}^n$ be nonzero and $\boldsymbol{k}_j = \boldsymbol{k} / \|\boldsymbol{k}\|$ for all $j = 1..m$. Then $\prod_{j=1}^m (\boldsymbol{I} - \beta_j \boldsymbol{k}\boldsymbol{k}^\top) = \boldsymbol{I} - \beta^* \boldsymbol{k}\boldsymbol{k}^\top$ for some real scalar $\beta^*$ depending on $\{\beta\}_{j=1}^m$. The product collapses to a single effective transformation of the same form. Consequently, if $\boldsymbol{A}$ is formed using only a single direction vector $\boldsymbol{k}_1$, it is symmetric and its spectrum is real.*

2. *(**Orthogonal Vectors**) If the direction vectors $\{\boldsymbol{k}_j\}_{j=1}^{n_h}$ form an orthonormal set (i.e., $\boldsymbol{k}_j^\top \boldsymbol{k}_l = \delta_{jl}$; this requires $n_h \leq n$), then the factors $\boldsymbol{H}_j$ commute, and the product simplifies to $\boldsymbol{A} = \boldsymbol{I} - \sum_{j=1}^{n_h} \beta_j \boldsymbol{k}_j \boldsymbol{k}_j^\top$. This matrix $\boldsymbol{A}$ is symmetric, and its spectrum is purely real: $\sigma(\boldsymbol{A}) = \{1 - \beta_1, \ldots, 1 - \beta_{n_h}\} \cup \{1 \text{ (multiplicity } n - n_h)\}$. When $\beta_j = 2$ for all $j \in \{1, ..., n_h\}$ then $\boldsymbol{A}$ is known as a block reflector [68].*

3. *(**Complex Spectrum via Non-orthogonal Directions**) For $n_h = 2$, $\boldsymbol{A}$ has complex eigenvalues if two consecutive direction vectors, e.g. $\boldsymbol{k}_1, \boldsymbol{k}_2$ satisfy $0 < |\boldsymbol{k}_1^\top \boldsymbol{k}_2| < 1$ and their coefficients $\beta_1, \beta_2$ exceed a threshold $\beta^*(\theta) < 2$ dependent on the angle $\theta$ between them. Conversely, if $0 \leq \beta_1 \leq 1$ or $0 \leq \beta_2 \leq 1$, these eigenvalues from the 2D span are guaranteed to be real.*

*Proof.* **Identical Direction Vector** If $m = 1$, then the statement is trivially satisfied with $\beta^* = \beta_1$. Suppose the statement is true for $m \geq 1$, i.e., $\prod_{j=1}^m (\boldsymbol{I} - \beta_j \boldsymbol{k}\boldsymbol{k}^\top) = \boldsymbol{I} - \beta^{(m)} \boldsymbol{k}\boldsymbol{k}^\top$. Multiplying by

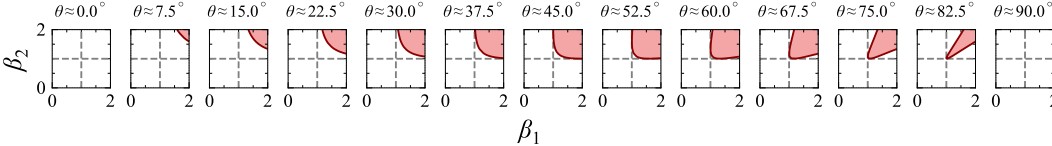

Figure 11: Visualization of complex eigenvalues of $(\boldsymbol{I}-\beta_1\boldsymbol{k}_1\boldsymbol{k}_1^\top)(\boldsymbol{I}-\beta_2\boldsymbol{k}_2\boldsymbol{k}_2^\top)$ with $\cos\theta = \boldsymbol{k}_1^\top\boldsymbol{k}_2$. Complex region in red, real in white.

$(\boldsymbol{I}-\beta_{m+1}\,\boldsymbol{k}\boldsymbol{k}^\top)$ produces $(\boldsymbol{I}-\beta^{(m)}\,\boldsymbol{k}\boldsymbol{k}^\top)(\boldsymbol{I}-\beta_{m+1}\,\boldsymbol{k}\boldsymbol{k}^\top) = \boldsymbol{I}-\left[\beta^{(m)}+\beta_{m+1}-\beta^{(m)}\beta_{m+1}\right]\boldsymbol{k}\boldsymbol{k}^\top$. Hence, by induction, the product of any number of such factors remains of the form $\boldsymbol{I} - \beta^*\boldsymbol{k}\boldsymbol{k}^\top$. Since the resulting matrix $\boldsymbol{A} = \boldsymbol{I} - \beta^*\boldsymbol{k}\boldsymbol{k}^\top$ (where $\boldsymbol{k} = \boldsymbol{k}_1$) is symmetric, its eigenvalues are real.

**Orthogonal Vectors** Assume $\{\boldsymbol{k}_j\}_{j=1}^{n_h}$ is an orthonormal set ($\boldsymbol{k}_j^\top\boldsymbol{k}_l = \delta_{jl}$, $n_h \leq n$). Let $\boldsymbol{P}_j = \boldsymbol{k}_j\boldsymbol{k}_j^\top$. Then $\boldsymbol{P}_j\boldsymbol{P}_l = \delta_{jl}\boldsymbol{P}_j$. The factors $\boldsymbol{H}_j = \boldsymbol{I} - \beta_j\boldsymbol{P}_j$ commute because $\boldsymbol{P}_j\boldsymbol{P}_l = \boldsymbol{0}$ for $j \neq l$. The product simplifies via induction to $\boldsymbol{A} = \boldsymbol{I} - \sum_{j=1}^{n_h}\beta_j\boldsymbol{P}_j$. This matrix is symmetric. Its eigenvalues are $(1-\beta_l)$ for $l = 1,\ldots,n_h$ (eigenvector $\boldsymbol{k}_l$) and 1 with multiplicity $n - n_h$ (subspace orthogonal to all $\boldsymbol{k}_j$). The spectrum is real.

**Complex Spectrum via Non-orthogonal Directions and Real Subcase** Let $\boldsymbol{k}_1, \boldsymbol{k}_2$ span a 2D subspace $S \subset \mathbb{R}^n$ with $\cos\theta = \boldsymbol{k}_1^\top\boldsymbol{k}_2$ such that $0 < |\cos\theta| < 1$. The product $\boldsymbol{A} = \boldsymbol{H}_2\boldsymbol{H}_1$ acts as the identity on $S^\perp$ (preserving $n - 2$ eigenvalues at 1) and non-trivially on $S$. The restriction $\boldsymbol{A}_S$ of $\boldsymbol{A}$ to $S$ has trace $\operatorname{tr}(\boldsymbol{A}_S) = 2 - \beta_1 - \beta_2 + \beta_1\beta_2\cos^2\theta$ and determinant $\det(\boldsymbol{A}_S) = (1-\beta_1)(1-\beta_2)$. The discriminant of its characteristic equation is $D = [\operatorname{tr}(\boldsymbol{A}_S)]^2 - 4\det(\boldsymbol{A}_S)$. Complex eigenvalues arise if $D < 0$.

To find the explicit bounds, we expand the inequality $D < 0$:

$$(2 - \beta_1 - \beta_2 + \beta_1\beta_2\cos^2\theta)^2 - 4(1 - \beta_1)(1 - \beta_2) < 0$$

Rearranging this expression as a quadratic in $x = \cos^2\theta$ yields:

$$(\beta_1\beta_2)^2 x^2 + 2(2 - \beta_1 - \beta_2)\beta_1\beta_2 x + (\beta_1 - \beta_2)^2 < 0$$

This inequality holds if and only if $x = \cos^2\theta$ lies strictly between the two roots of the corresponding equation. Solving for the roots gives the explicit bounds for complex eigenvalues:

$$\frac{(\sqrt{\beta_1 - 1} - \sqrt{\beta_2 - 1})^2}{\beta_1\beta_2} < \cos^2\theta < \frac{(\sqrt{\beta_1 - 1} + \sqrt{\beta_2 - 1})^2}{\beta_1\beta_2}$$

This inequality is only satisfiable when $\beta_1, \beta_2 \in (1, 2]$. For the special case of two standard reflections where $\beta_1 = \beta_2 = 2$, the condition simplifies to $0 < \cos^2\theta < 1$, confirming that the product of any two distinct reflections is a rotation.

Conversely, we show that if at least one coefficient $\beta_i \in [0, 1]$, the eigenvalues are real as $D \geq 0$. We analyze this by cases:

- **Case 1: One coefficient is in $[0, 1]$, the other is in $(1, 2]$.** Without loss of generality, let $\beta_1 \in [0, 1]$ and $\beta_2 \in (1, 2]$. This implies $(1 - \beta_1) \geq 0$ and $(1 - \beta_2) < 0$, so their product $\det(\boldsymbol{A}_S) \leq 0$. The term $-4\det(\boldsymbol{A}_S)$ is therefore non-negative. Since $[\operatorname{tr}(\boldsymbol{A}_S)]^2 \geq 0$, their sum $D$ must be non-negative.

- **Case 2: Both coefficients are in $[0, 1]$.** By the AM-GM inequality on the non-negative terms $(1 - \beta_1)$ and $(1 - \beta_2)$, we have $(1 - \beta_1) + (1 - \beta_2) \geq 2\sqrt{(1 - \beta_1)(1 - \beta_2)} = 2\sqrt{\det(\boldsymbol{A}_S)}$. Since $\operatorname{tr}(\boldsymbol{A}_S)$ includes an additional non-negative term $\beta_1\beta_2\cos^2\theta$, it also holds that $\operatorname{tr}(\boldsymbol{A}_S) \geq 2\sqrt{\det(\boldsymbol{A}_S)}$. Squaring both sides gives $[\operatorname{tr}(\boldsymbol{A}_S)]^2 \geq 4\det(\boldsymbol{A}_S)$, ensuring $D \geq 0$.

This analysis confirms that complex eigenvalues, which enable rotations, can only arise if and only if both $\beta_1 > 1$ and $\beta_2 > 1$. When at least one $\beta_i \leq 1$, real eigenvalues restrict the transformations in that subspace to scaling or reflection. This clear distinction in behavior, dictated by the $\beta$ values and $\theta$, is illustrated in Figure 11. $\qquad\square$

# B Expressivity of DeltaProduct

In this section, we characterize the expressivity of (Gated) DeltaProduct in solving group word problems and recognizing regular languages, in support of Section 4.1. The results hold in finite precision (since our constructions require a finite number of values), and take inspiration from Peng et al. [13, Appendix D] and [16]. We begin by stating and discussing our key assumptions in Appendix B.1. We then present our main results for group word problems in Appendix B.2, followed by our findings for regular languages in Appendix B.3. In Appendix B.5, we examine a result specific to dihedral groups. Finally, we explore the fundamental tradeoff between expressivity and stability of the recurrence in Appendix B.6.

## B.1 Assumptions

We consider a (Gated) DeltaProduct model where each layer is structured as

$$\boldsymbol{H}_i = \boldsymbol{A}(\boldsymbol{x}_i)\boldsymbol{H}_{i-1} + \boldsymbol{B}(\boldsymbol{x}_i), \qquad \hat{\boldsymbol{y}}_i = \mathrm{dec}(\boldsymbol{H}_i, \boldsymbol{x}_i) \qquad \text{where } i \in 1, \dots, t$$

$$\boldsymbol{A}(\boldsymbol{x}_i) = g_i \prod_{j=1}^{n_h} \Big( \boldsymbol{I} - \beta_{i,j}\, \boldsymbol{k}_{i,j}\boldsymbol{k}_{i,j}^\top \Big), \quad \boldsymbol{B}(\boldsymbol{x}_i) = \sum_{j=1}^{n_h} \Big( \prod_{k=j+1}^{n_h} \big( \boldsymbol{I} - \beta_{i,k}\, \boldsymbol{k}_{i,k}\boldsymbol{k}_{i,k}^\top \big) \Big) \beta_{i,j}\, \boldsymbol{k}_{i,j}\boldsymbol{v}_{i,j}^\top,$$

where $g_i$ is only present in the gated variant and there is only one head per layer. If $H$ heads are considered, head $j$ will run the recurrence $\boldsymbol{H}_i^j = \boldsymbol{A}^j(\boldsymbol{x}_i)\boldsymbol{H}_{i-1} + \boldsymbol{B}^j(\boldsymbol{x}_i)$, (with different learnable parameters from all other heads) and all the states will be passed to the decoder to get the output as $\hat{\boldsymbol{y}}_i = \mathrm{dec}((\boldsymbol{H}_i^1, \dots, \boldsymbol{H}_i^H), \boldsymbol{x}_i)$.

**Each layer receives the outputs of all previous layers.** We assume that the output of all layers is passed as input to all following layers: this can be achieved by using the residual connections (which would be inside dec) and by placing the output of different layers onto separate subspaces.

**Task-dependent initial state.** We assume that the initial state $\boldsymbol{H}_0$ can be set appropriately depending on the task and is of rank at most $n_h$.

**Arbitrary decoder and state transition functions.** We also assume that for every $i, j$, $g_i \in [0, 1]$, $\beta_{i,j} \in [0, 2]$, $\boldsymbol{k}_{i,j} \in \mathbb{R}^d$, $\|\boldsymbol{k}_{i,j}\| = 1$ and $\boldsymbol{v}_{i,j} \in \mathbb{R}^d$ are arbitrary continuous functions of $\boldsymbol{x}_i$. The function dec can also be arbitrary (continuous). Since in all our setups the possible values of $\boldsymbol{x}_i$ and $\boldsymbol{H}_i$ are finite, this implies that the functions dec, $\boldsymbol{A}, \boldsymbol{B}$ can model an arbitrary function of their discrete domain of interest, with the only restriction coming from the structural assumption of the output spaces of $\boldsymbol{A}$ (product of $n_h$ Householders) and $\boldsymbol{B}$ (rank $n_h$). This assumption can always be fulfilled in practice if dec is a sufficiently wide MLP (see the next section for practical concerns) and, in the case of $\boldsymbol{A}, \boldsymbol{B}$, which generally do not contain MLPs, by setting the dimension of $\boldsymbol{x}_i$ sufficiently large, which can be achieved by adjusting the embedding layer or the dimensionality of the output of dec. Note that the width of the MLP will grow with the complexity of the function to be approximated, which in our case depends on the complexity of the problem.

### B.1.1 Practical Considerations

**Beginning of sequence token.** Alternatively, the assumption on $\boldsymbol{H}_0$ (task-dependent with rank at most $n_h$) can be replaced by using a beginning of sequence token $x_1 = \$$ and setting $\boldsymbol{H}_0 = 0$, as done in practice, so that $\boldsymbol{H}_1 = \boldsymbol{B}(\$)$ is a learnable matrix of rank at most $n_h$ which acts as the $\boldsymbol{H}_0$ in our constructions.

**Decoder implementation.** In our implementation, dec is the same as in Gated DeltaNet:

$$\mathrm{dec}((\boldsymbol{H}_t^1, \dots \boldsymbol{H}_t^H), \boldsymbol{x}_t) = \mathrm{MLP}(\mathrm{RMSnorm}(\boldsymbol{x}_t + \boldsymbol{o}_t)),$$

$$\boldsymbol{o}_t = \sum_{j=1}^{H} \boldsymbol{W}_o^j \mathrm{RMSnorm}((\boldsymbol{H}_t^j)^\top \boldsymbol{q}_t^j), \quad \boldsymbol{q}_t^j = \psi(\boldsymbol{W}_q^j \boldsymbol{x}_t)/\|\psi(\boldsymbol{W}_q^j \boldsymbol{x}_t)\|$$

where $\boldsymbol{W}_o^j, \boldsymbol{W}_q^j$ are two learned matrices, $\psi = \mathrm{SiLU}$, $\mathrm{RMSnorm}(\boldsymbol{x}) = \boldsymbol{a} \odot \boldsymbol{x}/\sqrt{\epsilon + d^{-1}\sum_{i=1}^{d} x_i^2}$ corresponds (when $\epsilon = 0$) to a projection onto an axis aligned ellipse where $\boldsymbol{a} \in \mathbb{R}^d$ is learnable and determines the lengths of the axis and $\epsilon > 0$ is a small value set to avoid numerical errors.

Meanwhile, MLP is a two layer MLP. This structure can be limiting. For instance, when $\boldsymbol{H}_t^j \in \mathbb{R}^d$, then $b_t = (\boldsymbol{H}_t^j)^\top \boldsymbol{q}_t \in \mathbb{R}$ and if $b_t >> \epsilon$, then $|\text{RMSNorm}(b_t)| \approx 1$, which means that after RMSNorm we are left with effectively only 2 possible values, while our constructions might require more: as many as the number of possible states. Indeed, for all our constructions to work in practice, we would need a sufficiently large $\epsilon$, so that the output of RMSnorm can retain some magnitude information. Since in our constructions the number of states is finite, with $\epsilon > 0$ and an appropriate value for $\boldsymbol{q}_t^j$ we are guaranteed that the map $\boldsymbol{H}_t^j \mapsto \text{RMSnorm}((\boldsymbol{H}_t^j)^\top \boldsymbol{q}_t^j)$, and consequently (by appropriately setting $\boldsymbol{W}_o^j$) the map from states and inputs $\boldsymbol{x}_t$, to the input of the MLP, is injective, which we show in the next lemma. Hence, thanks to the MLP, the decoder can approximate any continuous function of $(\boldsymbol{H}_t, \boldsymbol{x}_t)$ even after the bottlenecks caused by the scalar product with $\boldsymbol{q}_t^j$ and the RMSnorm.

**Lemma 1.** *Let $\mathcal{S} \subset \mathbb{R}$ be a finite set of values. Define $\delta_{\min} = \min_{x,y \in \mathcal{S}, x \neq y} |x - y|$ and $\delta_{\max} = \max_{x,y \in \mathcal{S}} |x - y|$. Let $\boldsymbol{b} = (b, b^2, \dots, b^d)^\top$ and set $\boldsymbol{q} = \boldsymbol{b}/\|\boldsymbol{b}\|$. If $b$ satisfies $b \geq \frac{\delta_{\max}}{\delta_{\min}} + 1$, then the mapping $f : \mathcal{S}^d \to \mathbb{R}$ given by $f(\boldsymbol{H}) = \text{RMSnorm}(\boldsymbol{H}^\top \boldsymbol{q})$ is injective.*

*Proof.* The mapping $f$ is a composition $f = g_3 \circ g_2 \circ g_1$, where the component functions are:

$$g_1(\boldsymbol{H}) = \sum_{i=1}^d h_i b^i, \qquad g_2(x) = \frac{x}{\|\boldsymbol{b}\|}, \qquad g_3(x) = \text{RMSnorm}(x),$$

where $\boldsymbol{H} = (h_1, \dots, h_d)^\top$. The overall mapping is injective if each component is injective.

**1. Injectivity of $g_1$:** Intuitively, $g_1$ encodes the vector $\boldsymbol{H}$ as a scalar as a number in base $b$, where each $h_i$ acts as a "digit" drawn from the finite set $\mathcal{S}$. By choosing $b$ sufficiently large relative to the spread of $\mathcal{S}$ (captured by $\delta_{\max}/\delta_{\min}$), we ensure that different vectors produce distinct scalars. To prove this formally, we show that for any non-zero difference vector $\boldsymbol{\Delta} = (\Delta_1, \dots, \Delta_d)^\top = \boldsymbol{H}_1 - \boldsymbol{H}_2$, the difference $g_1(\boldsymbol{H}_1) - g_1(\boldsymbol{H}_2) = \sum_{i=1}^d \Delta_i b^i$ is also non-zero. This is true since $b$ is by definition greater than Cauchy polynomial upper bound for the roots of the polynomial $P(b) = \sum_{i=1}^d \Delta_i b^i$. For an elementary proof, let $k = \max\{i \mid \Delta_i \neq 0\}$. The magnitude of the highest-order term is lower-bounded by:

$$|\Delta_k b^k| = |\Delta_k| b^k \geq \delta_{\min} b^k$$

The magnitude of the sum of lower-order terms is upper-bounded as

$$\left| \sum_{i=1}^{k-1} \Delta_i b^i \right| \leq \sum_{i=1}^{k-1} |\Delta_i| b^i \leq \sum_{i=1}^{k-1} \delta_{\max} b^i = \delta_{\max} \left( \frac{b^k - b}{b - 1} \right) \leq \delta_{\min}(b^k - b),$$

where we used the triangle inequality and the assumption on $b$, which implies $\delta_{\max} \leq \delta_{\min}(b - 1)$. Comparing the bounds, we see that

$$|\Delta_k b^k| \geq \delta_{\min} b^k > \delta_{\min}(b^k - b) \geq \left| \sum_{i=1}^{k-1} \Delta_i b^i \right|$$

Since the magnitude of the highest-order term is strictly greater than that of the sum of all other terms, their sum cannot be zero. Thus, $g_1$ is injective.

**2. Injectivity of $g_2$ and $g_3$:** The function $g_2$ is a linear scaling by the non-zero constant $1/\|\boldsymbol{b}\|$ and is therefore injective. For $g_3(x) = \text{RMSnorm}(x)$, its derivative is strictly positive for $\epsilon > 0$, meaning $g_3$ is strictly monotonic and also injective.

Since $g_1$, $g_2$, and $g_3$ are all injective, their composition $f$ is also injective. $\qquad \square$

When the state is one-hot, i.e. $\boldsymbol{H}_t^j = \boldsymbol{e}_i \in \{0, 1\}^d$, with $1 \leq i \leq d$ (i-th element of the canonical basis), an alternative to the above construction is to replicate the recurrence onto $d$ heads, where the $j$-th head has $\boldsymbol{W}_o^j = \boldsymbol{q}_t^j = \boldsymbol{e}_j$, so that, assuming that in the RMSnorm $\boldsymbol{a} = (\sqrt{d}, \dots, \sqrt{d})^\top$ and $\epsilon = 0$, we get $\boldsymbol{o}_t = \boldsymbol{H}_t = \boldsymbol{e}_i$. This is the strategy used in Peng et al. [13, Appendix D]. However, for some problems using the one-hot encoding states $\boldsymbol{e}_1, \dots \boldsymbol{e}_n$ is not very efficient. For instance, to solve the $S_n$ word problem one would need $n!$-dimensional one-hot vectors as states, while in our Theorem 1 we use $n$-dimensional vectors. Moreover, learning multiple identical heads is redundant and indeed we observe that in our synthetic experiments, the model is learning to use only one head to solve the tasks (see Section 5.2).

## B.2 Group Word Problems

The next theorem establishes that DeltaProduct can solve the word problem for the symmetric group $S_n$, which implies that it can also solve any group word problem, since for every group $G$ there exists $n$ such that $G$ is isomorphic to a subgroup of $S_n$.

**Theorem 3** (Restatement of Theorem 1). *For any $n \in \mathbb{N}$ there exists a DeltaProduct model with one of the following configurations that can solve the word problem of the symmetric group $S_n$: (i) one layer with $n_h = n-1$ [16, Theorem 3] (ii) 3 layers with $n_h > 1$ (iii) 4 layers with $n_h = 1$. The construction for (ii) and (iii) requires that the MLP at the second last layer computes a lookup-table of size $2m \times (n!)^{2m}$, function of the last $2m$ input tokens and the position modulo $2m$ with $m = \lceil (n-1)/n_h \rceil$.*

*Proof.* One way to solve the group word problem for the symmetric group $S_n$ is to map each element of the group $g \in S_n$ to the corresponding permutation matrix $\boldsymbol{P}_g \in \{0,1\}^n$ and then for each input sequence $x_1, \ldots, x_t$ with $x_i \in S_n$ compute each element of the output sequence $y_1, \ldots, y_t$ as

$$y_i = x_i \cdot x_{i-1} \cdots x_1 = \phi(\boldsymbol{P}_{x_i} \cdots \boldsymbol{P}_{x_1} \boldsymbol{u}_0), \quad \boldsymbol{u}_0 = (1, \ldots, n)^\top,$$

where $\phi$ is a surjective map from vectors in $\{1, \ldots, n\}^n$ to the $n!$ elements of $S_n$, which we consider integers for simplicity, i.e. $x_i, y_i \in \{1, \ldots, n!\}$.

**(i).** Since a permutation of $n$ elements is a series of at most $n-1$ swaps of 2 elements, if $n_h = n-1$, then we can solve the problem with a 1-layer DeltaProduct by setting $\boldsymbol{H}_0 = \boldsymbol{u}_0$, $\mathrm{dec}(\boldsymbol{H}_i, x_i) = \phi(\boldsymbol{H}_i)$, $\boldsymbol{B}(x_i) = 0$ ($\boldsymbol{v}_{i,j} = 0$), $\boldsymbol{A}(x_i) = \boldsymbol{P}_{x_i}$. The latter is achieved by setting for the $j$-th element in the product $\prod_{j=1}^{n_h} (I - \beta_{i,j} \boldsymbol{k}_{i,j} \boldsymbol{k}_{i,j}^\top)$, either $\beta_{i,j} = 2$ and $\boldsymbol{k}_{i,j} = (\boldsymbol{e}_k - \boldsymbol{e}_p)/\sqrt{2}$ with $\boldsymbol{e}_i$ being the $i$-th element of the canonical basis of $\mathbb{R}^n$ (swap element at index $k$ with the one at index $p$), or $\beta_{i,j} = 0$ (identity).

**(ii) and (iii).** If $n_h < n-1$, then the state transition matrix is not sufficiently expressive to represent all permutations of $n$ elements. However, we can use additional layers to overcome this issue as follows. We divide the input sequence into blocks of $m$ elements: we factorize the position $i \in \{1, \ldots t\}$ into $i = lm + \tilde{i}$ where $l \geq 0$ is the index of the previous block and $\tilde{i} \in \{1, \ldots, m\}$ is the position within the current block (index $l+1$). First, consider the case when $l \geq 1$. Let $\tilde{\boldsymbol{P}}_l = \boldsymbol{P}_{x_{(l-1)m+m}} \cdots \boldsymbol{P}_{x_{(l-1)m+1}}$ be the product of the permutations of the previous block. Since $\tilde{\boldsymbol{P}}_l$ is a permutation matrix of $n$ elements, we can factor it into $\tilde{\boldsymbol{P}}_l = \boldsymbol{G}_{l,m} \cdots \boldsymbol{G}_{l,1}$ where we recall that $m = \lceil (n-1)/n_h \rceil$ and each of $\boldsymbol{G}_{l,1}, \ldots, \boldsymbol{G}_{l,m}$ is a product of $n_h$ generalized Householder matrices and thus can be a state-transition matrix of our model. We fix one factorization for each possible permutation matrix and we set $\tilde{\boldsymbol{P}}_0 = \boldsymbol{G}_{0,m} \cdots \boldsymbol{G}_{0,1}$, with $\boldsymbol{G}_{0,i} = I$ to handle the case when $l = 0$.

Now let $\boldsymbol{x}_i$ be the input of the last layer. if $\boldsymbol{x}_i$ contains enough information about previous tokens (as we specify later), we can set the recurrence and decoder of the last layer as

$$\boldsymbol{H}_i = \boldsymbol{G}_{l,\tilde{i}} \boldsymbol{H}_{i-1}, \quad \mathrm{dec}(\boldsymbol{H}_i, \boldsymbol{x}_i) = \phi\Big( \underbrace{\boldsymbol{P}_{x_i} \cdots \boldsymbol{P}_{x_{lm+1}}}_{\text{current block}} \underbrace{\boldsymbol{G}_{l,m} \cdots \boldsymbol{G}_{l,\tilde{i}+1}}_{\text{previous block}} \boldsymbol{H}_i \Big).$$

where $\boldsymbol{H}_0 = \boldsymbol{u}_0$, $\boldsymbol{B}(\boldsymbol{x}_i) = 0$, $\boldsymbol{A}(\boldsymbol{x}_i) = \boldsymbol{G}_{l,\tilde{i}}$, using the construction at point (i) since $\boldsymbol{G}_{l,\tilde{i}}$ is a product of at most $n_h$ Householders. Note that $\boldsymbol{H}_i$ contains the product of the input permutations only up to token $x_{(l-1)m}$ and a partial computation of previous block of permutations $\tilde{\boldsymbol{P}}_l$. Hence, the decoder completes the computation by applying two additional components: (1) the remaining transformations $\boldsymbol{G}_{l,\tilde{i}+1}$ through $\boldsymbol{G}_{l,m}$ needed to complete $\tilde{\boldsymbol{P}}_l$, and (2) the actual permutations from the current partial block $\boldsymbol{P}_{x_{lm+1}}$ through $\boldsymbol{P}_{x_i}$. The delay in the recurrence is necessary, since to compute even the first matrix of the factorization for a block of $m$ elements of the input sequence, all the elements in such a block need to be processed.

We can check that this ends up computing the correct output $y_i$ by substituting the expression for $\boldsymbol{H}_i$ and unrolling the recurrence as follows.

$$\mathrm{dec}(\boldsymbol{H}_i, \boldsymbol{x}_i) = \phi(\boldsymbol{P}_{x_{lm+\tilde{i}}} \cdots \boldsymbol{P}_{x_{lm+1}} \boldsymbol{G}_{l,m} \cdots \boldsymbol{G}_{l,1} \boldsymbol{G}_{l-1,m} \cdots \boldsymbol{G}_{l-1,1} \ldots \boldsymbol{G}_{0,m} \ldots \boldsymbol{G}_{0,1} \boldsymbol{H}_0).$$
$$= \phi(\boldsymbol{P}_{x_{lm+\tilde{i}}} \cdots \boldsymbol{P}_{x_{lm+1}} \tilde{\boldsymbol{P}}_l \tilde{\boldsymbol{P}}_{l-1} \ldots \tilde{\boldsymbol{P}}_0 \boldsymbol{u}_0).$$
$$= \phi(\boldsymbol{P}_{x_{lm+\tilde{i}}} \cdots \boldsymbol{P}_{x_{lm+1}} \boldsymbol{P}_{x_{(l-1)m+m}} \cdots \boldsymbol{P}_{x_{(l-1)m+1}} \ldots \boldsymbol{P}_{x_m} \cdots \boldsymbol{P}_{x_1} \tilde{\boldsymbol{P}}_0 \boldsymbol{u}_0).$$
$$= \phi(\boldsymbol{P}_{x_i} \cdots \boldsymbol{P}_{x_1} \boldsymbol{u}_0) = y_i,$$

Note that to compute $\boldsymbol{A}(\boldsymbol{x}_i) = \boldsymbol{G}_{l,\tilde{i}}$ and $\mathrm{dec}(\boldsymbol{H}_i, \boldsymbol{x}_i)$, $\boldsymbol{x}_i$ should contain $\tilde{i} = i \bmod m$ and the last $m + \tilde{i}$ (in general the last $2m$) tokens, corresponding to the current and previous blocks. Hence, the layers before the last one are dedicated to compute at each time-step $i$ a lookup table for the possible values of $(i \bmod 2m, x_i, \ldots, x_{i-2m+1})$ whose output will be included in the input of the last layer $\boldsymbol{x}_i$. The first layers (two layers if $n_h = 1$, one if $n_h > 1$) can provide $i \bmod 2m$ by using Lemma 2 with $d = 2m$. Finally, the second to last layer can output any function of the last $2m$ tokens and the position modulo $2m$ through Lemma 3 with $d = 2m$ and $a_t = x_t$, by using $i \bmod 2m$ from the first layer(s). $\qquad\square$

**Lemma 2.** *The following DeltaProduct configurations can count modulo $d \in \mathbb{N}$. (i) 2 layers each with one head and $n_h = 1$ [16, Theorem 6]. (ii) 1 layer with one head and $n_h \geq 2$.*

*Proof.* For (i), we can use the same construction as in [16, Theorem 6], where the first layer does counting modulo 2 and the second layer computes addition modulo $d$. In this case, since we just want to count modulo $d$ we can ignore input tokens and add 1 modulo $d$ at each time-step. For (ii), note that if $n_h > 2$, we can set, for any time-step $t$, $\boldsymbol{B}(\boldsymbol{x}_t) = 0$ and the state transition matrix $\boldsymbol{A}(\boldsymbol{x}_t)$ equal to a 2D rotation with an angle of $2\pi/d$ by appropriately setting two keys, say $\boldsymbol{k}_{t,1}, \boldsymbol{k}_{t,2}$, setting $\beta_{t,1}, \beta_{t,2} = 2$ (while for the other Householders we set $\beta_{i,j} = 0$). Then, we can count modulo $d$ by setting $\boldsymbol{H}_0$ in the span of $\boldsymbol{k}_{t,1}, \boldsymbol{k}_{1,2}$ and dec appropriately to map the $d$ values that $\boldsymbol{H}_t$ can take to the correspondent element in $\{1, \ldots, d\}$. $\qquad\square$

**Lemma 3.** *A DeltaProduct layer with $n_h = 1$, receiving in its input at time-step $t$ the tuple $(t \bmod d, a_t)$ $(t \geq 1)$ where $a_t \in D \subset \mathbb{R}$ with $D$ being a discrete set of values, can implement any function of $(t \bmod d, a_{t-d+1}, \ldots, a_t)$, where for simplicity we set $a_i = a \notin D$ for $i \in \{2 - d, \ldots, 0\}$.*

*Proof.* Let $\tilde{t} = t \bmod d + 1$ Set $\boldsymbol{H}_0 = 0 \in \mathbb{R}^d$ and the recurrence update as

$$\boldsymbol{H}_t = (I - \boldsymbol{e}_{\tilde{t}} \boldsymbol{e}_{\tilde{t}}^\top) \boldsymbol{H}_{t-1} + \boldsymbol{e}_{\tilde{t}} a_t,$$

where $\boldsymbol{e}_i$ is the $i$-th element of the canonical basis of $\mathbb{R}^d$. This can be implemented by setting $\beta_{t,1} = 1$, $\boldsymbol{k}_{t,1} = \boldsymbol{e}_{\tilde{t}}$, $\boldsymbol{v}_{t,1} = a_t$ and $\beta_{t,j}, \boldsymbol{k}_{t,j}, \boldsymbol{v}_{t,j} = 0$. With this choice, $\boldsymbol{H}_t$ contains $a_{t-d+1}, \ldots, a_t$. The result follows since dec can be an arbitrary continuous function of both $\boldsymbol{H}_t$ and $\boldsymbol{x}_t$ and the latter contains $t \bmod d$. $\qquad\square$

Finally, the next results concern finite subgroups of the orthogonal and special orthogonal groups.

**Theorem 4.** *Let $G$ be a group isomorphic either to a subgroup of $\mathrm{O}(n)$, or to a subgroup of $\mathrm{SO}(n+1)$ if $n$ is even, then if $n_h = n$, there exists a DeltaProduct model that solves the group word problem for $G$.*

*Proof.* From the assumption we can map each element $g \in G$ to an orthogonal matrix $\boldsymbol{G}_g$. For the word problem for $G$, each element of the input sequence belongs to $G$: $x_i \in G$ for every $i$.

If $\boldsymbol{G}_g \in \mathrm{O}(n)$, then, since $n_h = n$ and every orthogonal $n \times n$ matrix can be written as the product of at most $n$ Householder matrices, we can set $\boldsymbol{H}_0 = I \in \mathbb{R}^{n \times n}$ and $\boldsymbol{A}(\boldsymbol{x}_i) = \boldsymbol{G}_{x_i}$, $\boldsymbol{B}(\boldsymbol{x}_i) = 0$ and $\mathrm{dec}(\boldsymbol{H}_t, x_i) = \phi(\boldsymbol{H}_i)$ with $\phi : \mathrm{O}(n) \to G$ bijective (which exists due to the isomorphism). The Householder product structure enables $\boldsymbol{A}(\boldsymbol{x}_i)$ to represent general orthogonal matrices $\boldsymbol{G}_{x_i}$, including rotations.

If instead $\boldsymbol{G}_g \in \mathrm{SO}(n+1)$, since $n$ is even in this case then we can still write $\boldsymbol{G}_g$ as a product of an even number (at most $n$ since $n + 1$ is odd) of Householder matrices of dimension $n + 1 \times n + 1$. This is because the determinant of $\boldsymbol{G}_g$ is $+1$, which is only possible if it is a product of an even number of Householder matrices, each having determinant $-1$. Thus, we can set $\boldsymbol{A}(\boldsymbol{x}_i) = \boldsymbol{G}_{x_i} \in \mathbb{R}^{n+1 \times n+1}$, $\boldsymbol{B}(\boldsymbol{x}_i) = 0$. Now if we let $\bar{\boldsymbol{G}} = \boldsymbol{G}_{x_i} \boldsymbol{G}_{x_{i-1}} \cdots \boldsymbol{G}_{x_1}$ and set $\boldsymbol{H}_0 = \mathrm{diag}(1, \ldots, 1, 0) \in \mathbb{R}^{(n+1) \times (n+1)}$ (we are only allowed a rank n matrix), then $\boldsymbol{H}_i = \bar{\boldsymbol{G}} \boldsymbol{H}_0$ will have all the first $n$ columns equal to $\bar{\boldsymbol{G}}$ and the last set to zero. However, the last column can be found as a function of the others since it must be the unique unit vector orthogonal to all other columns of $\bar{\boldsymbol{G}}$ and for which $\det(\bar{\boldsymbol{G}}) = +1$. Therefore, there exists a bijective functon from states to elements of the group, which can be implemented in dec. $\qquad\square$

### B.3 Regular Languages

This section details how Gated DeltaProduct networks can recognize any regular language in a finite number of layers. The core idea is to show that Gated DeltaProduct can simulate any Finite State Automaton (FSA), since FSAs are the computational models that define regular languages. The proof proceeds in two main steps: First, we leverage the Krohn-Rhodes theorem, a fundamental result in automata theory, which states that any FSA can be decomposed into a cascade of simpler FSAs known as permutation-reset automata. These simpler automata only perform two types of operations: permuting their states or resetting all states to a single state. Second, we demonstrate in Lemma 4 that Gated DeltaProduct is well-equipped to simulate these permutation-reset automata. The "DeltaProduct" mechanism, using products of Householder transformations, naturally handles permutations, while the "Gated" aspect allows for the reset operations by nullifying the previous state's influence and setting a new one. By simulating these building blocks and cascading them, Gated DeltaProduct can thus simulate any FSA and, consequently, recognize any regular language.

**Definition 1** (Finite state automaton (FSA)). *A finite state automaton (FSA) is a tuple $(\Sigma, Q, q_0, \delta, F)$, where $\Sigma$ is a finite set called alphabet, $Q$ is the finite set of states, $q_0 \in Q$ is the initial state, for every $w \in \Sigma$, $\delta_w : Q \to Q$ is a state transition function and $F \subset Q$ is the set of accepting states.*

**Definition 2** (Permutation-reset automaton). *An FSA is permutation-reset if for every $w \in \Sigma$, $\delta_w$ is either bijective or constant.*

**Definition 3** (Regular language). *A regular language is a set of sequences $L$ such that there exists an FSA that accepts it, i.e. such that $L \subset \Sigma^*$, where $\Sigma^*$ is the set of sequences with elements in $\Sigma$, and that for every word $\boldsymbol{w} = w_1 w_2 \ldots w_t \in \Sigma^*$*

$$\delta_{\boldsymbol{w}}(q_0) := \delta_{w_t} \circ \delta_{w_{t-1}} \circ \cdots \circ \delta_{w_1}(q_0) \in F \iff \boldsymbol{w} \in L. \tag{4}$$

Notably, the computation of any FSA can be also done using only matrix and vector multiplications. Indeed, if we let $Q = \{1, \ldots, n\}$ (for simplicity), then we can map each state $q$ to the one hot vector $\boldsymbol{e}_q$ (element of the canonical basis of $\mathbb{R}^n$) and each transition $\delta_w$ to the matrix $\boldsymbol{M}_w \in \{0,1\}^{n \times n}$ with element at row $q$ and column $q'$ being 1 if and only if $\delta(q') = q$. This way, by setting $\boldsymbol{r} \in \{0,1\}^n$ such that $r_q = 1$ if $q \in F$ and $r_q = 0$ otherwise, we have that for every word $\boldsymbol{w} = w_1 w_2 \ldots w_n \in \Sigma^*$

$$\boldsymbol{r}^\top \boldsymbol{M}_{w_t} \boldsymbol{M}_{w_{t-1}} \cdots \boldsymbol{M}_{w_1} \boldsymbol{e}_{q_0} = 1 \iff \boldsymbol{w} \in L. \tag{5}$$

We observe that if $\delta_w$ is bijective and changes $k$ states, then the corresponding $\boldsymbol{M}_w$ is a permutation matrix that can be written as a product of $k-1$ Householder matrices, each corresponding to a swap of two elements. Moreover, if $\delta_w$ is a reset (constant), i.e. if $\delta_w(q) = \bar{q}$ for every $q \in Q$, then $\boldsymbol{M}_w \boldsymbol{e}_q = \boldsymbol{e}_{\bar{q}}$. As we will see, constant transitions can be modeled by setting the gate to zero. We are now ready to state our main result.

**Theorem 5** (Restatement of Theorem 2). *For any regular language $L$ and any $n_h \in \mathbb{N}$, there exists a Gated DeltaProduct model with a finite number of layers that recognizes the language, i.e., for every word $\boldsymbol{w} \in \Sigma^*$ outputs 1 if $\boldsymbol{w} \in L$ and 0 otherwise.*

*Proof.* Using the landmark theorem by Krohn and Rhodes [50] we can decompose the FSA corresponding to the regular language $L$ into a cascade of permutation-reset FSA. We can use a group of at most 4 consecutive layers to represent each automaton in the cascade via Lemma 4. Then, we can combine the different FSA in the cascade in a feedforward manner using the same construction as the one in the proof of [16, Theorem 3], where the input of each FSA is the output concatenated with the input of the previous FSA in the cascade. $\qquad \square$

**Lemma 4.** *For any permutation-reset FSA with $|Q| = n$ and $|\Sigma| = s$, where each bijective state-transition function $\delta_w$ changes at most $k$ states, there exists a Gated DeltaProduct model with the following configuration that can implement it, i.e., for any word $\boldsymbol{w} = w_1, \ldots, w_t \in \Sigma$ in input, it can output the corresponding sequence of states $q_1, \ldots, q_t$ of the FSA. (i) one layer with $n_h = k-1$. (ii) 3 layers with $n_h > 1$. (iii) 4 layers with $n_h = 1$. The construction for (ii) and (iii) requires that the MLP at the second last layer computes a lookup-table of size $2m \times s^{2m}$, function of the last $2m$ input tokens and the position modulo $2m$ with $m = \lceil (k-1)/n_h \rceil$.*

*Proof.* We use the matrix vector multiply construction to implement the FSA. For every time-step $i$ we set $x_i = w_i$ as the input to the model. The proof follows a path similar to the one of Theorem 3

(where more details are provided), where in addition to modeling permutations, the state-transition matrix uses the gate to model constant transitions.

**(i).** Set $\boldsymbol{H}_0 = \boldsymbol{e}_{q_0}$. When $\delta_{w_i}$ is bijective, by assumption it changes at most $k$ states. Thus, by setting $n_h = k - 1$ we can represent the corresponding $\boldsymbol{M}_{w_i}$ matrix using $\boldsymbol{A}(w_i)$ with gate $g_i = 1$ (product of $k - 1$ generalized Householder matrices), and thus we set $\boldsymbol{B}(w_i) = 0$. If instead $\delta_{w_i}$ is constant, i.e., $\delta_i(q) = \bar{q}$ and $\boldsymbol{M}_w \boldsymbol{e}_q = \boldsymbol{e}_{\bar{q}}$ for every $q \in Q$, then we can set the gate $g_i = 0$ so that $\boldsymbol{A}(w_i) = 0$ and $\boldsymbol{B}(w_i) = \boldsymbol{k}_i = \boldsymbol{e}_{\bar{q}}$. Finally, we set $\mathrm{dec}(\boldsymbol{H}_i, x_i) = \boldsymbol{H}_i^\top (1, \ldots, n)^\top$ to retrieve the correct state at step $i$ (for simplicity $Q = \{1, \ldots, n\}$).

**(ii) and (iii).** If $n_h < k - 1$, then the state transition matrix is not sufficiently expressive to represent all permutations of $k$ elements. However, we can use additional layers to overcome this issue.

We factorize the position $i \in \{1, \ldots t\}$ into $i = lm + \tilde{i}$ for integers $l \geq 0$ and $\tilde{i} \in \{1, \ldots, m\}$. First, consider the case when $l \geq 1$. The product $\tilde{\boldsymbol{M}}_l = \boldsymbol{M}_{w_{(l-1)m+1}} \ldots \boldsymbol{M}_{w_{(l-1)m+m}}$ is either a permutation matrix of $k$ elements or, if for some $i$ $\delta_{w_i}$ is constant, then there exists $\bar{q}$ such that $\tilde{\boldsymbol{M}}_l \boldsymbol{e}_q = \bar{q}$ for every $q \in Q$. Therefore, we factor $\tilde{\boldsymbol{M}}_l$ into $\tilde{\boldsymbol{M}}_l = \boldsymbol{G}_{l,m} \cdots \boldsymbol{G}_{l,1}$ where each of $\boldsymbol{G}_{l,1}, \ldots, \boldsymbol{G}_{l,m}$ is either a product of $n_h$ generalized Householder matrices or $\boldsymbol{G}_{l,i} \boldsymbol{e}_q = \boldsymbol{e}_{\bar{q}}$ for every $q \in Q$, which can be modeled setting the gate to zero as in point (i). We fix one factorization for each possible permutation matrix. In the last layer and with enough information in its input $\boldsymbol{x}_i$ about past tokens, we can thus set $\boldsymbol{H}_0 = \boldsymbol{e}_0$ and

$$\boldsymbol{H}_i = \boldsymbol{G}_{l,\tilde{i}} \boldsymbol{H}_{i-1}, \quad \mathrm{dec}(\boldsymbol{H}_i, \boldsymbol{x}_i) = (\boldsymbol{M}_{lm+\tilde{i}} \cdots \boldsymbol{M}_{lm+1} \boldsymbol{G}_{l,m} \cdots \boldsymbol{G}_{l,\tilde{i}+1} \boldsymbol{H}_i)^\top (1, \ldots, n)^\top$$

The case when $l = 0$ is handled by setting $\tilde{\boldsymbol{M}}_0 = \boldsymbol{G}_{0,m} \cdots \boldsymbol{G}_{0,1}$ with $\boldsymbol{G}_{0,i} = I$. Note that both $\boldsymbol{M}_{l,\tilde{i}}$ and $\mathrm{dec}(\boldsymbol{H}_t, \boldsymbol{x}_t)$ are functions of $\tilde{i} = i \bmod m$ and the last $m + \tilde{i}$ (in general the last $2m$) tokens. Hence, the layers before the last are dedicated to output at each time-step $i$ a lookup table for the possible values of $(i \bmod 2m, w_i, \ldots, w_{i-2m+1})$. The first layers (2 if $n_h = 1$, 1 if $n_h > 1$) can provide $i \bmod 2m$ by using Lemma 2 with $d = 2m$. Finally, the second last layer can output any function of the last $2m$ tokens and the position modulo $2m$ through Lemma 3 with $d = 2m$ and $a_t = w_t$, by using $i \bmod 2m$ from the first layer(s). $\quad\square$

### B.4 Regular language recognition through products of RWKV-7 matrices

To enhance the expressivity at the cost of stability, we can replace the product of Householder matrices of DeltaProduct with a product of RWKV-7 matrices, i.e. for each layer set

$$\boldsymbol{A}(\boldsymbol{x}_i) = \prod_{j=1}^{n_h} (\mathrm{diag}(\boldsymbol{w}_{i,j}) - c\boldsymbol{k}_{i,j}(\boldsymbol{k}_{i,j} \odot \boldsymbol{a}_{i,j})), \tag{6}$$

where $\boldsymbol{w}_{i,j}$, $\boldsymbol{a}_{i,j}$, $\boldsymbol{k}_{i,j}$ are computed from $\boldsymbol{x}_i$ such that $\boldsymbol{a}_{i,j}, \boldsymbol{w}_{i,j} \in [0,1]^n$, $\|\boldsymbol{k}_{i,j}\| = 1$ and $c \in \{1, 2\}$. Even without using gates, the resulting model will be capable of recognizing regular languages effectively as the following theorem shows.

**Theorem 6.** *For any **regular language** recognized by a finite-state automaton (FSA) with $n \in \mathbb{N}$ states, there exists a linear RNN using products of RWKV-7 matrices as state-transition matrices (as in (6) with $c = 2$) with one of the following configurations that can recognize it: (i) one layer with $n_h = n$ (ii) 3 layers with $n_h > 1$ (iii) 4 layers with $n_h = 1$ [13, Theorem 3]. The construction for (ii) and (iii) requires that the MLP at the second last layer computes a lookup-table of size $2m \times (n!)^{2m}$, function of the last $2m$ input tokens and the position modulo $2m$ with $m = \lceil n/n_h \rceil$.*

*Proof.* The computation of an FSA with $n$ states can be done using matrix-vector multiplications as shown in (4), where the $n \times n$ state transition matrices $\boldsymbol{M}_{w_t}$ have elements in $\{0, 1\}$ and a single one in each column. Peng et al. [13, Lemma 3] prove that any of those matrices can be expressed as products of $n$ matrices, each of which is either a swap (identity with two columns swapped), copy (identity with one row copied onto another), or the identity matrix and can be modeled by a single RWKV-7 matrix. The proof for (ii) and (iii) follows similarly to Theorem 1 but now tackling all state transition matrices, while Theorem 1 could handle only permutations since a generalized Householders matrix cannot be a copy matrix. $\quad\square$

## B.5 Dihedral Groups

In Grazzi et al. [16, Theorem 6] it is shown that with 2 layers and the extended eigenvalue range, DeltaNet can compute addition modulo $m$, which corresponds to solving the group word problem for the cyclic group $\mathbb{Z}_m$, for any $m \in \mathbb{N}$. We extend this result and prove that, under identical assumptions, DeltaNet (DeltaProduct with $n_h = 1$) can solve the group word problem for the dihedral group $D_m$, for any $m \in \mathbb{N}$. The dihedral group $D_m$ represents the symmetries (both rotations and reflections) of a regular $m$-sided polygon. As a notable example, $D_3$ is isomorphic to the symmetric group $S_3$.

The linear RNN construction used in this result can be implemented using a 2-layer DeltaNet Model with two heads in the first layer. In the first layer, the linear RNN will compute parity for rotations and reflections separately, i.e. it will record if the number of past rotations (reflections) is even or odd. The recurrent state of the second layer will have $2m$ possible values (same as the order of $D_m$) and each will be decoded differently based on the parity of reflections. The parity of rotations, combined with the group element, determines which reflection matrix to use as the state transition matrix of the second layer.

**Theorem 7** (Dihedral group word problems with reflections). *For any $m \in N$, consider the group word problem of the dihedral group $D_m$. There exist DeltaProduct models with the following configurations that can solve it. (ii) Two layers with $n_h = 1$ and at least two heads in the first layer and one in the second layer. (ii) One layer with $n_h \geq 2$.*

*Proof.* The elements of the dihedral group $D_m$ can be divided into $m$ rotations $\mathcal{R} = \{r_0, \ldots, r_{m-1}\}$ and $m$ reflections $\mathcal{S} = \{s_0, \ldots, s_{m-1}\}$. The identity is $r_0$. To be able to solve the corresponding word problem, we would like to map sequences of group elements $x_1, \ldots, x_t$ with $x_i \in \mathcal{R} \cup \mathcal{S}$ into sequences $y_1, \ldots, y_t$ with $y_i = x_i \cdot x_{i-1} \cdots x_1$ and $\cdot$ is the group operation, that for dihedral groups is defined as

$$r_i \cdot r_j = r_{i+j \bmod m}, \quad r_i \cdot s_j = s_{i+j \bmod m}, \quad s_i \cdot r_j = s_{i-j \bmod m}, \quad s_i \cdot s_j = r_{i-j \bmod m}. \quad (7)$$

Note that a product of two rotations is commutative, while the product of two reflections or a reflection with a rotation is not. Indeed for $m \geq 3$ $D_m$, is not an abelian group.

**(i)** The constructions of the two layers of DeltaProduct with $n_h = 1$ builds upon the one for the cyclic group $Z_m$ outlined in [16, Theorem 6]. The first layer functions as a pre-processor, calculating auxiliary information from the input sequence. Specifically, for each time step $t$, it determines two parities: the parity of the total number of reflections and the parity of the total number of rotations in the sequence $x_1, \ldots, x_t$. This information is then passed to the second layer.

The second layer is responsible for computing the cumulative group product. It uses a 2D hidden state to geometrically model the group elements and their compositions. The core challenge lies in modeling the group operations, i.e. rotations and reflections, using only a reflection as state-transition matrices (rotation matrices cannot be represented with $n_h = 1$). To address this, the state representation in the second layer must encode more than just the previous group product. It is designed to also incorporate the rotation parity computed by the first layer. This is achieved by maintaining two distinct sets of $m$ state vectors each and two distinct sets of $2m$ reflection matrices each to represent the group elements. The choice of which set to use is determined by the rotation parity. Moreover, the reflection parity is also used but only in the decoder. This design allows a single, unified update mechanism, based solely on geometric reflections, to correctly implement all four of the distinct multiplication rules defined in (7).

We define rotation by and reflection matrices as

$$\text{Rotation:} \quad \boldsymbol{R}(\alpha) = \begin{pmatrix} \cos(\alpha) & -\sin(\alpha) \\ \sin(\alpha) & \cos(\alpha) \end{pmatrix} \quad \text{Reflection:} \quad \boldsymbol{H}(\alpha) = \begin{pmatrix} \cos(\alpha) & \sin(\alpha) \\ \sin(\alpha) & -\cos(\alpha) \end{pmatrix}, \quad (8)$$

where $\boldsymbol{R}(\alpha)$ is a rotation by an angle of $\alpha$, while $\boldsymbol{H}(\alpha)$ is a reflection by a line having an angle of $\alpha/2$ with the line passing from the origin and the point $(1, 0)$. Note that both $\boldsymbol{R}$ and $\boldsymbol{H}$ are periodic with period $2\pi$. Moreover, let $\alpha, \gamma \in \mathbb{R}$, the following are standard identities of products of matrix representations of 2D rotations and reflections.

$$\begin{aligned} \boldsymbol{R}(\alpha)\boldsymbol{R}(\gamma) &= \boldsymbol{R}(\alpha + \gamma), & \boldsymbol{H}(\alpha)\boldsymbol{H}(\gamma) &= \boldsymbol{R}(\alpha - \gamma), \\ \boldsymbol{R}(\alpha)\boldsymbol{H}(\gamma), &= \boldsymbol{H}(\alpha + \gamma) & \boldsymbol{H}(\gamma)\boldsymbol{R}(\alpha) &= \boldsymbol{H}(\gamma - \alpha). \end{aligned} \quad (9)$$

For the first layer we use the following diagonal recurrence which indicates in the first (second) coordinate whether the number of rotations (reflections) is even (0) or odd (1).

$$\boldsymbol{h}_0^{(1)} = 0, \quad \boldsymbol{h}_t^{(1)} = \boldsymbol{a}(x_t) \odot \boldsymbol{h}_{t-1}^{(1)} + \boldsymbol{b}(x_t), \quad \boldsymbol{y}_t^{(1)} = \operatorname{dec}^{(1)}(\boldsymbol{h}_t, x_t) = (x_t, h_{t,1}, h_{t,2}).$$

$$\boldsymbol{a}(x_i)_1 = \begin{cases} -1 & \text{if } x_i \in \mathcal{R} \\ 1 & \text{if } x_i \in \mathcal{S} \end{cases} \quad \boldsymbol{a}(x_i)_2 = \begin{cases} -1 & \text{if } x_i \in \mathcal{S} \\ 1 & \text{if } x_i \in \mathcal{R} \end{cases}$$

$$\boldsymbol{b}(x_i)_1 = \begin{cases} 1 & \text{if } x_i \in \mathcal{R} \\ 0 & \text{if } x_i \in \mathcal{S} \end{cases} \quad \boldsymbol{b}(x_i)_2 = \begin{cases} 1 & \text{if } x_i \in \mathcal{S} \\ 0 & \text{if } x_i \in \mathcal{R} \end{cases}$$

This recurrence can be implemented also by DeltaProduct with $n_h = 1$ using 2 heads each with scalar hidden states: one for the rotations and the other for the reflections. For the second layer, we have instead the following constructions, which selects the appropriate reflection based on the parity of the rotations and uses the parity of the reflections for dec.

$$\boldsymbol{h}_0^{(2)} = (1,0)^\top, \quad \boldsymbol{h}_t^{(2)} = \boldsymbol{A}^{(2)}(\boldsymbol{y}_t^{(1)})\boldsymbol{h}_{t-1}^{(2)}, \quad \boldsymbol{y}_t^{(2)} = \operatorname{dec}^{(2)}(\boldsymbol{h}_t^{(2)}, \boldsymbol{y}_t^{(1)}),$$

$$\boldsymbol{A}^{(2)}(\boldsymbol{y}) = \boldsymbol{H}(\theta(y_1, y_2)),$$

$$\operatorname{dec}^{(2)}(\boldsymbol{h}, \boldsymbol{y}) = \begin{cases} r_{i^*} & \text{if } y_3 = 0 \\ s_{m-i^*} & \text{if } y_3 = 1 \end{cases}, \quad i^* = \operatorname*{arg\,max}_{i \in \{0,\dots,m-1\}} \max(\boldsymbol{c}_i^\top \boldsymbol{h}, \boldsymbol{d}_i^\top \boldsymbol{h})$$

where $\boldsymbol{y} = (y_1, y_2, y_3)^\top \in \mathcal{R} \cup \mathcal{S} \times \{0,1\} \times \{0,1\}$ and $\theta : \mathcal{R} \cup \mathcal{S} \times \{0,1\} \to \mathbb{R}$ determines the angle of the reflection and is defined for all $i \in \{0,\dots,m-1\}$ as

$$\theta(r_i, 1) = \frac{(1-2i)\pi}{m}, \quad \theta(r_i, 0) = \frac{(1+2i)\pi}{m}, \quad \theta(s_i, 1) = \frac{-2i\pi}{m}, \quad \theta(s_i, 0) = \frac{(2+2i)\pi}{m}.$$

Moreover, $\mathcal{C} = \{\boldsymbol{c}_0, \dots, \boldsymbol{c}_{m-1}\}$ and $\mathcal{D} = \{\boldsymbol{d}_0, \dots, \boldsymbol{d}_{m-1}\}$ are two sets of states and are defined as

$$\boldsymbol{d}_0 = \boldsymbol{h}_0^{(2)} = (1,0)^\top, \quad \boldsymbol{c}_0 = \boldsymbol{H}(\pi/m)\boldsymbol{d}_0,$$
$$\boldsymbol{d}_i = \boldsymbol{R}(2i\pi/m)\boldsymbol{d}_0, \quad \boldsymbol{c}_i = \boldsymbol{R}(-2i\pi/m)\boldsymbol{c}_0 \quad \text{for all } i \in \{0, \dots, m-1\}.$$

From our choice of $\boldsymbol{d}_0 = (1,0)^\top$ and $\boldsymbol{c}_0$ and from (8)-(9), for any $\alpha \in \mathbb{R}$ we have

$$\boldsymbol{R}(\alpha)\boldsymbol{d}_0 = \boldsymbol{H}(\alpha)\boldsymbol{d}_0, \quad \text{and}$$
$$\boldsymbol{R}(\alpha)\boldsymbol{c}_0 = \boldsymbol{R}(\alpha)\boldsymbol{H}(\pi/m)\boldsymbol{d}_0 = \boldsymbol{R}(\alpha)\boldsymbol{R}(\pi/m)\boldsymbol{d}_0 = \boldsymbol{R}(\alpha + \pi/m + \pi/m - \pi/m)\boldsymbol{d}_0$$
$$= \boldsymbol{H}(\alpha + 2\pi/m)\boldsymbol{H}(\pi/m)\boldsymbol{d}_0 = \boldsymbol{H}(\alpha + 2\pi/m)\boldsymbol{c}_0.$$

Moreover, from our choice of $\theta$, $\boldsymbol{d}_i$ and $\boldsymbol{c}_i$, using the identities above and the the fact that $\boldsymbol{R}$ is a periodic function with period $2\pi$ we have that

$$\boldsymbol{d}_i = \boldsymbol{R}(2i\pi/m)\boldsymbol{d}_0 = \boldsymbol{R}(2i\pi/m)\boldsymbol{H}(\pi/m)\boldsymbol{c}_0 = \boldsymbol{H}(\theta(r_i, 0))\boldsymbol{c}_0$$
$$\boldsymbol{c}_i = \boldsymbol{R}(-2i\pi/m)\boldsymbol{c}_0 = \boldsymbol{R}(-2i\pi/m)\boldsymbol{H}(\pi/m)\boldsymbol{d}_0 = \boldsymbol{H}(\theta(r_i, 1))\boldsymbol{d}_0$$
$$\boldsymbol{d}_{m-i} = \boldsymbol{R}(-2i\pi/m)\boldsymbol{d}_0 = \boldsymbol{H}(-2i\pi/m)\boldsymbol{d}_0 = \boldsymbol{H}(\theta(s_i, 1))\boldsymbol{d}_0$$
$$\boldsymbol{c}_{m-i} = \boldsymbol{R}(+2i\pi/m)\boldsymbol{c}_0 = \boldsymbol{H}((2+2i)\pi/m)\boldsymbol{c}_0 = \boldsymbol{H}(\theta(s_i, 0))\boldsymbol{c}_0$$

for every $i \in \{0, \dots, m-1\}$. Therefore, we can write

$$\boldsymbol{H}(\theta(r_j, 1))\boldsymbol{d}_i = \boldsymbol{R}(\theta(r_j, 1) - \theta(r_i, 0))\boldsymbol{c}_0 = \boldsymbol{R}(-2(i+j)\pi/m)\boldsymbol{c}_0 = \boldsymbol{c}_{i+j \bmod m},$$
$$\boldsymbol{H}(\theta(r_j, 0))\boldsymbol{c}_i = \boldsymbol{R}(\theta(r_j, 0) - \theta(r_i, 1))\boldsymbol{d}_0 = \boldsymbol{R}(2(i+j)\pi/m)\boldsymbol{d}_0 = \boldsymbol{d}_{i+j \bmod m},$$
$$\boldsymbol{H}(\theta(s_j, 1))\boldsymbol{d}_i = \boldsymbol{R}(\theta(s_j, 1) - \theta(s_{m-i}, 1))\boldsymbol{d}_0 = \boldsymbol{R}(-2(i+j)\pi/m)\boldsymbol{d}_0 = \boldsymbol{d}_{-i-j \bmod m},$$
$$\boldsymbol{H}(\theta(s_j, 0))\boldsymbol{c}_i = \boldsymbol{R}(\theta(s_j, 0) - \theta(s_{m-i}, 0))\boldsymbol{c}_0 = \boldsymbol{R}(2(i+j)\pi/m)\boldsymbol{c}_0 = \boldsymbol{c}_{-i-j \bmod m},$$

$$(10)$$

for every $i, j \in \{0, \dots, m-1\}$. We proceed to verify that the output of the second layer is computed correctly: satisfying the product rule for the dihedral group in (7), i.e., we want to verify that

$$y_t^{(2)} = \begin{cases} r_{i+j \bmod m} & \text{if } y_{t-1}^{(2)} = r_i, x_t = r_j \\ s_{i+j \bmod m} & \text{if } y_{t-1}^{(2)} = r_i, x_t = s_j \\ s_{i-j \bmod m} & \text{if } y_{t-1}^{(2)} = s_i, x_t = r_j \\ r_{i-j \bmod m} & \text{if } y_{t-1}^{(2)} = s_i, x_t = s_j \end{cases}$$

$$(11)$$

Where we set $y_0^{(2)} = r_0$. First note that when $y_t^{(2)} \in \mathcal{S}$, then $y_{t,3}^{(1)} = 1$ and when $y_t^{(2)} \in \mathcal{R}$, then $y_{t,3}^{(1)} = 0$. We consider two cases.

**Case 1.** If $y_{t-1}^{(2)} = r_i$ and hence $y_{t-1,3}^{(1)} = 0$, then using (10) we obtain

$$
\boldsymbol{h}_t^{(2)} = \boldsymbol{A}^{(2)}(\boldsymbol{y}^{(1)})\boldsymbol{h}_{t-1}^{(2)} = \begin{cases}
\boldsymbol{H}(\theta(r_j, 1))\boldsymbol{d}_i = \boldsymbol{c}_{i+j \bmod m} & \text{if } x_t = r_j, y_{t,2}^{(1)} = 1 \\
\boldsymbol{H}(\theta(r_j, 0))\boldsymbol{c}_i = \boldsymbol{d}_{i+j \bmod m} & \text{if } x_t = r_j, y_{t,2}^{(1)} = 0 \\
\boldsymbol{H}(\theta(s_j, 1))\boldsymbol{d}_i = \boldsymbol{d}_{-i-j \bmod m} & \text{if } x_t = s_j, y_{t,2}^{(1)} = 1 \\
\boldsymbol{H}(\theta(s_j, 0))\boldsymbol{c}_i = \boldsymbol{c}_{-i-j \bmod m} & \text{if } x_t = s_j, y_{t,2}^{(1)} = 0
\end{cases}
$$

This, together with the definition of $\mathrm{dec}^{(2)}$ implies that

$$
y_t^{(2)} = \mathrm{dec}^{(2)}(\boldsymbol{h}_t^{(2)}, \boldsymbol{y}_t^{(1)}) = \begin{cases}
r_{i+j \bmod m} & \text{if } x_t = r_j, y_{t,3}^{(1)} = 0 \\
s_{i+j \bmod m} & \text{if } x_t = s_j, y_{t,3}^{(1)} = 1
\end{cases} \tag{12}
$$

**Case 2.** If instead $y_{t-1}^{(2)} = s_i$ and hence $y_{t-1,3}^{(1)} = 1$, then using (10) we obtain

$$
\boldsymbol{h}_t^{(2)} = \boldsymbol{A}^{(2)}(\boldsymbol{y}^{(1)})\boldsymbol{h}_{t-1}^{(2)} = \begin{cases}
\boldsymbol{H}(\theta(r_j, 1))\boldsymbol{d}_{m-i} = \boldsymbol{c}_{j-i \bmod m} & \text{if } x_t = r_j, y_{t,2}^{(1)} = 1 \\
\boldsymbol{H}(\theta(r_j, 0))\boldsymbol{c}_{m-i} = \boldsymbol{d}_{j-i \bmod m} & \text{if } x_t = r_j, y_{t,2}^{(1)} = 0 \\
\boldsymbol{H}(\theta(s_j, 1))\boldsymbol{d}_{m-i} = \boldsymbol{d}_{i-j \bmod m} & \text{if } x_t = s_j, y_{t,2}^{(1)} = 1 \\
\boldsymbol{H}(\theta(s_j, 0))\boldsymbol{c}_{m-i} = \boldsymbol{c}_{i-j \bmod m} & \text{if } x_t = s_j, y_{t,2}^{(1)} = 0
\end{cases}
$$

This, together with the definition of $\mathrm{dec}^{(2)}$ implies that

$$
y_t^{(2)} = \mathrm{dec}^{(2)}(\boldsymbol{h}_t^{(2)}, \boldsymbol{y}_t^{(1)}) = \begin{cases}
s_{i-j \bmod m} & \text{if } x_t = r_j, y_{t,3}^{(1)} = 1 \\
r_{i-j \bmod m} & \text{if } x_t = s_j, y_{t,3}^{(1)} = 0
\end{cases}. \tag{13}
$$

Note that (12) and (13) imply (11). Setting the output of the linear RNN equal to the output of the second layer concludes the proof.

**(ii)** It follows from Theorem 4 since $D_m$ is a finite subgroup of $O(2)$, the group of 2D orthogonal transformations: rotations and reflections. $\qquad\square$

### B.6   Stability vs. Expressivity of Linear RNNs

In this section, we discuss the tradeoff between expressivity and stability of a linear RNN recurrence $\boldsymbol{H}_i = \boldsymbol{A}_i\boldsymbol{H}_{i-1} + \boldsymbol{B}_i$, where $\boldsymbol{A}_i = \boldsymbol{A}(\boldsymbol{x}_i)$, $\boldsymbol{B}_i = \boldsymbol{B}(\boldsymbol{x}_i)$. We say that such a recurrence is stable if

$$
\exists M \in [0, \infty) \text{ such that } \left\| \prod_{j=1}^{i} \boldsymbol{A}_j \right\| < M \quad \forall i \in \mathbb{N}, \tag{14}
$$

where $\|\cdot\|$ is the spectral norm. This property is true if and only if $\rho(\prod_{j=1}^{i} \boldsymbol{A}_j) \leq 1$ where $\rho(\boldsymbol{M})$ is the spectral radius of $\boldsymbol{M}$, i.e. the maximum modulus of its eigenvalues. When this property is not satisfied, the norm of the state will diverge. An effective way to satisfy (14) with $M = 1$ is to enforce $\|\boldsymbol{A}_i\| \leq 1$ for every $i$, since the norm of the product less than or equal to the product of the norms (due to the submultiplicativity property). However, this restriction excludes some boolean matrices which are useful for recognizing regular languages. Indeed, in the construction shown in (4), all matrices involved are $n \times n$ with entries taking values in $\{0, 1\}$ and having only a single one in each column. This class of matrices $\mathcal{B}$ satisfies (14) with $M = \sqrt{n}$ because it is closed under matrix multiplication, i.e. $\forall \boldsymbol{B}, \boldsymbol{B}' \in \mathcal{B}$, we have $\boldsymbol{B}\boldsymbol{B}' \in \mathcal{B}$, and $\max_{\boldsymbol{B} \in \mathcal{B}} \|\boldsymbol{B}\| = \sqrt{n}$, which is achieved by matrices with ones only in one row. In particular, all matrices in $\mathcal{B}$ that are not permutations have spectral norm greater than one and therefore cannot be expressed if we enforce $\|\boldsymbol{A}_i\| \leq 1$.

The (Gated) DeltaProduct state transition matrix $\boldsymbol{A}_i = \prod_{l=1}^{n_h} g_l(I - \beta_l \boldsymbol{k}_l \boldsymbol{k}_l^\top)$ satisfies $\|\boldsymbol{A}_i\| \leq 1$ since $g_l \in [0, 1]$, $\beta_l \in [0, 2]$, and $\|\boldsymbol{k}_l\| = 1$. Thus, from the matrices in $\mathcal{B}$, it can represent only permutations of up to $n_h + 1$ elements. Instead, the state-transition matrix of RWKV-7, $\boldsymbol{A}_i = \mathrm{diag}(w_i) - c\boldsymbol{k}_i(\boldsymbol{k}_i \odot \boldsymbol{a}_i)^\top$ with $c = 2$, can represent not only the identity and permutations of two

elements, but also any copy matrix, which is obtained by copying one column of the identity onto another and has spectral norm equal to $\sqrt{2}$. However, as we show in the next theorem, even with the less expressive $c = 1$ setup that is used in practice, the RWKV-7 recurrence is not stable unless $\boldsymbol{a}_i$ is the same for every $i$, which is the case studied in Peng et al. [13, Theorem 1]. Having different $\boldsymbol{a}_i$ values is key to modeling the copy matrix, since this requires a value different from that of a permutation matrix.

**Theorem 8.** *Consider the RWKV-7 state transition matrix $\boldsymbol{A}_i = \mathrm{diag}(w_i) - c\boldsymbol{k}_i(\boldsymbol{k}_i \odot \boldsymbol{a}_i)^\top$ with $c = 1$ (as set in practice), $w_i = (1,\ldots,1)^\top \in \mathbb{R}^n$, $\boldsymbol{a}_i \in [0,1]^n$, $\boldsymbol{k}_i \in \mathbb{R}^n$ with $\|\boldsymbol{k}_i\| = 1$, and $n \geq 2$. There exists an infinite set $\mathcal{M}$ of matrix pairs such that for every $(\boldsymbol{A}, \boldsymbol{A}') \in \mathcal{M}$, we have $\rho(\boldsymbol{A}\boldsymbol{A}') > 1.2$, where $\rho$ denotes the spectral radius. Thus, if we set*

$$\boldsymbol{A}_i = \begin{cases} \boldsymbol{A} & \text{if } i \bmod 2 = 0 \\ \boldsymbol{A}' & \text{if } i \bmod 2 = 1 \end{cases}, \quad \text{which implies} \quad \lim_{i \to \infty} \left\| \prod_{j=1}^{i} \boldsymbol{A}_j \right\| = \lim_{i \to \infty} = \infty.$$

*Proof.* We demonstrate this by construction for $n = 2$; the generalization to $n \geq 2$ is straightforward.

Let $\theta = \pi/3$, $\boldsymbol{a} = (0,1)^\top$, $\boldsymbol{a}' = (1,0)^\top$. $\boldsymbol{k} = (\cos\theta, \sin\theta)^\top = [1/2, \sqrt{3}/2]^\top$. $\boldsymbol{k}' = (\sin\theta, \cos\theta)^\top = (\sqrt{3}/2, 1/2)^\top$. Note that $\|\boldsymbol{k}\| = \|\boldsymbol{k}'\| = 1$. We construct $\boldsymbol{A}$ and $\boldsymbol{A}'$ as

$$\boldsymbol{A} = I - \boldsymbol{k}(\boldsymbol{k} \odot \boldsymbol{a})^\top = I - \begin{pmatrix} \sqrt{3}/2 \\ 1/2 \end{pmatrix} [0 \quad 1/2] = \begin{pmatrix} 1 & -\sqrt{3}/4 \\ 0 & 3/4 \end{pmatrix}$$

$$\boldsymbol{A}' = I - \boldsymbol{k}'(\boldsymbol{k}' \odot \boldsymbol{a}')^\top = I - \begin{pmatrix} 1/2 \\ \sqrt{3}/2 \end{pmatrix} [1/2 \quad 0] = \begin{pmatrix} 3/4 & 0 \\ -\sqrt{3}/4 & 1 \end{pmatrix}$$

Now, consider the product matrix

$$\boldsymbol{M} = \boldsymbol{A}\boldsymbol{A}' = \begin{pmatrix} 1 & -\sqrt{3}/4 \\ 0 & 3/4 \end{pmatrix} \begin{pmatrix} 3/4 & 0 \\ -\sqrt{3}/4 & 1 \end{pmatrix} = \begin{pmatrix} 15/16 & -\sqrt{3}/4 \\ -3\sqrt{3}/16 & 3/4 \end{pmatrix}$$

To find the spectral radius $\rho(M)$, we examine its eigenvalues. The characteristic equation is $\lambda^2 - \mathrm{Tr}(\boldsymbol{M})\lambda + \det(\boldsymbol{M}) = 0$, with $\mathrm{Tr}(\boldsymbol{M}) = 27/16$ and $\det(\boldsymbol{M}) = 9/16$. Hence, the characteristic equation is $16\lambda^2 - 27\lambda + 9 = 0$. Thus, the eigenvalues are $\lambda_1 = \frac{27+\sqrt{153}}{32}$ and $\lambda_2 = \frac{27-\sqrt{153}}{32}$ and the spectral radius is $\rho(\boldsymbol{M}) = \max\{|\lambda_1|, |\lambda_2|\} = \lambda_1 \approx 1.23$. Also, since the spectral radius is a continuous function of the matrix entries, which are a continuous function of $\theta$, then this means that there is an infinite set of matrices, namely $\mathcal{M}$, obtained by varying $\theta$ around $\pi/3$ whose product has spectral radius greater than 1.2.

From our construction of $\boldsymbol{A}_i$, we have $\prod_{j=1}^{2i} \boldsymbol{A}_j = \boldsymbol{M}^i$. By the definition of the spectral norm and spectral radius, $\|\boldsymbol{M}^i\| \geq \|\lambda_1^i \boldsymbol{x}\| = |\lambda|^i = \rho(\boldsymbol{M})^i$, where $\boldsymbol{x}$ is the eigenvector associated with the dominant eigenvalue $\lambda_1$. The result follows since $\lim_{i \to \infty} \rho(\boldsymbol{M})^i = \infty$. $\qquad\square$

## C  Experiments

### C.1  State-Tracking

**Clarification on the isomorphisms of $S_3$, $S_4$, $A_5$, and $S_5$**

$S_3$: The group consisting of all isometries that map an equilateral triangle onto itself, including both orientation-preserving rotations and orientation-reversing reflections, is isomorphic to $S_3$.

$S_4$: The rotation group of a cube is isomorphic to the symmetric group $S_4$. This correspondence arises because the cube has exactly four space diagonals, and every *proper rotation*—that is, every orientation-preserving isometry of the cube about an axis through its center—permutes these diagonals in all possible ways (see Figure 6 for an example). In particular, these proper rotations include, for example, the $90°$, $180°$, and $270°$ rotations about axes passing through the centers of opposite faces, the $180°$ rotations about axes through the midpoints of opposite edges, and the $120°/240°$

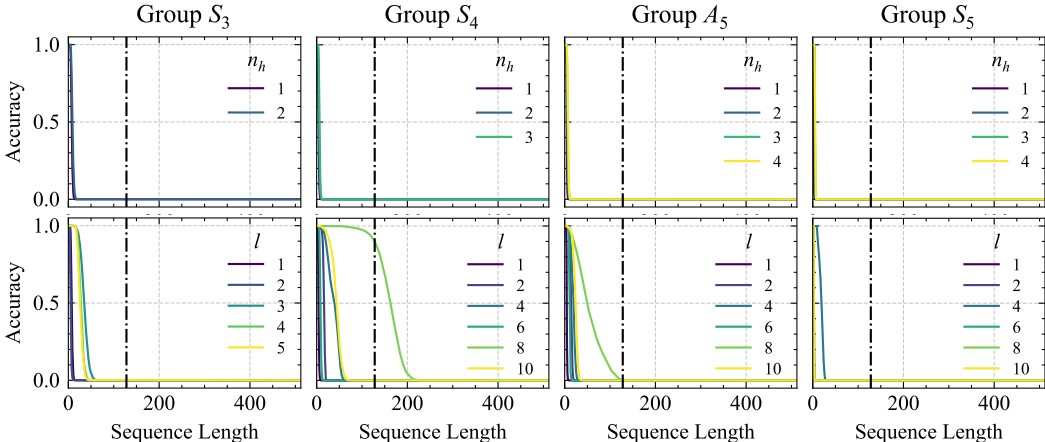

Figure 12: Results for permutation groups $S_3$, $S_4$, $A_5$, and $S_5$ when limiting the eigenvalue range of the state-transition matrix to $[0, 1]$. *(Top row)* Varying the number of Householder products $n_h$ for a single layer DeltaProduct$_{n_h}[0, 1]$. *(Bottom row)* Varying the number of layers $l$ of DeltaProduct$_1[0, 1]$/DeltaNet$[0, 1]$ (single Householder). Dashed vertical line at training context length 128. Higher $n_h$ improves extrapolation to longer sequences of permutations, e.g., $S_3$ can be learned with $n_h = 2$ with a single layer while three layers are required when keeping $n_h = 1$.

rotations about axes through opposite vertices. Hence, the proper rotational symmetries of the cube correspond precisely to the permutations of its four space diagonals [69].

$A_5$: Similarly, a regular dodecahedron contains exactly five special cubes symmetrically arranged within it. Each *proper rotation* of the dodecahedron—that is, every orientation-preserving rigid motion mapping the dodecahedron onto itself—rearranges these inscribed cubes by an *even permutation*. This property makes the rotation group of the dodecahedron isomorphic to the alternating group $A_5$, the group of all even permutations of five elements [70].

$S_5$: When both proper rotations and reflections (orientation-reversing symmetries) are considered, the full symmetry group of the dodecahedron corresponds exactly to the symmetric group $S_5$, since reflections allow both even and odd permutations of the five hidden cubes [70].

**Experimental Details.** We used the experimental setup from Merrill et al. [3] and sampled 2,000,000 training datapoints at sequence length 128 and 500,000 test datapoints at sequence length 512. We did not use a curriculum over sequence length during training. The models were trained using AdamW optimizer [71] with parameters $\beta_1 = 0.9$, $\beta_2 = 0.999$, and $\epsilon = 10^{-8}$ in PyTorch [72]. We used a learning rate of $10^{-3}$ with cosine annealing [73] and trained for 100 epochs with a batch size of 1024, except for the $S_3$ models which required a batch size of 2048 for more reliable results. All models used a single-layer DeltaProduct architecture featuring 12 heads (more heads made the results more reliable) and a head dimension of 32. We applied a weight decay coefficient of $10^{-6}$. The $\beta$ values were extracted from the forward pass of the trained models using NNsight [74]. We use the PCA implementation in scikit-learn [75].

## C.2 Chomsky Hierarchy

**Setup.** We conducted experiments on selected formal language tasks originally introduced by Delétang et al. [52]. Our goal was to demonstrate the improvements in length extrapolation that can be achieved using multiple Householder matrices in the state-transition matrix compared to DeltaNet. Following Grazzi et al. [16], we focus on three tasks: parity, modular arithmetic without brackets (both regular languages), and modular arithmetic with brackets (a context-free language). We trained DeltaProduct$_{n_h}$ with $n_h \in \{2, 3, 4\}$ on sequences of length 3 to 40 and tested on sequences ranging from 40 to 256 to evaluate generalization to longer inputs. We compare our results against the results obtained by Grazzi et al. [16] for Transformer, mLSTM and sLSTM from Beck et al. [9], Mamba [6], and DeltaNet [10]. For both Mamba and DeltaNet, we experiment with an eigenvalue range restricted to $[0, 1]$ and extended to $[-1, 1]$.

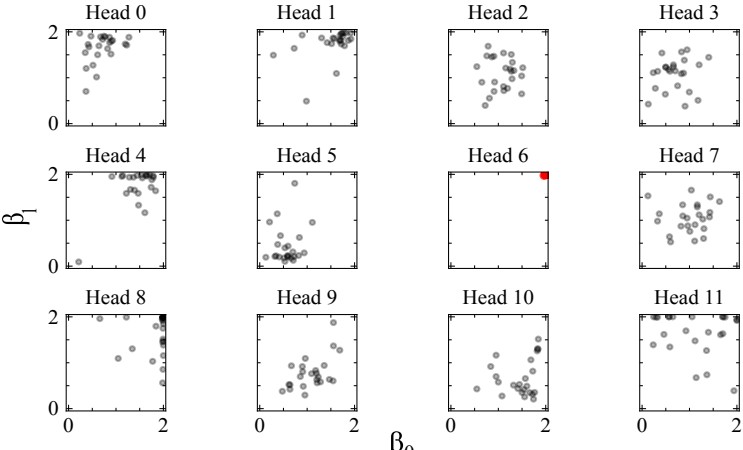

Figure 13: $\beta_0$ and $\beta_1$ values across all 24 permutations in $S_4$ in DeltaProduct$_2[-1, 1]$. We find that only head 6 (shown in Figure 7) learns to use both Householders as reflections ($\beta_0 \approx 2$, $\beta_1 \approx 2$) allowing it to learn the rotations to solve $S_4$.

**Experimental Details.** All DeltaProduct and DeltaNet models contain 3 layers with 1 head each and heads' dimensions set to 128, except for modular arithmetic with brackets, where we use 12 heads and set the heads' dimensions to 32. Both models use a causal depthwise 1D convolution with a kernel size of 4 after the query/key/value projection. For modular arithmetic, we also use a gradient clipping norm of 1.0. We train each model using AdamW [71] using a learning rate of 5e-4, batch size of 1024, 0.1 weight decay, and a cosine annealing learning rate schedule [73] (minimum learning rate: 1e-6) after 10% warm-up steps. We train on the modular arithmetic and parity tasks for 100k and 20k steps in total, respectively. At each training step, we make sure to generate a valid random sample from the task at hand (see below). We repeat the runs 3 times with different seeds each, and later pick the best to report in Table 2.

**Considered Tasks.** We empirically evaluated three tasks—parity, modular arithmetic without brackets, and modular arithmetic with brackets—spanning different levels of the Chomsky Hierarchy. Below, we provide details for each task, where $|\Sigma|$ denotes the vocabulary size and $Acc_{rand}$ represents the accuracy of random guessing:

- **Parity** ($|\Sigma| = 2$, $Acc_{rand} = 0.5$). Given a binary sequence $\boldsymbol{x} = x_1 \ldots x_t \in \{0, 1\}^t$, the parity label $y_t \in \{0, 1\}$ is 1 if the total number of ones in the sequence is odd, and 0 otherwise. This task is equivalent to computing the sum of all previous values modulo 2, i.e., $y_t = (\sum_{i=1}^t x_i) \bmod 2$.

- **Modular Arithmetic without Brackets** ($|\Sigma| = 10$, $Acc_{rand} = 1/5$). Given a set of special tokens $\Sigma_s = \{+, -, *, =, [\texttt{PAD}]\}$ and a modulus $m \geq 1$, we define $\Sigma = \Sigma_s \cup \{0, \ldots, m-1\}$. The label $y_t$ corresponds to the result of evaluating the arithmetic operations in the sequence $\boldsymbol{x} = x_1, \ldots, x_t$, computed modulo $m$. In our experiments, we set $m = 5$. An example is:

$$2 + 1 - 2 * 2 - 3 = 1\ [\texttt{PAD}]$$

- **Modular Arithmetic with Brackets** ($|\Sigma| = 12$, $Acc_{rand} = 1/5$). This task follows the same definition as modular arithmetic without brackets but includes an extended set of special tokens, $\Sigma_s = \{+, -, *, =, ), (, [\texttt{PAD}]\}$, allowing for nested expressions. Again, we set $m = 5$. An example sequence is:

$$((1 - (-2)) + ((4) + 3)) = 0\ [\texttt{PAD}]$$

**Results.** As shown in Table 2, DeltaProduct$_{n_h}$ with $n_h \geq 2$ has better average accuracy compared to DeltaNet and other baselines. This performance improvement is particularly pronounced when using the extended eigenvalue range $[-1, 1]$, which aligns with the findings of Grazzi et al. [16]. Notably, we observe the most significant improvement in the modular arithmetic with brackets task, which is also the most challenging.

Table 2: Performance of DeltaProduct$_{n_h}[-1, 1]$, $n_h \in \{2, 3, 4\}$, on formal language tasks. We report the best of 3 runs. Scores are scaled accuracy, with 1.0 indicating perfect performance and 0.0 random guessing. The results for the other models were taken directly from Grazzi et al. [16].

| Model | Parity | Mod. Arith. (w/o brackets) | Mod. Arith. (w/ brackets) | Avg. |
|---|---|---|---|---|
| Transformer | 0.022 | 0.031 | 0.067 | 0.040 |
| mLSTM | 0.087 | 0.040 | 0.114 | 0.080 |
| sLSTM | **1.000** | **0.787** | **0.178** | **0.655** |
| Mamba $[0, 1]$ | 0.000 | 0.095 | **0.123** | 0.073 |
| Mamba $[-1, 1]$ | **1.000** | **0.241** | 0.116 | **0.452** |
| DeltaNet $[0, 1]$ | 0.233 | 0.302 | 0.253 | 0.263 |
| DeltaProduct$_2$ $[0, 1]$ | 0.264 | **0.402** | 0.249 | 0.305 |
| DeltaProduct$_3$ $[0, 1]$ | 0.285 | **0.402** | **0.288** | **0.325** |
| DeltaProduct$_4$ $[0, 1]$ | **0.295** | 0.369 | **0.288** | 0.317 |
| DeltaNet $[-1, 1]$ | **0.982** | **0.915** | 0.281 | 0.726 |
| DeltaProduct$_2$ $[-1, 1]$ | 0.896 | 0.887 | 0.329 | 0.704 |
| DeltaProduct$_3$ $[-1, 1]$ | 0.932 | 0.736 | 0.330 | 0.666 |
| DeltaProduct$_4$ $[-1, 1]$ | **0.982** | 0.893 | **0.342** | **0.739** |

## C.3 Language Modeling

### C.3.1 Experimental setup

We follow the same basic training setup as in [16]. We use the training pipeline `flame` from the flash-linear-attention [22] repository. All of our models are trained on NVIDIA L40s, NVIDIA A100 40GB or NVIDIA H100 94GB GPUs. We used 16 to 32 GPUs at a time to train one model, in a 2 to 8 node setup, depending on resource availability. We used DeepSpeed with ZeRO-2 [76] for distributed training. All models were trained with an effective batch size of $524\,288$ tokens, and a learning rate of 3e-4. We optimized the models with AdamW [71] (0.01 weight decay) and used cosine annealing [73] for the learning rate schedule with linear warm up for 512 steps. We used a total of $10\,500$ GPU hours to train all of our models.

### C.3.2 Throughput

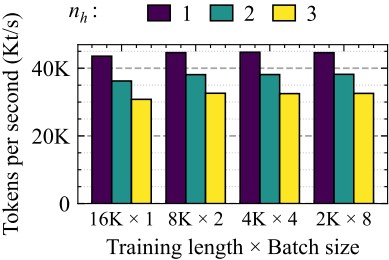

Figure 14: Training throughput of a parameter matched DeltaProduct 1.3B. Parameter matching is achieved by decreasing the inner dimension in the SwiGLU MLP for $n_h > 1$.

### C.3.3 Additional Benchmarks

In Table 3 we report evaluations for the models in Figure 10 on tasks from lm-eval-harness [61]. In addition, we also train and evaluate models with $2048$ context length at the $340M$ parameter scale and report the results in Table 5 and compare them with the results in [10] which are trained under a comparable setup. We observe that DeltaProduct outperforms DeltaNet in terms of average accuracy for both training setups.

**Tasks Details.** We use the lm-eval-harness benchmark [61] to assess model performance. Following Yang et al. [10], the evaluation encompasses multiple task categories: **Language Understanding Tasks.** The evaluation includes LAMBADA (LMB) [77] for testing text comprehension, PIQA [78] for physical reasoning assessment, HellaSwag (Hella.) [79] for situational understanding, and Winogrande (Wino.) [80] for commonsense reasoning evaluation. **Reasoning.** The ARC dataset provides two distinct testing sets: ARC-easy (ARC-e) and ARC-challenge (ARC-c) [81], measuring varying levels of scientific knowledge comprehension.

Table 3: Performance comparison of models shown in Figure 10. Parameter equivalence was achieved by scaling the head dimension. To account for the increased parameter count we scaled the training token budget from 19B (213M parameters) to 55B (805M parameters) on FineWeb [57]. Models were trained on 4096 token context length.

| | Model | Wiki. ppl↓ | LMB. ppl↓ | LMB. acc↑ | PIQA acc↑ | Hella. acc_n↑ | Wino. acc↑ | ARC-e acc↑ | ARC-c acc_n↑ | Avg. ↑ |
|---|---|---|---|---|---|---|---|---|---|---|
| *19B / 213M* | DeltaNet[−1, 1] | 32.39 | 107.41 | 20.4 | 65.3 | 35.1 | 52 | 44.1 | 24.5 | 40.23 |
| | DeltaProduct$_2$[−1, 1] | 31.46 | 78.98 | 23.9 | 64.6 | 36.2 | 52.6 | 45 | 23 | 40.88 |
| | DeltaProduct$_3$[−1, 1] | 30.94 | 70.5 | 24.6 | 66.3 | 36.8 | 49.1 | 46 | 23.7 | **41.01** |
| *35B / 392M* | DeltaNet[−1, 1] | 25.5 | 40.32 | 30.2 | 68.5 | 41 | 51.9 | 47.3 | 23.3 | 43.7 |
| | DeltaProduct$_2$[−1, 1] | 24.82 | 34.31 | 33.3 | 68.9 | 43.4 | 50.7 | 49.2 | 25 | **45.08** |
| | DeltaProduct$_3$[−1, 1] | 24.81 | 37.13 | 31.1 | 68.5 | 43.3 | 50 | 48.2 | 23.7 | 44.1 |
| *55B / 805M* | DeltaNet[−1, 1] | 20.81 | 20.57 | 37.8 | 71.5 | 48.9 | 55.6 | 51.9 | 25.6 | 48.55 |
| | DeltaProduct$_2$[−1, 1] | 20.54 | 19.56 | 38.3 | 71 | 50.7 | 55.2 | 52.1 | 26.7 | 49 |
| | DeltaProduct$_3$[−1, 1] | 20.01 | 15.56 | 42.9 | 71.4 | 51.4 | 53 | 54.6 | 26.4 | **49.95** |

Table 4: Performance comparison of models shown in Figure 21. Parameter equivalence was achieved by scaling the number of heads in the attention. To account for the increased parameter count we scaled the training token budget from 19B (213M parameters) to 55B (805M parameters) on FineWeb [57]. Models were trained on 4096 token context length.

| | Model | Wiki. ppl↓ | LMB. ppl↓ | LMB. acc↑ | PIQA acc↑ | Hella. acc_n↑ | Wino. acc↑ | ARC-e acc↑ | ARC-c acc_n↑ | Avg. ↑ |
|---|---|---|---|---|---|---|---|---|---|---|
| *19B / 213M* | DeltaNet[−1, 1] | 31.96 | 85.36 | 22.5 | 65.2 | 35.4 | 50.8 | 44.7 | 22.4 | 40.17 |
| | DeltaProduct$_2$[−1, 1] | 30.87 | 89.23 | 23.1 | 65.4 | 36.5 | 51.2 | 43.5 | 22.4 | 40.35 |
| | DeltaProduct$_3$[−1, 1] | 30.85 | 71.52 | 24.1 | 66.3 | 36.1 | 51.6 | 44.1 | 23.9 | **41.02** |
| *35B / 392M* | DeltaNet[−1, 1] | 24.86 | 39.1 | 30.8 | 69.2 | 41.4 | 50.5 | 46.7 | 24.4 | 43.83 |
| | DeltaProduct$_2$[−1, 1] | 24.97 | 35.68 | 31.9 | 69.6 | 42.5 | 52.6 | 47.4 | 25.9 | **44.98** |
| | DeltaProduct$_3$[−1, 1] | 25.2 | 40.96 | 30.5 | 69.1 | 42.3 | 51.4 | 47.7 | 23.9 | 44.15 |
| *55B / 805M* | DeltaNet[−1, 1] | 20.6 | 21.18 | 38.7 | 71.5 | 48.7 | 52.8 | 51.9 | 25.7 | 48.22 |
| | DeltaProduct$_2$[−1, 1] | 20.26 | 17.41 | 40.7 | 72.6 | 50.3 | 53.9 | 52.4 | 24.9 | 49.13 |
| | DeltaProduct$_3$[−1, 1] | 19.97 | 17.78 | 40.79 | 72.3 | 50.9 | 52.1 | 53.9 | 26.4 | **49.4** |

Table 5: Performance comparison of models trained with 2048 context length. (SlimPajama (SPJ) reproduced from Yang et al. [10], Fine-Web (FW) ours). Results are shown for DeltaProduct and Gated DeltaProduct. We use 8 heads for each layer, unless otherwise specified.

| | Model | Wiki. ppl↓ | LMB. ppl↓ | LMB. acc↑ | PIQA acc↑ | Hella. acc_n↑ | Wino. acc↑ | ARC-e acc↑ | ARC-c acc_n↑ | Avg. ↑ |
|---|---|---|---|---|---|---|---|---|---|---|
| *15B tokens SPJ* | *340M params* | | | | | | | | | |
| | Transformer++ | 28.39 | 42.69 | 31.0 | 63.3 | 34.0 | 50.4 | 44.5 | 24.2 | 41.2 |
| | Mamba [0, 1] | 28.39 | 39.66 | 30.6 | 65.0 | **35.4** | 50.1 | **46.3** | 23.6 | 41.8 |
| | GLA [0, 1] | 29.47 | 45.53 | 31.3 | **65.1** | 33.8 | 51.6 | 44.4 | **24.6** | 41.8 |
| | DeltaNet [0, 1] | **28.24** | **37.37** | **32.1** | 64.8 | 34.3 | **52.2** | 45.8 | 23.5 | **42.1** |
| *35B FW* | DeltaNet[−1, 1] 340M | 26.92 | 43.07 | 29.8 | 69.0 | 41.0 | 50.9 | 46.6 | 24.5 | 43.6 |
| | DeltaNet[−1, 1] 12 heads, 392M | 26.57 | 36.76 | 31.8 | 69.2 | 42.3 | 50.9 | 47.2 | 24.4 | 44.3 |
| | DeltaProduct$_2$[−1, 1] 392M | 26.43 | 30.66 | 34.0 | 68.9 | 42.4 | 53.1 | 48.9 | 25.9 | 45.5 |
| | DeltaProduct$_3$[−1, 1] 443M | 25.94 | **29.91** | **34.2** | 69.9 | 43.2 | 51.9 | 48.2 | 24.1 | 45.2 |
| | Gated DeltaNet[−1, 1] 340M | 25.97 | 33.57 | 33.1 | 69.5 | **44.1** | 51.1 | **50.9** | **26.7** | 45.9 |
| | Gated DeltaProduct$_2$[−1, 1] 393M | **25.12** | 30.03 | **34.2** | 69.1 | 44.6 | **55.3** | 49.8 | 25.3 | **46.4** |

### C.3.4 Training behavior

The training behavior of DeltaProduct$_{n_h}$ is stable as shown in Figure 15. This is also true for all considered model sizes in Figures 10 and 21 and Appendix C.3.5.

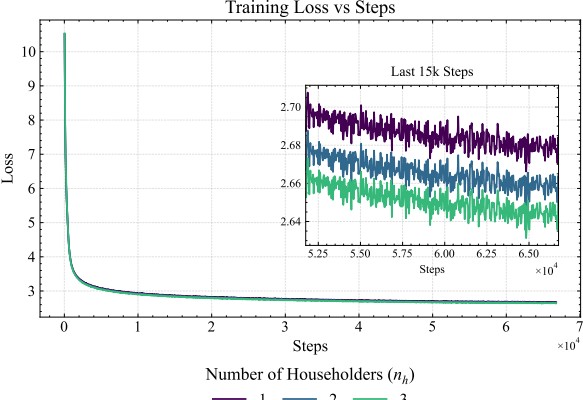

Figure 15: Training loss curves of DeltaProduct$_{n_h}[-1, 1]$. The curves demonstrate stable training behavior as $n_h$ increases, with higher values of $n_h$ consistently yielding lower losses throughout training and convergence. While the absolute differences in loss between different $n_h$ values are relatively small, they correspond to significant differences in length extrapolation performance.

### C.3.5 Additional results on Length Extrapolation

In this section we show additional plots on length extrapolation. In Figure 16 we show the length extrapolation behavior of (Gated) DeltaProduct$_{n_h}$ scaling up $n_h$ without adjusting any of the other model configuration parameters. As discussed in Section 5.3, increasing $n_h$ increases the parameter count of the model. Hence, Figures 17 and 18 show the per-token loss and perplexity of DeltaProduct$_{n_h}$ at three different scales where the parameter counts are matched at the respective scales following the configuration parameters shown in Table 6. Note that these are the same models as shown in Figure 21.

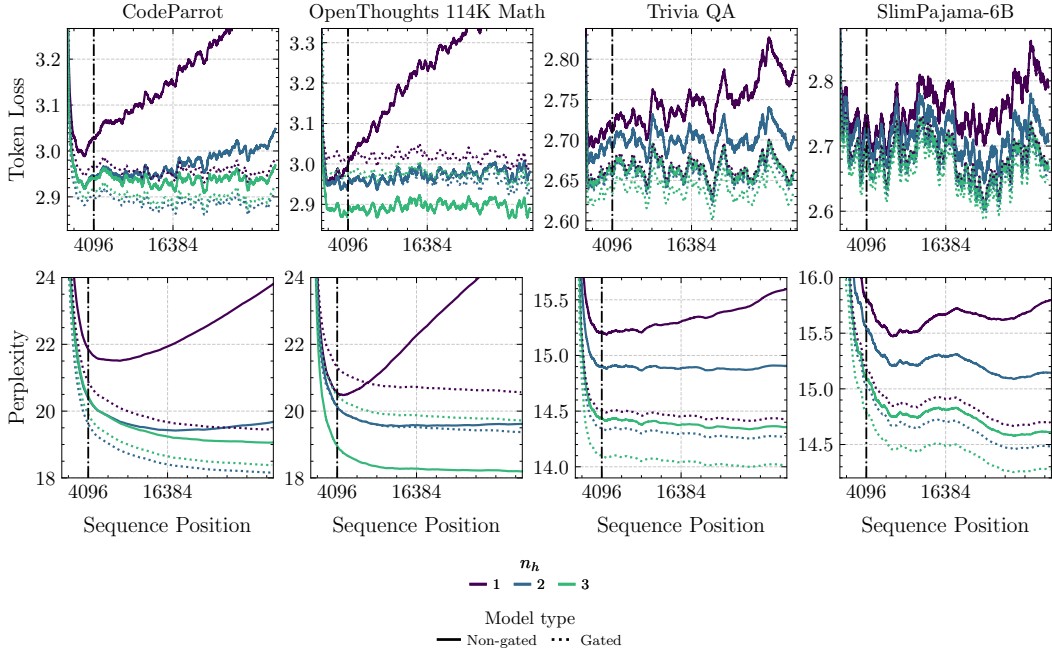

Figure 16: Per token loss and perplexity on context lengths up to 32.768 for (Gated) DeltaProduct$_{n_h}$. *(Top)* per token loss. *(Bottom)* Perplexity. Per token losses smoothed with a window-size of 300.

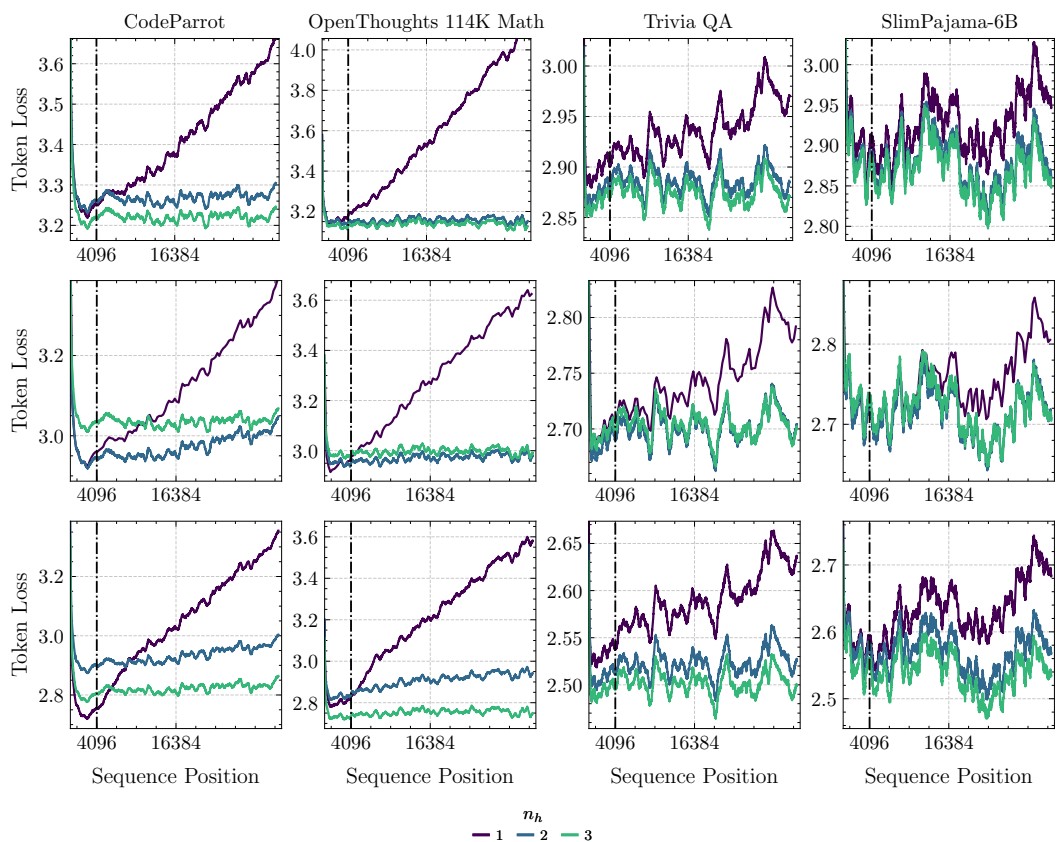

Figure 17: Per token loss of DeltaProduct$_{n_h}$ on contexts up to 32.768 at different model sizes. Models are parameter equivalent at each scale. Parameter equivalence is achieved by scaling the number of heads. Exact model configurations can be found in Table 6 *(Top)* 213M parameters. *(Middle)* 392M parameters. *(Bottom)* 805M parameters.

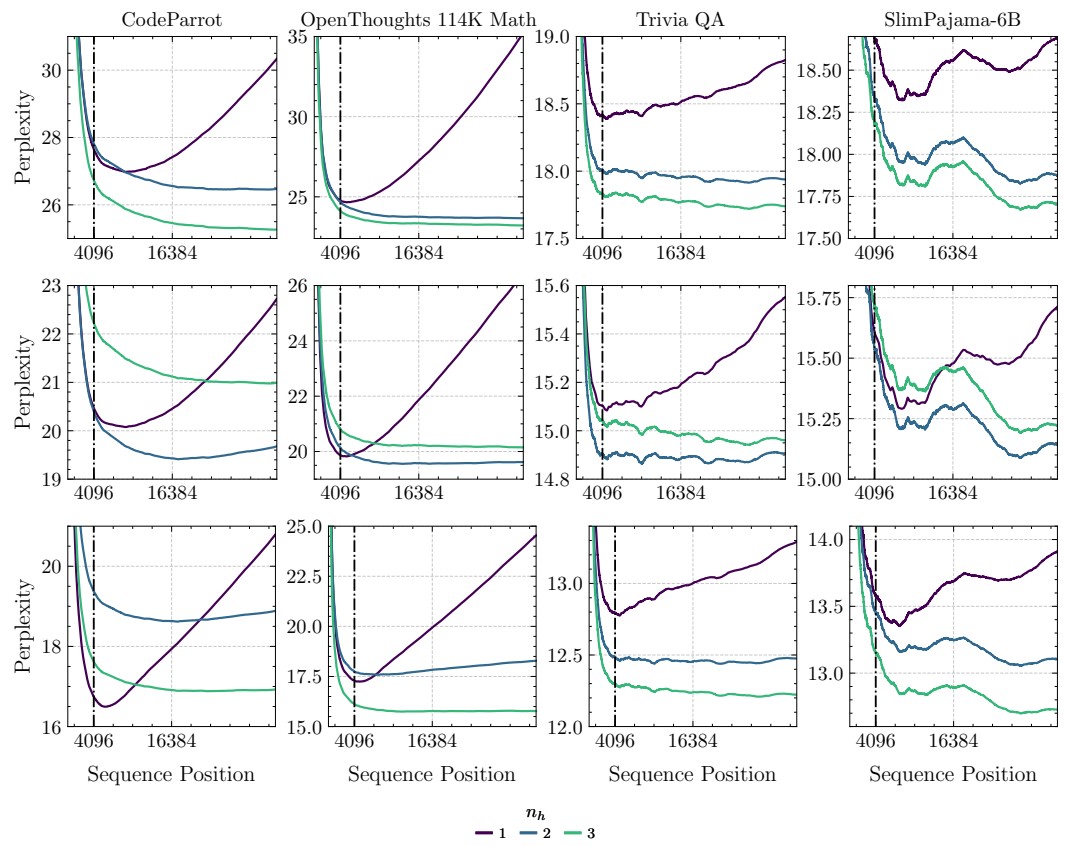

Figure 18: Perplexity analogue of Figure 17

Table 6: Model configuration parameters for models shown in Figures 17 and 18. All other configuration parameters are the same as in [16].

| Model Scale | # Householders | Hidden size | # Heads |
|---|---|---|---|
| **213M** | 1 | 768 | 8 |
|  | 2 | 736 | 6 |
|  | 3 | 768 | 4 |
| **392M** | 1 | 1024 | 12 |
|  | 2 | 1024 | 8 |
|  | 3 | 1024 | 6 |
| **805M** | 1 | 1536 | 16 |
|  | 2 | 1468 | 12 |
|  | 3 | 1536 | 8 |

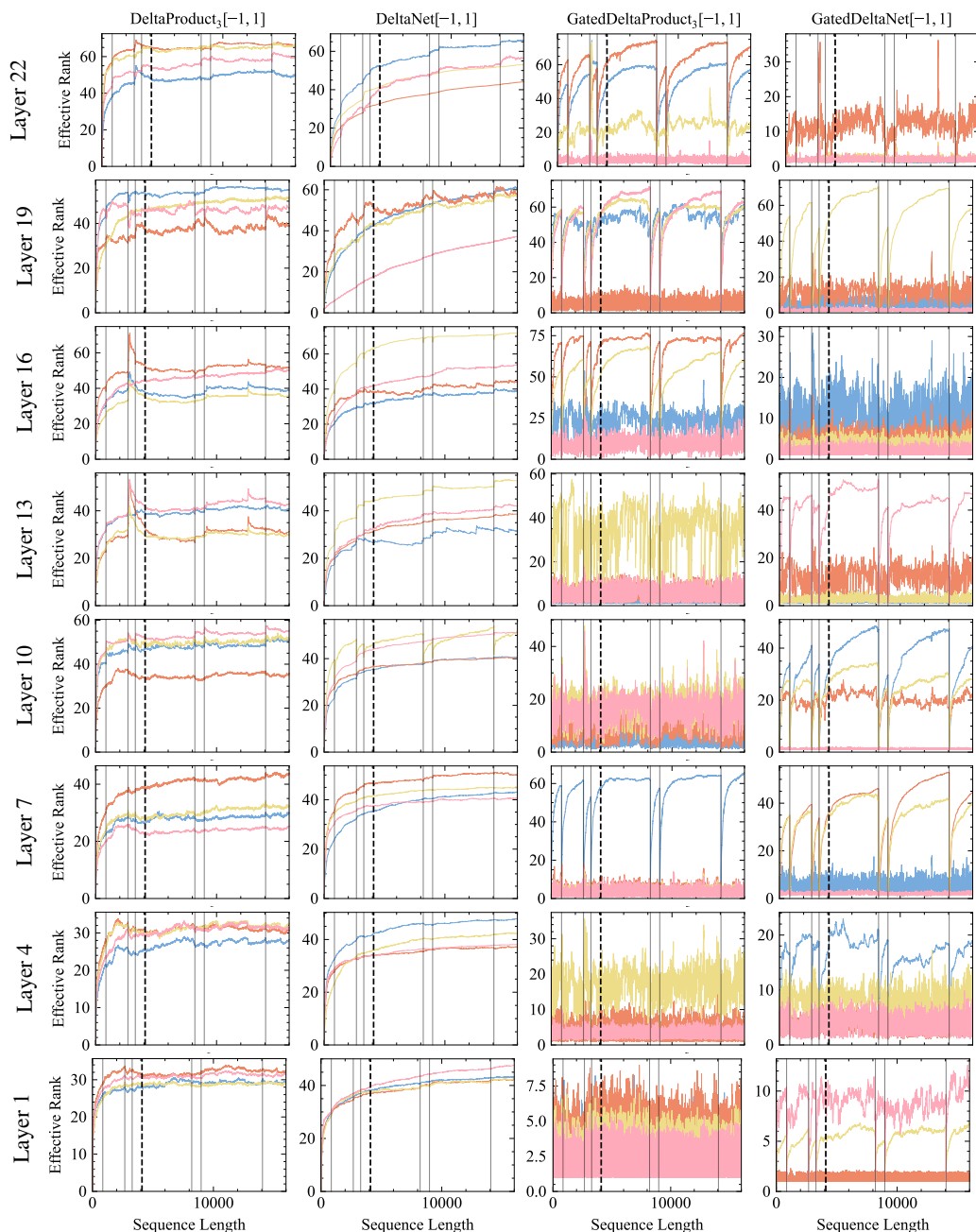

Figure 19: Effective rank of $\boldsymbol{H}_i$ for 4 of 8 heads for a selection of the layers on **CodeParrot** sequences. Solid vertical lines mark new code sequences; dashed vertical line indicates 4096-token training context length; colored lines show effective rank per head over the sequence.

### C.3.6 Additional Results on Scaling Behavior

In Figure 10 parameter equivalence is achieved at each scale mainly by decreasing the the head dimension for models with $n_h > 1$. In Figure 21 we show perplexity of FineWeb for another set of scaling results where parameter equivalence is reached by reducing the the number of heads in the attention. The result for this alternative type of scaling still shows the superiority of DeltaProduct compared to DeltaNet. However, in this case, models with higher $n_h$ are not strictly better than those with fewer Householders.

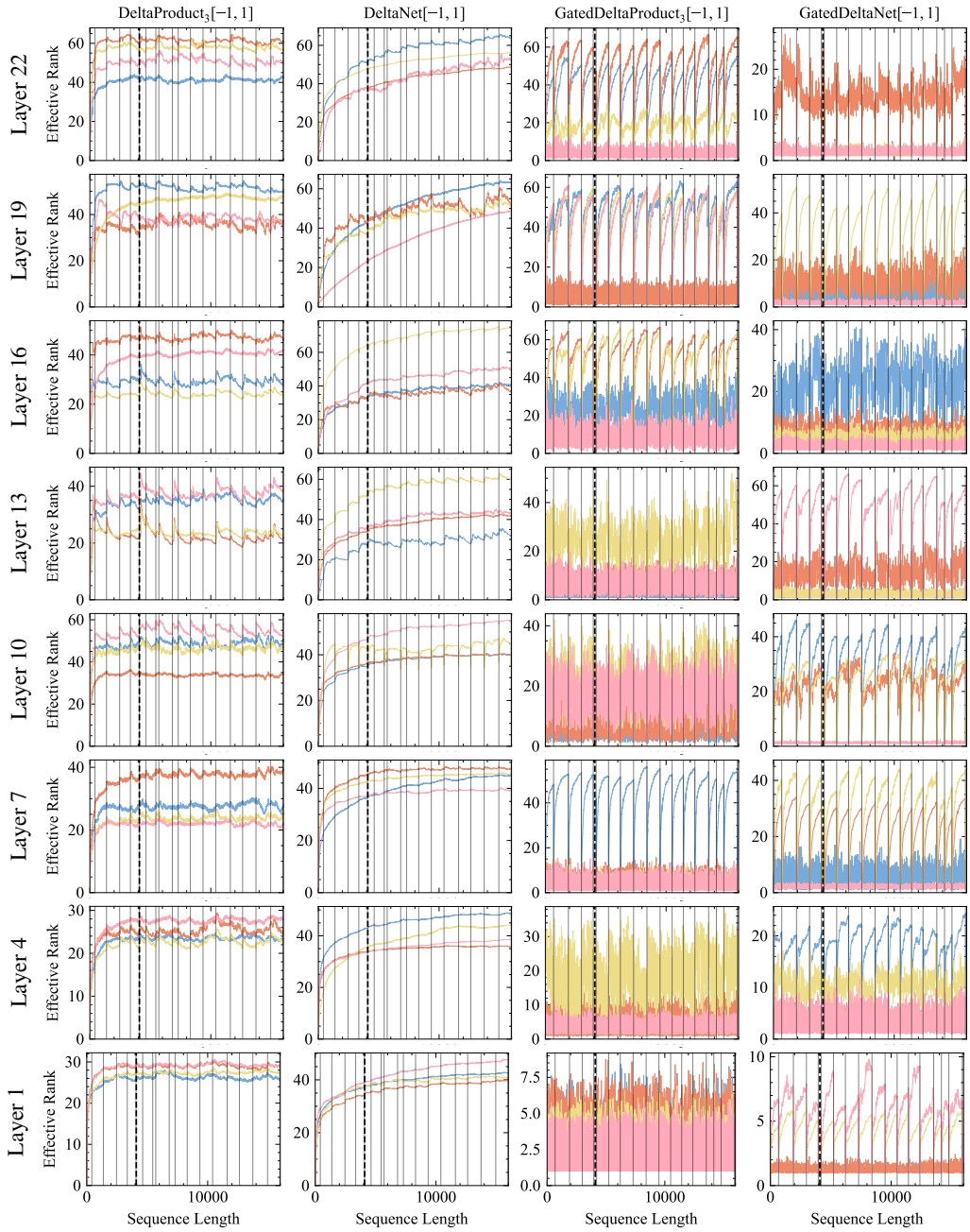

Figure 20: Effective rank of $H_i$ for 4 of 8 heads for a selection of the layers on **TriviaQA** sequences. Solid vertical lines mark new question-answer pairs; dashed vertical line indicates 4096-token training context length; colored lines show effective rank per head over the sequence.

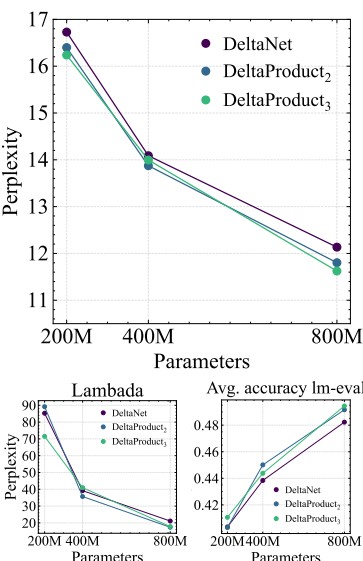

Figure 21: Scaling analysis w.r.t. (*top*) final perplexity on FineWeb, (*bottom*) Lambada and lm-eval tasks. Parameter equivalence is achieved by scaling the number of heads. Models trained at each scale with number of tokens as reported in Table 4.

