# OpenReview forum: "DeltaProduct: Improving State-Tracking in Linear RNNs via Householder Products"
_NeurIPS.cc/2025/Conference — NeurIPS 2025 poster_

### Official Review · Reviewer_bkha · 2025-06-27

**Clarity:** 3
**Significance:** 4
**Originality:** 4
**Rating:** 5
**Confidence:** 5

**Summary:**

This paper addresses the fundamental trade-off between expressiveness and computational efficiency in State Space Models (SSMs). While diagonal SSM blocks are computationally tractable but limited in expressive power, and unstructured transition matrices offer full expressiveness at prohibitive computational costs, the authors propose a novel middle-ground approach. Their solution involves taking several rank-one steps for each token, effectively using low-rank transition matrices with rank equal to the number of steps.

The paper provides theoretical results proving the well-posedness and expressive power of the new SSM on state tracking tasks, supported by extensive experiments comparing against state-of-the-art competing architectures.

**Questions:**

### Notation and Consistency Issues

- **Line 97**: The definition "yi = x1 · x2 · · · xi" is inconsistent with parts of the appendix where "yi = xi ... x1". For non-abelian products, this distinction is crucial. Please maintain consistency or clearly indicate when different conventions are used.

### Direct Parametrization Questions

- **Lines 184-188**: You have a couple of lines on direct parametrization of the low-rank structure. It is true that in the presented case (line 188) the matrix is restricted to be symmetric, but is there something blocking you from choosing the second ki to be different from the first? In that case there would not be any imposed symmetry. Is there some reason why you did not consider this?

## Appendix Issues

### Proposition 1

- **Line 1034 (CRITICAL)**: In computing D explicitly, you appear to have omitted the term -4det(A). Fortunately, since (2-b1-b2)² - 4(1-b1)(1-b2) = (b1-b2)², the conclusion still holds. With this correction, you could derive explicit bounds on cos(θ).

### Unclear Statements and Missing Information

- **Line 1064 (Pedantic)**: Can this really model ANY product of Householders, or do restrictions on operator norm apply? This may be inconsistent with later comments.
- **Line 1069**: "The assumption on H0" - which specific assumption are you referring to?
- **Line 1073**: In the RMS formula, the xi terms need to be squared.
- **Lines 1078-1082**: Please clarify this section as it's currently difficult to follow.
- **Line 1084**: "H_t^j = ei" - the indices i and j are not quantified. Are they related? Can they be any possible tuple?

### Theorem 3 Issues

- **Lines 1117-1119**: Please expand for clarity. This is where the definition of m is crucially used as $m \cdot n_h \geq n - 1$.

- **Lines 1120-1127 (CRITICAL)**: This section requires substantial clarification as it's a crucial part of the proof but appears hastily written:

    - Line 1120: What do you mean by "with enough information in xi about previous tokens"?
    - Lines 1121-1122: Please place this within your established framework (A, B, k, v, etc.). Why is x bold now? What are the matrices "P_{lm+i}" - these haven't been defined. Aren't the "G_{l,m}...G_{l,i+1}" functions on tokens coming after the current one? How is this information encoded in the tuple (Hi,xi)?
- **Line 1128**: The 'a' should not be bold here.
### Lemma 2 Concerns

- **Line 1141**: Use clearer set notation: write "2-d ≤ i ≤ 0" instead of "i ∈ {2-d, ..., 0}".
- **Line 1142**: Where does the bold 'a' come from? Is this the tuple (t mod d, a_{t-d+1}, ..., at)?
- **Line 1145**: There is no j>1 since nh = 1.
- **Line 1146 (Important)**: While claiming that the system can learn any function on the specified domain is technically correct, it seems somewhat disingenuous. What you actually prove is that your recurrence can memorize input up to a certain time, while learning is performed by a universal approximator in the decoder (e.g., MLP). Do you believe this occurs in practice, or can your system perform meaningful computation independently of decoder universality? This deserves discussion.

### Theorem 4 Issues

- **Line 1152**: Which specific isomorphism are you referring to?
- **Line 1153**: "that" appears to be a typo.
- **Lines 1159-1161**: Please cite this result - it's unlikely the intended audience is familiar with it.
- **Lines 1162-1164**: Why recover y only at the last step? Shouldn't it be recovered at all steps according to your state tracking problem definition?

### Lemma 3

- **Line 1198**: "exist" should be "exists".
- **Lines 1215-1229**: Same critical issues as found in Theorem 3.

### Theorem 6

- **Lines 1247-1248**: There are two (ii) labels and no (i); line 1256 should be (i).
- **Line 1252**: Consistency issue with "yi = x1 · x2 · · · xi" as discussed above.
- **Line 1265**: Is the angle clockwise or counterclockwise? To be pedantic the line passes from the origin to the POINT (1,0), this line in some sense is the vector (1,0) itself.
- **Lines 1268-1269**: Mismatch between j and i indices in the last two θ expressions.
- **Line 1270 (Important)**: When referencing "standard identities of products of 2D rotations and reflections," clarify that these are matrix representations of the group as given in equation (5). You should consider placing this paragraph (1270-1275) at the beginning for clarity, then introduce the system. In general I believe this theorem lacks a paragraph giving intuition, explaining what d_0 and c_0 are supposed to represent, etc.
- **Lines 1271-1272**: First line - d0 is not bold. Second line - need to prove "H(π/m)d0 = R(π/m)d0"; avoid ambiguous ± notation.
- **Lines 1273-1274**: "d0 = H(π/m)c0" appears true but unproven. "R(-2iπ/m)d0 = H(-2iπ/m)d0" is not proven.
- **Lines 1274-1275 (Critical Error)**: There appears to be an error in the last line. θ(sj, 0) - θ(s{m-i}, 0) = (-2m + 2(i+j))π/m since (2+2j)-(2+2(m-i)) = 2j - 2m + 2i = -2m + 2(i+j). The expression in R should be (-2m + 2(i+j))π/m instead of (2 + 2(i+j))π/m.

### Additional Issues

- **Line 1292 (Pedantic)**: "When this property is not satisfied, the norm of the state will diverge exponentially fast to infinity." Does this not depend on how fast the norms diverge?
- **Lines 1375-1377 and 1407-1411**: Using identical hyperparameters for different models without justification raises fairness concerns. How do you ensure fair comparison?


## Recommendation

This paper tackles an important problem with substantial theoretical and empirical evidence supporting it. The approach is well-motivated and the experimental work is comprehensive. However, the seeming mistakes in the main proofs (particularly in Proposition 1, where it is not a typo) and unclear arguments in Theorem 3 and Lemma 3 prevent acceptance of the paper in its current form.

**Path to Acceptance**: Please correct the identified errors in the proofs (or prove me wrong) and clarify the murky arguments in Theorems 3 and Lemma 3. With these corrections, I would raise my recommendation to at least a 5, as the underlying contribution is valuable and the work is otherwise very well-executed.

**Ethical Concerns:**

["NO or VERY MINOR ethics concerns only"]

**Final Justification:**

This paper tackles an important problem with substantial theoretical and empirical evidence supporting it. The approach is well-motivated and the experimental work is comprehensive. Mistakes pointed out before rebuttal have been addressed and corrected.

 As the contribution is valuable I raise my score and recommend acceptance.

**Limitations:**

yes

**Quality:**

4

**Strengths And Weaknesses:**

## Strengths

1. **Thorough Theoretical Foundation**: The paper provides comprehensive theoretical backing for the proposed approach, with formal proofs of well-posedness and expressive power.

2. **Extensive Experimental Validation**: The experimental section offers comprehensive comparisons with state-of-the-art architectures.

3. **Clear Problem Motivation**: The paper tackles an important and well-defined problem in the SSM literature with a principled approach.

4. **Transparency**: The authors are transparent about both the limitations and strengths of their approach.


## Major Weaknesses

1. **Errors in Main Proofs**: There are errors in key theoretical results that need correction, though these appear to be fixable without changing the main conclusions.

2. **Inconsistent Proof Quality**: Some proofs are hastily written and inconsistent with the rest of the paper, making them very difficult to follow and verify.

3. **Experimental Fairness Concerns**: The use of identical hyperparameters and training setups across different models without justification raises questions about fair comparison.

---

> ### Author Rebuttal · Authors · 2025-07-30
>
> We greatly thank the reviewer for the thoughtful and thorough response and apologize for the lack of clarity, typos and small mistakes in the theoretical sections. We modified our draft incorporating the reviewer feedback, which greatly improved the clarity of the paper. After careful checking and as the reviewer points out, none of our theoretical results will change. Due to space constraints, we have to be concise in our response, but will be happy to provide any clarification during the discussion period.
>
> ## Direct Parametrization Questions
> Lines 184-188: The issue discussed in that paragraph concerns bounding the norm of the recurrence. If we choose the second key to be different from the first we cannot in general guarantee the recurrence stability even when $n_h = 1$, as explained in Appendix B.5 for RWKV-7 which also has two different keys. Please also see our first response to Reviewer HNXg, where we expand on the expressivity of RWKV-7 products.
> ## Appendix Issues
> ### Proposition 1
> Thank you for identifying this error in Proposition 1.
> Our corrected proof shows that real eigenvalues are guaranteed if at least one $\beta_i$ is in $[0,1]$ by proving the discriminant $D = [\text{tr}(\mathbf{A}_S)]^2 - 4\det(\mathbf{A}_S) \ge 0$ where $\mathbf{A} = \mathbf{H}_2\mathbf{H}_1$ and $\mathbf{A}_S$ is the subspace spanned by the vectors $\mathbf{k}_1$, $\mathbf{k_2}$.
> - If one $\beta_i \in [0,1]$ and the other is in $(1,2]$, then $\det(\mathbf{A}_S) \le 0$, which ensures $D \ge 0$.
> - If both $\beta_1, \beta_2 \in [0,1]$, the AM-GM inequality shows $\text{tr}(\mathbf{A}_S) \ge 2\sqrt{\det(\mathbf{A}_S)}$, which implies $[\text{tr}(\mathbf{A}_S)]^2 \ge 4\det(\mathbf{A}_S)$ and thus $D \ge 0$.
> Hence complex eigenvalues only arise when both $\beta_1, \beta_2 > 1$.
>
> Following your suggestion, we computed explicit bounds on $\cos(\theta)$ for which complex eigenvalues occur:
> $$
> \frac{(\sqrt{\beta_1-1} - \sqrt{\beta_2-1})^2}{\beta_1\beta_2} < \cos^2\theta < \frac{(\sqrt{\beta_1-1} + \sqrt{\beta_2-1})^2}{\beta_1\beta_2}
> $$
> For the special case of two reflections ($\beta_1 =\beta_2 =2$), this simplifies to $0 < \cos^2\theta < 1$, which holds whenever the keys are linearly independent but not orthogonal.
>
> ### Unclear Statements and Missing Information
> Line 1078-1082: We agree that that section was difficult to follow. The general idea is that for the decoder to support our theoretical constructions, we need the map from all possible states of the RNN and the inputs to the decoder to be injective so that these can be distinguished by the MLP. There are two challenges to make this happen. The first is the dimensionality reduction via the scalar product with the query vector: the vector-valued state that we generally use would be compressed to a scalar after this operation. The second is the RMSNorm operation, which, in the case of a scalar input and $\epsilon = 0$ (in practice also if too small) gives a binary output, so no matter how many states our RNN has, we only have two different inputs to the MLP input. Since for our construction the number of states that we use is always finite, by setting the query as $\\textbf{b}/||\\textbf{b}||$ and $\epsilon > 0$ (In practice not too small) we can still guarantee injectivity despite these drawbacks. We’ll include a proof of this fact in the revised paper and we’ll be happy to clarify this further during the discussion period if needed.
> We also noticed a small mistake: $\\textbf{b}$ should be defined as $\\textbf{b} = (b, b^2, \\dots, b^d)^\\top$ with $b \geq \\delta_{\\max}/\\delta_{\\max} + 1$, $\\delta_{\\max} = \\max_{x,y \\in \\mathcal{S}} |x - y|$, $\\delta_{\\min} = \\min_{x,y \\in \\mathcal{S}, x \\neq y} |x - y| $ and $\\mathcal{S} \\subset \\mathbb{R}$ is the finite set of values of each element of the RNN state matrix.
>
> Line 1064: You are correct. The restriction on operator norm applies here. We’ll rephrase this in the updated version.
>
> Line 1069:  We will explicitly state which assumption: “Alternatively, the assumption on $ \\textbf{H}\_0 $ (task-dependent with rank at most $n\_h$)”.
>
> Line 1084: j is the index of the head, which is not related to the content of the hidden state which is $e_i$. We will write that $1 \\leq i \\leq d$ to make this clearer.
>
> Line 1073, Line 1128: Thank you, we corrected it.
> ### Theorem 3
> We agree with the reviewer that this section was not clear. We’ll expand and clarify it in the revised version. The main idea of the proof is to divide the tokens into blocks of $m$ matrices. The recurrence of the last layer computes the decomposition for the tokens in the previous blocks, while the current block is computed via the last layer decoder. We’ll be happy to provide further clarification and possibly the complete proof during the discussion period if needed.
>
> Line 1117-1119: We’ll rewrite as “We divide the input sequence into blocks of $m$ elements: we factorize the position $i \in \\{1,\dots t\\}$ into $ i = lm + \\tilde i$ where $l \\geq 0$ is the index of the previous block and $\\tilde{i} \in \\{1,\\dots, m\\}$ is the position within the current  block (index $l+1$).”
>
> Line 1120: We explain what we mean by “with enough information in xi about previous tokens” later in the text (we’ll clarify this): each token of the last layer should contain the position modulo 2m and the last 2$m$ elements of the input sequence: including both the (partial) current and previous blocks. $x$ is now bold because it is the input of the last layer, which can be a vector, as opposed to the input to the model. It should be $P_{x_{lm+i}}$, not $P_{lm+i}$, we’ll fix it. The matrices $G_{l,m}...G_{l,i+1}$ are not functions of the tokens coming after the current ones because they are part of the “previous block” of tokens. The current block of permutations is computed entirely in the decoder via lookup-table.
> ## Notation and Consistency Issues
> Line 97: Thanks for pointing this out, we will make the main paper consistent with the appendix by reverting the order, i.e. using $y_i = x_i \\cdot x_{i-1} \cdots x_1$.
> ### Lemma 2
> Line 1146: You are right to highlight the role of the decoder. Our theoretical constructions (as RWKV-7’s) rely on the universality of the MLP, whose required size grows with the complexity of the task (e.g., the group size). We’ll clarify this. The MLP complexity for complex groups with small values of $n_h$ is quite high since to learn $S_n$, the lookup table that it has to learn grows exponentially with $m=(n-1)/n_h$. Indeed, in our experiments (Figure 5 bottom row) the $S_5$ world problem cannot be learned when $n_h=1$ even with 10 layers. However it might be possible to learn the construction if $n_h$ is closer in value to $n$.
>
> Line 1141, 1142, 1145, 1153, 1198, 1247-1248, 1252, 1268-1269:  Thank you, we agree and we fixed the typos in our manuscript.
>
> Line 1265: The direction of the angle does not matter for the proof. We will change “vector” for “point” as suggested.
> ### Theorem 4
> Line 1152: We will change the word isomorphism to the word assumption. From the context, it will be clear that we are talking about the assumption of the group being isomorphic to the orthogonal or rotation groups.
>
> Line 1159-1161: The fact that we only need products of $n_h$ Householder matrices to represent all elements in $SO(n_h + 1)$ when $n_h$ is even, is a consequence of the Cartan-Dieudonné theorem, which we already mention since it explains DeltaProduct’s expressivity, and the fact that now we are focusing on orthogonal transformations with determinant +1. The determinant of the product of matrices equals the product of the determinants and each Householder has determinant -1. Hence, an even number of Householders cannot be in $SO(n_h + 1)$. We’ll add this clarification.
>
> Lines 1162-1164: The same reasoning can be applied at any step (not only the last one), we will change the index from t to i to highlight this.
> ### Lemma 3
> Line 1215-1229: The proof for this will follow the same structure as our clarified proof for Theorem 3, and we will refer to it to maintain conciseness.
> ### Theorem 6
> Line 1270: We have added an introductory part to the proof explaining the geometric intuition and defining the matrix representations of reflections and rotations upfront.
>
> Line 1271-1272: We’ll ensure that $d_0$ is always bold. The unproven equation can be directly derived from the definition $d_0 = (1,0)^T$ and the definitions of rotation and reflection matrices, given that the first column of a rotation matrix is identical to that of a reflection matrix (based on our notation).
>
> Lines 1273-1274: We do not use that equality but just use the definition $c_0 =\textbf{H}(\\pi/m)d_0$ here. See the above point for the proof of the second equation.
>
> Lines 1274-1275 (Critical Error): You’re right, that was a typo, your expression is correct. The $-2m$ term can be removed due to the periodicity of the rotation matrix, yielding the desired result $\textbf{R}(2(i+j))\pi/m)$. We’ll fix this in the revised version.
>
> ## Additional Issues
> Line 1292: Correct. We will revise to remove "exponentially fast".
>
> Lines 1375-1377 (Chomsky Hierarchy): In our experiments, we compared against the results by Grazzi et al. 2025 for DeltaNet, Mamba and mLSTM / sLSTM. Our hyperparameters were chosen based on the ones found to produce the best results on the state-tracking experiments.
>
> Lines 1407-1411 (Language Modeling): Due to compute budget constraints, we decided to use the hyperparameters used by Yang et al. 2024 for similar experiments.
>
>
> ## References:
> Grazzi, R., Siems, J., Zela, A., Franke, J. K., Hutter, F., & Pontil, M. Unlocking State-Tracking in Linear RNNs Through Negative Eigenvalues. ICLR. 2025
>
> Yang, S., Wang, B., Zhang, Y., Shen, Y., & Kim, Y. (2024). Parallelizing linear transformers with the delta rule over sequence length. Advances in neural information processing systems, 37, 115491-115522.

---

> > ### Comment · Reviewer_bkha · 2025-08-03
> >
> > Thank you for the extensive response, you seem to have addressed all the issues raised.
> > Before the final decision, it would be really helpful to have access to the revised proofs.
> > Would you be able to provide them in any manner? I'm thinking chiefly about Theorem 3.

---

> ### Author Response · Authors · 2025-08-03
>
> Given the constraints of the rebuttal, we think the only way to provide the proofs is in the comments. We would like to note that this requires re-formatting due to the absence of macros and the different syntax, which might produce typos.
> In the following comments we provide the proof and statement of Theorem 3, the proof of Proposition 1 case 3 and the first part of the Decoder implementation section. Please let us know if other parts would be helpful.
>
> ### Theorem 3.
>
> For any $n \in \mathbb{N}$ there exists a DeltaProduct model with one of the following configurations that can solve the word problem of the symmetric group $S_n$: (i) one layer with $n_h =n-1$  (ii) 3 layers with $n_h>1$ (iii) 4 layers with $n_h=1$.
>
> The construction for (ii) and (iii) requires that the MLP at the second last layer computes a lookup-table of size $2m \times (n!)^{2m}$, function of the last $2m$ input tokens and the position modulo $2m$ with $m = \lceil(n-1)/n_h\rceil$.
>
> ---
>
> **Proof.**
> One way to solve the group word problem for the symmetric group $S_n$ is to map each element of the group $g \in S_n$ to the corresponding permutation matrix $\mathbf{P}\_g \in \\{0,1\\}^n$ and then for each input sequence $x_1,\dots, x_t$ with $x_i \in S_n$ compute each element of the output sequence $y_1,\dots, y_t$ as
> $$
>     y_i = x_{i} \cdot x_{i-1} \cdots x_1 = \phi(\mathbf{P}\_{x_i}\cdots\mathbf{P}\_{x_1}\mathbf{u}\_0 ), \quad \mathbf{u}\_0 = (1,\dots, n)^\top,
> $$
> where $\phi$ is a surjective map from vectors in $\\{1,\dots, n\\}^n$ to the $n!$ elements of $S_n$, which we consider integers for simplicity, i.e., $x_i, y_i \in \\{1,\dots, n!\\}$.
>
> **(i).** Since a permutation of $n$ elements is a series of at most $n-1$ swaps of $2$ elements, if $n_h = n-1$, then we can solve the problem with a 1-layer DeltaProduct by setting $\mathbf{H}\_0 = \mathbf{u}\_0$, $\mathrm{dec}(\mathbf{H}\_i, x_i) =\phi(\mathbf{H}\_i)$, $\mathbf{B}(x_i) = 0$ ($\mathbf{v}\_{i,j} = 0$),  $\mathbf{A}(x_i) = \mathbf{P}\_{x_i}$. The latter is achieved by setting for the $j$-th element in the product $\prod_{j=1}^{n_h}(I - \beta_{i,j} \mathbf{k}\_{i,j}\mathbf{k}\_{i,j}^\top$), either $\beta_{i,j} = 2$ and $\mathbf{k}\_{i,j} = (\mathbf{e}\_k -\mathbf{e}\_p)/\sqrt{2}$ with $\mathbf{e}\_i$ being the $i$-th element of the canonical basis of $\mathbb{R}^n$ (swap element at index $k$ with the one at index $p$), or $\beta_{i,j} = 0$ (identity).
>
> **(ii) and (iii).** If $n_h < n-1$, then the state transition matrix is not sufficiently expressive to represent all permutations of $n$ elements. However, we can use additional layers to overcome this issue as follows. We divide the input sequence into blocks of $m$ elements: we factorize the position $i \in \\{1,\dots t\\}$ into $ i = lm + \tilde i$ where $l \geq 0$ is the index of the previous block and $\tilde{i} \in \\{1,\dots, m\\}$ is the position within the current block (index $l+1$). First, consider the case when $l \geq 1$. Let $\tilde{\mathbf{P}}\_l = \mathbf{P}\_{x_{(l-1)m+m}} \dots \mathbf{P}\_{x_{(l-1)m + 1}}$ be the product of the permutations of the previous block. Since $\tilde{\mathbf{P}}\_l$ is a permutation matrix of $n$ elements, we can factor it into $\tilde{\mathbf{P}}\_l = \mathbf{G}\_{l,m} \cdots \mathbf{G}\_{l,1}$ where we recall that $m =\lceil(n-1)/n_h\rceil$ and each of $\mathbf{G}\_{l,1}, \dots, \mathbf{G}\_{l,m}$ is a product of $n_h$ generalized householder matrices and thus can be a state-transition matrix of our model. We fix one factorization for each possible permutation matrix and we set $\tilde{\mathbf{P}}\_0 = \mathbf{G}\_{0,m}\cdots \mathbf{G}\_{0,1}$, with $\mathbf{G}\_{0,i} = I$ to handle the case when $l=0$.
>
> Now let $\mathbf{x}\_i$ be the input of the last layer. if $\mathbf{x}\_i$ contains enough information about previous tokens (as we specify later), we can set the recurrence and decoder of the last layer as
> $$
>     \mathbf{H}\_i = \mathbf{G}\_{l,\tilde i} \mathbf{H}\_{i-1}, \quad \mathrm{dec}(\mathbf{H}\_i, \mathbf{x}\_i) = \phi\Big(\underbrace{\mathbf{P}\_{x_i}\cdots \mathbf{P}\_{x_{lm + 1}}}\_{\text{current block}}  \underbrace{\mathbf{G}\_{l, m}\cdots \mathbf{G}\_{l,\tilde{i} + 1}}\_{\text{previous block}} \mathbf{H}\_i\Big).
> $$
> where $\mathbf{H}\_0 = \mathbf{u}\_0$, $\mathbf{B}(\mathbf{x}\_i) = 0$, $\mathbf{A}(\mathbf{x}\_i) = \mathbf{G}\_{l,\tilde i}$, using the construction at point (i) since $\mathbf{G}\_{l,\tilde i}$ is a product of at most $n_h$ Householers.
> Note that $\mathbf{H}\_i$ contains the product of the input permutations only up to token $x_{(l-1)m}$ and a partial computation of previous block of permutations $\tilde{\mathbf{P}}\_l$.
> Hence, the decoder completes the computation by applying two additional components: (1) the remaining transformations $\mathbf{G}\_{l,\tilde{i}+1}$ through $\mathbf{G}\_{l,m}$ needed to complete $\tilde{\mathbf{P}}\_l$, and (2) the actual permutations from the current partial block $\mathbf{P}\_{x_{lm + 1}}$ through $\mathbf{P}\_{x_i}$.

---

> > ### Author Response · Authors · 2025-08-03
> >
> > **Proof of Theorem 3 — Part 2**
> >
> > We can check that this ends up computing the correct output $y_i$ by substituting the expression for $\mathbf{H}\_i$ and unrolling the recurrence as follows.
> > $$
> > \begin{aligned}
> >     \mathrm{dec}(\mathbf{H}\_i, \mathbf{x}\_i) &= \phi(\mathbf{P}\_{x_{lm + \tilde{i}}}\cdots \mathbf{P}\_{x_{lm + 1}} \mathbf{G}\_{l, m}\cdots \mathbf{G}\_{l,1} \mathbf{G}\_{l-1, m}\cdots \mathbf{G}\_{l-1,1} \dots  \mathbf{G}\_{0,m} \dots \mathbf{G}\_{0,1} \mathbf{H}\_0). \\\\
> >     &= \phi(\mathbf{P}\_{x_{lm + \tilde{i}}}\cdots \mathbf{P}\_{x_{lm + 1}} \tilde{\mathbf{P}}\_l \tilde{\mathbf{P}}\_{l-1} \dots \tilde{\mathbf{P}}\_{0} \mathbf{u}\_0). \\\\
> >     &= \phi(\mathbf{P}\_{x_{lm + \tilde{i}}}\cdots \mathbf{P}\_{x_{lm + 1}} \mathbf{P}\_{x_{(l-1)m +m}}\cdots \mathbf{P}\_{x_{(l-1)m + 1}} \dots \mathbf{P}\_{x_{m}}\cdots \mathbf{P}\_{x_{1}} \tilde{\mathbf{P}}\_{0} \mathbf{u}\_0). \\\\
> >     &= \phi(\mathbf{P}\_{x_{i}}\cdots \mathbf{P}\_{x_{1}}\mathbf{u}\_0) = y_i,
> > \end{aligned}
> > $$
> > Note that to compute $\mathbf{A}(\mathbf{x}\_i) = \mathbf{G}\_{l,\tilde i}$ and $\mathrm{dec}(\mathbf{H}\_i, \mathbf{x}\_i)$, $\mathbf{x}\_i$ should contain $\tilde{i} = i \bmod m$ and the last $m + \tilde{i}$ (in general the last $2m$) tokens, corresponding to the current and previous block. Hence, the layers before the last one are dedicated to compute at each time-step $i$ a lookup table for the possible values of $(i \bmod 2m, x_i, \dots, x_{i-2m+1})$ whose output will be included in the input of the last layer $\mathbf{x}\_i$. The first layers (two layers if $n_h=1$, one if $n_h>1$) can provide $i \bmod 2m$ by using Lemma 2 with $d=2m$. Finally, the second to last layer can output any function of the last $2m$ tokens and the position modulo $2m$ through Lemma 3 with $d=2m$ and $a_t = x_t$, by using $i \bmod 2m$ from the first layer(s).
> >
> > **End of Proof of Theorem 3**
> >
> > ### Proposition 1
> > Proof of Case 3: **Complex Spectrum via Non-orthogonal Directions and Real Subcase**
> >
> > Let $\mathbf{k}_1,\mathbf{k}_2$ span a 2D subspace $S \subset \mathbb{R}^n$ with $\cos\theta = \mathbf{k}_1^\top\mathbf{k}_2$ such that $0 < |\cos\theta| < 1$. The product $\mathbf{A} = \mathbf{H}_2\mathbf{H}_1$ acts as the identity on $S^\perp$ (preserving $n-2$ eigenvalues at 1) and non-trivially on $S$. The restriction $\mathbf{A}_S$ of $\mathbf{A}$ to $S$ has trace $\text{tr}(\mathbf{A}_S) = 2 - \beta_1 - \beta_2 + \beta_1\beta_2\cos^2\theta$ and determinant $\det(\mathbf{A}_S) = (1-\beta_1)(1-\beta_2)$. The discriminant of its characteristic equation is $D = [\text{tr}(\mathbf{A}_S)]^2 - 4\det(\mathbf{A}_S)$. Complex eigenvalues arise if $D < 0$.
> >
> > To find the explicit bounds, we expand the inequality $D < 0$:
> > $$
> > (2 - \beta_1 - \beta_2 + \beta_1\beta_2\cos^2\theta)^2 - 4(1-\beta_1)(1-\beta_2) < 0
> > $$
> > Rearranging this expression as a quadratic in $x = \cos^2\theta$ yields:
> > $$
> > (\beta_1\beta_2)^2 x^2 + 2(2 - \beta_1 - \beta_2)\beta_1\beta_2 x + (\beta_1 - \beta_2)^2 < 0
> > $$
> > This inequality holds if and only if $x=\cos^2\theta$ lies strictly between the two roots of the corresponding equation. Solving for the roots gives the explicit bounds for complex eigenvalues:
> > $$
> > \frac{(\sqrt{\beta_1-1} - \sqrt{\beta_2-1})^2}{\beta_1\beta_2} < \cos^2\theta < \frac{(\sqrt{\beta_1-1} + \sqrt{\beta_2-1})^2}{\beta_1\beta_2}
> > $$
> > This inequality is only satisfiable when $\beta_1, \beta_2 \in (1, 2]$. For the special case of two standard reflections where $\beta_1=\beta_2=2$, the condition simplifies to $0 < \cos^2\theta < 1$, confirming that the product of any two distinct reflections is a rotation.
> >
> > Conversely, we show that if at least one coefficient $\beta_i \in [0,1]$, the eigenvalues are real as $D \ge 0$. We analyze this by cases:
> >
> > **Case 1: One coefficient is in $[0,1]$, the other is in $(1,2]$.** Without loss of generality, let $\beta_1 \in [0,1]$ and $\beta_2 \in (1,2]$. This implies $(1-\beta_1) \ge 0$ and $(1-\beta_2) < 0$, so their product $\det(\mathbf{A}_S) \le 0$. The term $-4\det(\mathbf{A}_S)$ is therefore non-negative. Since $[\text{tr}(\mathbf{A}_S)]^2 \ge 0$, their sum $D$ must be non-negative.
> >
> > **Case 2: Both coefficients are in $[0,1]$.** By the AM-GM inequality on the non-negative terms $(1-\beta_1)$ and $(1-\beta_2)$, we have $(1-\beta_1) + (1-\beta_2) \ge 2\sqrt{(1-\beta_1)(1-\beta_2)} = 2\sqrt{\det(\mathbf{A}_S)}$. Since $\text{tr}(\mathbf{A}_S)$ includes an additional non-negative term $\beta_1\beta_2\cos^2\theta$, it also holds that $\text{tr}(\mathbf{A}_S) \ge 2\sqrt{\det(\mathbf{A}_S)}$. Squaring both sides gives $[\text{tr}(\mathbf{A}_S)]^2 \ge 4\det(\mathbf{A}_S)$, ensuring $D \ge 0$.
> >
> > This analysis confirms that complex eigenvalues, which enable rotations, can only arise if and only if both $\beta_1 > 1$ and $\beta_2 > 1$. When at least one $\beta_i \le 1$, real eigenvalues restrict the transformations in that subspace to scaling or reflection.

---

> > > ### Author Response · Authors · 2025-08-03
> > >
> > > ### Decoder implementation.
> > >
> > >  In our implementation, $\mathrm{dec}$ is the same as in Gated DeltaNet:
> > > $$
> > >     \mathrm{dec}((\mathbf{H}^1_t,\dots \mathbf{H}\_t^H),\mathbf{x}\_t) = \mathrm{MLP}(\mathrm{RMSnorm}(\mathbf{x}\_t + \mathbf{o}\_t)), \\\\ \quad \mathbf{o}\_t = \sum_{j=1}^H\mathbf{W}^j_o\mathrm{RMSnorm}( (\mathbf{H}^j_t)^\top
> > >     \mathbf{q}^j_t), \quad \mathbf{q}^j_t = \psi(\mathbf{W}^j_q \mathbf{x}\_t)/||\psi(\mathbf{W}^j_q \mathbf{x}\_t)||
> > > $$
> > > where $\mathbf{W}^j_o, \mathbf{W}^j_q$ are two learned matrices, $\psi = \mathrm{SiLU}$, $\mathrm{RMSnorm}(\mathbf{x}) =
> > > \mathbf{a} \odot \mathbf{x}/\sqrt{\epsilon + d^{-1}\sum_{i=1}^d x_i^2}$ corresponds (when $\epsilon=0$) to a projection onto an axis aligned ellipse where $\mathbf{a} \in \mathbb{R}^d$ is learnable and determines the lengths of the axis and $\epsilon>0$ is a small value set to avoid numerical errors.
> > > Meanwhile, $\mathrm{MLP}$ is a two layer MLP. This structure can be limiting. For instance, when $\mathbf{H}\_t^j \in \mathbb{R}^{d}$, then  $b_t = (\mathbf{H}^j_t)^\top \mathbf{q}\_t \in \mathbb{R}$ and if $b_t >> \epsilon$, then $|\mathrm{RMSNorm}(b_t) | \approx 1$, which means that after RMSNorm we are left with effectively only 2 possible values, while our constructions might require more: as many as the number of possible states. Indeed, for all our constructions to work in practice, we would need a sufficiently large $\epsilon$, so that the output of $\mathrm{RMSnorm}$ can retain some magnitude information.
> > > Since in our constructions the number of states is finite, with $\epsilon > 0$ and an appropriate value for  $\mathbf{q}\_t^j$ we are guaranteed  that the map  $\mathbf{H}^j_t \mapsto \mathrm{RMSnorm}((\mathbf{H}^j_t)^\top\mathbf{q}^j_t)$, and consequently (by appropriately setting $\mathbf{W}\_o^j$) the map from states and inputs $\mathbf{x}\_t$, to the input of the MLP, is injective, which we show in the next lemma. Hence, thanks to the MLP, the decoder can approximate any continuous function of $(\mathbf{H}\_t, \mathbf{x}\_t)$ even after the bottlenecks caused by the scalar product with $\mathbf{q}\_t^j$ and the RMSnorm.
> > >
> > > **Lemma 1.**
> > > Let $\mathcal{S} \subset \mathbb{R}$ be a finite set of values. Define $\delta_{\min} = \min_{x,y \in \mathcal{S}, x \neq y} |x - y|$ and $\delta_{\max} = \max_{x,y \in \mathcal{S}} |x - y|$. Let $\mathbf{b} = (b, b^2, \dots, b^d)^\top$ and set $\mathbf{q} = \mathbf{b} / ||\mathbf{b}||$. If $b$ satisfies $b \ge \frac{\delta_{\max}}{\delta_{\min}} + 1$, then the mapping $f: \mathcal{S}^d \to \mathbb{R}$ given by $f(\mathbf{H}) = \mathrm{RMSnorm}(\mathbf{H}^\top \mathbf{q})$ is injective.
> > >
> > > **Proof.**
> > > The mapping $f$ is a composition $f = g_3 \circ g_2 \circ g_1$, where the component functions are:
> > > $$ g_1(\mathbf{H}) = \sum_{i=1}^d h_i b^i, \qquad g_2(x) = \frac{x}{||\mathbf{b}||}, \qquad g_3(x) = \mathrm{RMSnorm}(x), $$
> > > where $\mathbf{H} =  (h_1,\dots, h_d)^\top$.
> > > The overall mapping is injective if each component is injective.
> > >
> > > **1. Injectivity of $g_1$:** Intuitively, $g_1$ encodes the vector $\mathbf{H}$ as a scalar as a number in base $b$, where each $h_i$ acts as a ``digit'' drawn from the finite set $\mathcal{S}$.
> > > By choosing $b$ sufficiently large relative to the spread of $\mathcal{S}$ (captured by $\delta_{\max}/\delta_{\min}$), we ensure that different vectors produce distinct scalars.
> > > To prove this formally, we show that for any non-zero difference vector $\boldsymbol{\Delta} = (\Delta_1,\dots, \Delta_d)^\top = \mathbf{H}\_1 - \mathbf{H}\_2$, the difference $ g_1(\mathbf{H}\_1) - g_1(\mathbf{H}\_2) = \sum_{i=1}^d \Delta_i b^i$ is also non-zero. Let $k = \max\\{i \mid \Delta_i \neq 0\\}$. The magnitude of the highest-order term is lower-bounded by:
> > > $$ |\Delta_k b^k| = |\Delta_k| b^k \ge \delta_{\min} b^k $$
> > > The magnitude of the sum of lower-order terms is upper-bounded as
> > > $$ \left|\sum_{i=1}^{k-1} \Delta_i b^i\right| \le \sum_{i=1}^{k-1} |\Delta_i| b^i \le \sum_{i=1}^{k-1} \delta_{\max} b^i = \delta_{\max} \left(\frac{b^k-b}{b-1}\right) \leq \delta_{\min}(b^k-b), $$
> > > where we used the triangle inequality and the assumption on $b$, which implies $\delta_{\max} \le \delta_{\min}(b-1)$.
> > > Comparing the bounds, we see that
> > > $$ |\Delta_k b^k| \ge \delta_{\min} b^k > \delta_{\min}(b^k-b) \ge \left|\sum_{i=1}^{k-1} \Delta_i b^i\right| $$
> > > Since the magnitude of the highest-order term is strictly greater than that of the sum of all other terms, their sum cannot be zero. Thus, $g_1$ is injective.
> > >
> > > **2. Injectivity of $g_2$ and $g_3$:** The function $g_2$ is a linear scaling by the non-zero constant $1/||\mathbf{b}||$ and is therefore injective. For $g_3(x) = \mathrm{RMSnorm}(x)$, its derivative is strictly positive for $\epsilon > 0$, meaning $g_3$ is strictly monotonic and also injective.
> > >
> > > Since $g_1$, $g_2$, and $g_3$ are all injective, their composition $f$ is also injective.

---

> > > > ### Comment · Reviewer_bkha · 2025-08-03
> > > >
> > > > ## Theorem 3.
> > > >
> > > > ### i)
> > > >
> > > > For how I understand it you are trying to show that a system of type defined in sec 4 lines 156-162 can be constructed such that $\mathbf{A}(x_i) = P_{x_i}$ . The strategy is to write the permutation $x_i$ as a sequence of transpositions $x_i= z_{m(x_{i})}(x_{i}) ... z_{1}(x_{i})$ (order matters here)
> > > > each of which can be written in matrix form as $z_{j}(x_{i}) = I + \kappa_{j}(x_{i})  \kappa_{j}(x_{i})^\top$ for vectors $\kappa_{j}(x_{i}) $ of the type $(e_k(x_{i},j) - e_p(x_{i},j))/\sqrt{2}$ (here I am making all dependencies explicit).
> > > >
> > > > What I'm missing here is how you choose the right $e_k(x_i,j)$ and $e_p(x_i,j)$ with a map of type $\psi(W_j x_i) / ||\psi(W_j x_i)||$.
> > > > I see a way of doing it by fixing the sequence of transpositions a-priori, sending $x_i$ in a space in spirit equal to $(\mathbb{R}^{ n \choose{2}})^{n-1}$ in order to encode this ordering of transpositions and then using the $W_j$ to choose the $j$th transposition and encode it, with $\psi$ the identity.
> > > > Is this similar to what you had in mind?
> > > > If I stand correctly this construction is assumed for all the parts of the proof.
> > > >
> > > > ### ii and ii)
> > > > I believe that for a first time reader it could be useful to have a comment giving intuition on why there is a "delay" in the construction of $H_i = H_{lm + \tilde{i}}$. By this I mean that $H_i$ is constructed from $H_{i-1}$ by using $G_{l, \tilde{i}}$ which is a component of the "previous" block.
> > > >
> > > >
> > > >
> > > >
> > > > ## Lemma 1.
> > > >
> > > > 1. Once you have $\Delta$ you could simply conclude by Lagrange's bounds on polynomial roots (https://en.wikipedia.org/wiki/Geometrical_properties_of_polynomial_roots), which implies your result.

---

> > > > > ### Author Response · Authors · 2025-08-04
> > > > >
> > > > > Thank you again for taking the time to check the details of the updated theorems.
> > > > >
> > > > > ### Theorem 3
> > > > > i) That is correct, if for example $x_i \in  \\{0,1\\}^{ \\binom{n}{2} + 1 }$ is a one-hot vector (where the dimension accounts for all possible swaps of $n$ elements and the identity) then it can select the appropriate columns of $W_1, \dots, W_{n_h}$ that will results in the right decomposition for the permutation $\mathbf{P}\_{x_i}$. We also note that the construction holds not only if $\psi$ is the identity, but also when each component of the  image of $\psi$ (which is applied elementwise) includes an interval that contains zero in its interior, since this implies that by setting appropriately $W_j$,  $\mathbf{k}_{i,j} = \psi(W_j x_i) / || \psi(W_j x_i)|| $ can be  any point on the sphere of unit radius. This is the case for the SiLU which is used in our experiments.
> > > > >
> > > > > ii and iii) Thank you for the suggestion. Right above “We can check that this ends up computing the correct output ” we will add the comment “The  delay in the recurrence is necessary, since to compute even the first matrix of the factorization for a block of $m$ elements of the input sequence, all the elements in such a block need to be processed.”
> > > > >
> > > > > ### Lemma
> > > > > Thanks for the pointer to Cauchy upper bound for polynomial roots. We will add it to the lemma.

---

### Official Review · Reviewer_RGKp · 2025-07-03

**Clarity:** 4
**Significance:** 4
**Originality:** 4
**Rating:** 5
**Confidence:** 5

**Summary:**

DeltaProduct extends DeltaNet by performing rank-n updates instead of rank-1, yielding state-transition matrices as products of generalized Householder transformations. This creates a principled mechanism to interpolate between diagonal matrices (efficient) and more expressive transformations while maintaining stability through an operator norm less than 1. The authors prove that DeltaProduct can solve symmetric group word problems with improved trade-offs and that Gated DeltaProduct recognizes any regular language with finite layers.

Experimentally, DeltaProduct demonstrates superior performance on state-tracking tasks involving permutation groups and achieves dramatically improved length extrapolation in language modeling compared to DeltaNet, with minimal performance degradation on sequences beyond training context. The effective rank analysis reveals that multiple Householder products enable more efficient state forgetting mechanisms. The work provides both theoretical insights and practical improvements in synthetic reasoning and real-world language modeling tasks.

**Questions:**

I'm curious about a FLOPs comparison in the scaling analysis in Figure 10. To match parameters, the number of heads was reduced, but how do these variants compare w.r.t. FLOPs?

Did DeltaProduct display any training instabilities?

In Figure 5 bottom 2nd and 4th plots, an increased number of layers seems to hurt performance on the state tracking tasks. You addressed it, but isn't this unexpected? Is there an explanation? Also, the number of layers is not consistent in that row, which reduces the readability of the figure as a whole.

In Figure 9, the Gated DeltaNet effective ranks are extremely jagged. Why is this?

**Ethical Concerns:**

["NO or VERY MINOR ethics concerns only"]

**Final Justification:**

The work is convincing, and after seeing the other review,s I'm even more certain about it. This work should be accepted.

**Limitations:**

yes

**Paper Formatting Concerns:**

line 42 should be an em-dash (https://en.wikipedia.org/wiki/Dash)

**Quality:**

4

**Strengths And Weaknesses:**

This paper demonstrates exceptional scientific rigor with a clear, meaningful contribution that advances linear RNN research. The theoretical foundation is mathematically sound yet accessible through intuitive geometric explanations. The experimental validation is comprehensive, effectively combining synthetic state-tracking tasks with realistic language modeling benchmarks at an appropriate scale. The work exemplifies proper incremental research, extending DeltaNet in a principled direction that yields genuine improvements. The related work section is particularly thorough and well-organized, while the authors' commitment to reproducibility strengthens the work's impact. I consider the presentation quality to be outstanding throughout.

---

> ### Author Rebuttal · Authors · 2025-07-31
>
> We thank the reviewer for the positive feedback.
>
> > I'm curious about a FLOPs comparison in the scaling analysis in Figure 10. To match parameters, the number of heads was reduced, but how do these variants compare w.r.t. FLOPs?
>
> This is a great question. The FLOPs of the recurrence scale linearly in $n_h$ such that the complexity is $O(n_h \times (LCd + Ld^2))$ (Yang et al., 2024), where $L$ is the sequence length, $C$ is the chunk size and $d$ is the dimension of keys and values. The recurrence computation also requires $n_h \times L/C$ sequential steps. However, in our setup the total FLOPs of the model are dominated by the large matrix multiplications in the input/output projections and the MLP, rather than the recurrence itself, and so, the contribution of the recurrence to total model FLOPs is minor.
>
> In our parameter-matched scaling experiments (Figure 10 and Table 4), we adjusted the number of heads to equalize parameters. For instance, at the ~800M scale, we compare DeltaNet (16 heads) with DeltaProduct$_2$ (12 heads). The increase in FLOPs from the additional key/value projections and the doubled recurrence in DeltaProduct$_2$ is largely offset by the reduction in FLOPs from having smaller query, key, and value projections due to the fewer number of heads. Our results in Figure 13 in Appendix C.4 also support this. There we can see that for non-parameter matched models, the training throughput scales sub-linearly in $n_h$. We conducted additional throughput experiments with an improved model implementation that fuses the projection layers and causal 1d-convolutions. Here we find that DeltaProduct$_3$ with 805M parameters achieves 86% (compared to 75% without fusing projections and causal 1d-convolutions) of the training throughput of DeltaNet with 805M parameters at 4096 sequence length and a per-device batch size of 12 (on a NVIDIA H100-94GB). Moreover, we expect this to further improve with an optimized custom triton kernel (we are using DeltaNet’s).
>
> > Did DeltaProduct display any training instabilities?
>
> No, we did not encounter any training instabilities with (Gated) DeltaProduct. We attribute this to the inherent stability of the recurrence, as the operator norm of the transition matrix is bounded by 1, which keeps the gradients well-behaved. The training loss curves for $n_h=1,2,3$ in Figure 14 of the supplementary material demonstrate this stability empirically.
>
> > In Figure 5 bottom 2nd and 4th plots, an increased number of layers seems to hurt performance on the state tracking tasks. You addressed it, but isn't this unexpected? Is there an explanation? Also, the number of layers is not consistent in that row, which reduces the readability of the figure as a whole.
>
> Indeed, increasing layers from 8 to 10 reduces performance for the $S_4$ and $S_5$ groups.  However, we see that these experiments are quite challenging and require multiple seeds to find a run that could fit the training context length of 128 permutations and show extrapolation to 512 permutations: we use 5 seeds per configuration yet sometimes still couldn't fit the context length. This observation reinforces our claim: increasing expressivity horizontally via $n_h$ offers a more parameter efficient and stable optimization path for these state-tracking tasks than increasing it vertically via depth. We thank you for pointing out the inconsistent legend labels in the colormap; we will fix the colormap in the revised figure to improve readability.
>
> > In Figure 9, the Gated DeltaNet effective ranks are extremely jagged. Why is this?
>
> Both Gated DeltaNet and Gated DeltaProduct produce very jagged effective ranks. We found that both models learned to set the forget gate close to zero at the beginning of sequence tokens, effectively erasing the memory for the new sequence. Since the forget gate is a scalar for each hidden state per head the reset is very drastic.
>
> > line 42 should be an em-dash (https://en.wikipedia.org/wiki/Dash)
>
> Thank you for pointing this out, we have fixed it in our manuscript.
>
> Thank you again for helping us improve the paper.
>
> ### References
> S. Yang, B. Wang, Y. Zhang, Y. Shen, and Y. Kim. Parallelizing Linear Transformers with the
> Delta Rule over Sequence Length. In A. Globerson, L. Mackey, D. Belgrave, A. Fan, U. Pa-
> quet, J. Tomczak, and C. Zhang, editors, Proceedings of the 37th International Conference on
> Advances in Neural Information Processing Systems (NeurIPS’24), 2024

---

> > ### Comment · Reviewer_RGKp · 2025-08-04
> > **Response**
> >
> > Thank you for your response. I'm looking forward to seeing this paper presented at the conference.

---

### Official Review · Reviewer_HNXg · 2025-07-03

**Clarity:** 2
**Significance:** 2
**Originality:** 2
**Rating:** 5
**Confidence:** 3

**Summary:**

This paper introduces DeltaProduct, a novel linear Recurrent Neural Network (RNN) architecture that improves upon existing models like DeltaNet by enhancing state-tracking and expressivity. While DeltaNet's recurrence is interpreted as a single online gradient descent step per token, DeltaProduct performs multiple (nh) steps, resulting in a more expressive state-transition matrix formed by the product of nh generalized Householder transformations. This creates a diagonal plus rank-nh matrix structure, providing a tunable mechanism to balance expressivity with computational efficiency. The authors provide theoretical analysis showing that increasing nh improves the model's capacity for complex state-tracking, such as solving group word problems. Through extensive experiments, they demonstrate that DeltaProduct outperforms DeltaNet in both state-tracking and language modeling tasks, and exhibits significantly improved length extrapolation capabilities.

**Questions:**

None.

**Ethical Concerns:**

["NO or VERY MINOR ethics concerns only"]

**Final Justification:**

The authors' rebuttal, particularly the new experiments and the generalization of their framework, has addressed my initial concerns. The planned revisions will substantially improve the paper's quality and significance. Therefore, I have updated my recommendation.

**Limitations:**

Yes.

**Paper Formatting Concerns:**

None.

**Quality:**

2

**Strengths And Weaknesses:**

Strengths :-

- The paper constructs complex, non-symmetric state-transition matrices by composing products of simple, symmetric Householder reflections. This allows the model to learn rotational and oscillatory dynamics, crucial for advanced state-tracking, while mathematically guaranteeing stability since the operator norm remains bounded.
- The paper demonstrates nice scientific rigor by not just showing that the model solves a task (S4 permutations), but by verifying how it solves it. Through model introspection, it provides strong evidence that the network learns the underlying geometric structure of the problem (the isomorphism to a cube's rotation group) exactly as predicted by theory.

Weaknesses :-

- The model's core design using Householder products enforces a strict norm ||A(xi)|| <= 1 on its state-transition matrix. While this guarantees stability, it is also a fundamental expressive bottleneck that prevents the model from performing norm-increasing operations, which are crucial for amplifying signals and are a key reason why competing models like RWKV-7 achieve stronger theoretical guarantees.

- Increasing the model's expressivity via the hyperparameter nh (the number of Householder products) is inefficient. It leads to a linear growth in both model parameters and, more critically, the sequential computation time of the recurrence, making it nh times slower per token. This architectural choice limits the practical scalability of using a large nh to achieve higher performance.

- The number of operations, nh, is a fixed, global hyperparameter. The model applies the same amount of costly computation to every token, regardless of its importance. This is wasteful for simple tokens (e.g., punctuation) and may be insufficient for complex ones (e.g., resolving a long-range dependency), representing a missed opportunity for adaptive computation that could allocate resources more efficiently.

---

> ### Author Rebuttal · Authors · 2025-07-30
>
> We thank the reviewer for the feedback and address each point below.
>
> > The model's core design using Householder products enforces a strict norm ||A(xi)|| <= 1 on its state-transition matrix. While this guarantees stability, it is also a fundamental expressive bottleneck that prevents the model from performing norm-increasing operations, which are crucial for amplifying signals and are a key reason why competing models like RWKV-7 achieve stronger theoretical guarantees.
>
> We agree this is a crucial trade-off. We intentionally designed DeltaProduct with Householder products to prioritize guaranteed stability ($\Vert A(x_i)\Vert \leq 1$), a criterion for stable training on long sequences. This design choice directly contrasts with the potential instability of models like RWKV-7. As we demonstrate in Appendix B.6, the RWKV-7 recurrence becomes unstable in the very cases that grant it higher expressivity (i.e., when vectors $a_i$ vary per token); we show that the product of just two such matrices can have a spectral radius > 1.2.
>
> However, the core product-of-matrices mechanism we propose is flexible. It can incorporate other matrix families, including the norm-increasing ones from RWKV-7  Indeed, a minor extension of our theory, building on Merrill et al. (2024) (Theorem 5.2) and Peng et al. (2025) (Lemma 3), shows that this generalized framework has stronger theoretical guarantees. For instance, a single-layer model using $n_h = n$ RWKV-7 matrix products can recognize any regular language accepted by a finite state automaton with $n$ states, while if $n_h \geq 2$, a three-layer model can recognize any regular language. We will add these results to the revised paper.
>
> To validate this, we ran new experiments with products of RWKV-7 type state-transition matrices. We used $(\mathbf{I} - 2 * \mathbf{k} (\mathbf{k} \odot \mathbf{a})^T)$ as state-transition matrix where $\mathbf{k}$ is unit norm and $\mathbf{a}\in [0, 1]$ by transforming through a sigmoid. We find that a single layer using products of 2 RKWV-7 matrices obtains comparable results to DeltaProduct$_2$ on $S_3$. We will add new experiments for the other permutation groups using products of RWKV-7 state-transition matrices to a revised version of our manuscript.
>
> > Increasing the model's expressivity via the hyperparameter $n_h$ (the number of Householder products) is inefficient. It leads to a linear growth in both model parameters and, more critically, the sequential computation time of the recurrence, making it nh times slower per token. This architectural choice limits the practical scalability of using a large nh to achieve higher performance.
>
> We see the linear increase in sequential computation with $n_h​$ as a tunable trade-off rather than a fixed limitation. Our experiments in Figure 8 show that a small increase in $n_h$​ (e.g., to 2 or 3) yields substantial gains in length extrapolation, making it a highly effective trade-off for many applications. Despite the theoretical linear increase in computational complexity, we find that in practice the training time increases sub-linearly with $n_h$ as shown in Figure 13 in Appendix C.4. The scaling-rate is also dependent on training sequence length and batch size and could be further improved by a custom triton kernel. After submission, we benchmarked the training throughput of the parameter-matched 805M parameter DeltaNet and DeltaProduct$_3$ on an improved implementation of our model with fused projections and causal 1d-convolutions (see also the response to reviewer RGKp). We find that DeltaProduct$_3$ achieves 86% (compared to our previous 75% without fusing projections and causal 1d-convolutions)  of the training throughput of DeltaNet at 4096 token sequence length and a per-device batch size of 12.
>
> > The number of operations, nh, is a fixed, global hyperparameter. The model applies the same amount of costly computation to every token, regardless of its importance. This is wasteful for simple tokens (e.g., punctuation) and may be insufficient for complex ones (e.g., resolving a long-range dependency), representing a missed opportunity for adaptive computation that could allocate resources more efficiently.
>
> Thank you for the suggestion. Adaptive computation is a promising direction that could greatly enhance the model's efficiency by allocating computation dynamically. While this falls outside the scope of our current work, which focuses on establishing the fundamental properties of the Householder product structure, we agree it is a valuable avenue for future research. We will add a discussion of adaptive computation, such as using a Mixture-of-Experts (MoE) approach to select $n_h$ per token, to our future work section.
>
>
> ### References:
> Merrill, W., Petty, J., & Sabharwal, A. (2024, July). The Illusion of State in State-Space Models. In International Conference on Machine Learning (pp. 35492-35506). PMLR.
>
> Peng, B., Zhang, R., Goldstein, D., Alcaide, E., Du, X., Hou, H., ... & Zhou-Zheng, C. (2025). Rwkv-7" goose" with expressive dynamic state evolution. arXiv preprint arXiv:2503.14456.

---

> > ### Comment · Reviewer_HNXg · 2025-08-05
> >
> > I thank the authors for their detailed and thoughtful rebuttal.
> >
> > Regarding my primary concerns:
> >
> > 1.  **On the Stability vs. Expressivity Trade-off:** The authors' response was helpful. They rightly framed the guaranteed stability (`||A(xi)|| <= 1`) of the Householder product as an intentional and important design choice, providing a clear counterpoint to models like RWKV-7 which can suffer from instability. The proposed extension to incorporate norm-increasing matrices (like those in RWKV-7) and the new preliminary experiments showing comparable performance are very convincing. This generalization elevates the contribution from a specific architecture to a more flexible and powerful framework. I support the inclusion of these new theoretical and experimental results in the revised manuscript, as they turn a perceived limitation into a significant strength.
> >
> > 2.  **On Computational Inefficiency:** The authors' new throughput benchmarks are very helpful. The updated finding that an improved DeltaProduct-2 model can achieve 86% of DeltaNet's throughput demonstrates that the practical cost of increasing `nh` is manageable and represents a worthwhile trade-off for the performance gains observed. This alleviates my concerns about the practical scalability of the approach for small, effective values of `nh`.
> >
> > 3.  **On Adaptive Computation:** I accept the authors' position that implementing adaptive computation is a promising avenue for future work and falls outside the scope of this foundational paper. Their commitment to discussing this limitation and future direction is appropriate.
> >
> > The authors' rebuttal, particularly the new experiments and the generalization of their framework, has addressed my initial concerns. The planned revisions will substantially improve the paper's quality and significance. Therefore, I have updated my recommendation.

---

### Decision · Program_Chairs · 2025-09-17

**Decision:**

Accept (poster)

**Comment:**

***Summary:***

Linear recurrent neural networks are an area of active study for language models because they avoid the quadratic complexity of dense softmax attention in transformers.  Earlier models such as Mamba or GLA use a diagonal transition matrix, which limits their expressivity.  More recent models such as DeltaNet and RWKV-7 improve on certain tasks by using a diagonal plus rank-1 transition matrix.  In this paper the authors introduce a new LRNN, DeltaProduct, which extends DeltaNet by taking multiple steps per token.  The state-transition matrix thus becomes rank-N rather than rank 1, and is formed as a product of N generalized Householder transformations.

The paper provides both mathematical proofs, and experimental results which demonstrate that the technique works.

***Meta-review.***

The reviewers were unanimously in favor of accepting this paper, and I see no reason to disagree.  The content of the paper lies outside my area of expertise, but it seems to be well-written.  The reviews also seem to be careful and thorough.

Reviewer HNXg thought that "the paper demonstrates nice scientific rigor", but was concerned that the use of householder products was a "fundamental expressive bottleneck."  However, the authors provided new throughput benchmarks, and a detailed explanation of the "stability vs. expressivity tradeoff" in their rebuttal, which addressed the reviewer's concerns.

Reviewer RGKp had nothing but praise:
* "This paper demonstrates exceptional scientific rigor with a clear, meaningful contribution that advances linear RNN research."
*  "The related work section is particularly thorough and well-organized."

Reviewer bkha did an exceptionally detailed review (thank you!), went through the paper and the proofs line-by-line, and found some errors in key theoretical results. However, the authors provided updated proofs during the rebuttal period.  In their final justification, reviewer bkha writes: "Mistakes pointed out before rebuttal have been addressed and corrected."

Given the attention to detail with which bkha reviewed this paper, I feel confident that the paper was fairly reviewed, and should be accepted.  However, given that none of the reviewer scores were above 5, it probably does not qualify for a spotlight or oral.